

# Future diversity and lifespan of metazoans under global warming and oxygen depletion

Kunio Kaiho

Department of Earth Science, Graduate School of Science, Tohoku University, Sendai, 980-8578, Japan

*Correspondence to*: Kunio Kaiho (kunio.kaiho.a6@tohoku.ac.jp)

**Abstract.** The diversification of metazoans, from cnidarians to vertebrates, began approximately 700–500 million years ago and has been shaped by dynamic environmental changes. Recurrent climate fluctuations—driven by large-scale volcanism, meteorite impacts—have caused major shifts in biodiversity. Understanding these historical patterns provides a critical foundation for projecting future biodiversity trends amid ongoing and future climate change. Building on these insights, this
study integrates additional environmental drivers—including icehouse and greenhouse states, solar luminosity–induced warming, gradual declines in atmospheric carbon dioxide and oxygen, plant-related crises, and anthropogenic influences—to model future metazoan diversity across ecosystems. The results suggest that metazoans will undergo complete extinction approximately 700 million years from now—300 to 400 million years earlier than previously estimated. Over the next 400 million years, biodiversity is projected to fluctuate through cycles of mass extinction and recovery. Beyond this, increasing
solar luminosity is expected to raise peak global temperatures at ~300-million-year intervals, while declining oxygen and carbon dioxide levels will impose increasing physiological stress. These gradual changes will drive a progressive loss of biodiversity, even in the absence of distinct extinction events. Ultimately, a final extinction—likely triggered by large-scale volcanism or a meteorite impact—will eradicate all remaining metazoans. The total lifespan of metazoans on Earth is thus projected to be approximately 1.4 billion years, representing about 12% of Earth's anticipated 12-billion-year lifespan.
Humanity currently occupies the midpoint (~50%) of this evolutionary timespan.

## 1 Introduction

The long-term warming trend, primarily driven by the gradual increase in solar luminosity, is a critical determinant of Earth's future climate trajectory (Franck et al., 2006; Abe et al., 2011; O'Malley-James et al., 2013; Mello and Friaça, 2019; Ozaki
and Reinhard, 2021). As nuclear fusion progresses within the Sun, the amount of energy received by Earth in the form of light and heat steadily increases. Over the next 1.1 billion years (Gyr), global average surface temperatures are projected to rise from the current 14 °C to approximately 40 °C (Mello and Friaça, 2019), while atmospheric and oceanic oxygen levels may decline to just 1% of present atmospheric levels (PAL) (Ozaki and Reinhard, 2021). By approximately 1.6 Gyr into the



future, surface temperatures could exceed 100 °C, rendering the planet uninhabitable for all known forms of life (Mello and
Friaça, 2019).

Global warming will accelerate terrestrial weathering, depleting atmospheric $CO_2$ and triggering a cascade of ecological
crises. Initially, $C_3$ plants will experience a severe decline, followed by $C_4$ plants, ultimately leading to significant reductions
in metazoan diversity (Reinfelder et al., 2000, 2004; Mello and Friaça, 2019; Ozaki and Reinhard, 2021). However, the
precise timeline for metazoan extinction remains uncertain.

Superimposed on this long-term warming trend is a cyclical climate rhythm with a periodicity of approximately 0.3 Gyr
(Scotese et al., 2021; Torsvik et al., 2024; Vérard, 2024). These cycles are primarily driven by large-scale tectonic and
mantle dynamics, including continental amalgamation through oceanic seafloor subduction, which leads to the termination of
oceanic basins (e.g., Wilson cycles) and the formation of supercontinents (Heron, 2018). Such cycles influence global
temperatures, ocean circulation, and atmospheric composition, persisting as long as mantle convection remains active and
liquid oceans exist (Mello and Friaça, 2020). However, these cyclical climate rhythms have not yet been fully incorporated
into projections of future surface temperatures.

In addition to long-term trends and cyclical rhythms, abrupt climate events—including large-scale volcanic eruptions
and meteorite impacts—have historically triggered mass extinctions. These events release substantial amounts of $SO_2$, soot,
and $CO_2$, causing short-term global cooling followed by prolonged warming (Bond and Grasby, 2017; Kaiho, 2024). Over
the past 0.5 Gyr, such events have repeatedly led to abrupt temperature fluctuations and significant biodiversity loss (Erwin
et al., 1987; Kaiho, 2022). Despite their profound impact, these abrupt climate shifts have not yet been factored into long-
term climate projections.

This study integrates seven critical factors—the anthropogenic crisis (Waters et al., 2011; Ceballos et al., 2015; Waters
et al., 2016; Kaiho, 2022, 2023), long-term warming trends, cyclical climate rhythms, abrupt climate events, $C_3$ plant crises,
$C_4$ plant crises, and atmospheric oxygen depletion—to project metazoan extinction dynamics over the next 1.5 billion years.
Projections are based on temperature modeling, thermal tolerance limits, oxygen modeling, and metazoan diversity trends.
As solar luminosity increases over time, intensified weathering processes are expected to lower atmospheric $CO_2$ levels,
leading first to a crisis for $C_3$ plants, including trees, followed by a more severe decline affecting $C_4$ plants.

Metazoans, which emerged and diversified approximately 700–500 million years ago, evolved from simple cnidarian-
like organisms into complex forms such as arthropods and vertebrates (Erwin, 2015; Kaiho et al., 2024). Experimental data
on the thermal tolerance of metazoans provide a foundation for estimating their potential longevity. By integrating these data
with projected temperature changes, this study evaluates diversity trends and extinction timelines across various ecological
niches, including superterranean, surface-water, subterranean, and deep-sea habitats.



## 2 Methods

### 2.1 Method summary

To illustrate the lifespan of metazoans on Earth, we conducted a multi-step analysis incorporating past and future climate dynamics:

1. Past Records: To estimate future metazoan diversity, we compiled records of global surface temperatures during the five largest mass extinctions, analyzed the relationship between atmospheric oxygen levels and biodiversity, and examined the number of insect, terrestrial tetrapod, and marine metazoan families.

2. Future Global Surface Temperature Projections: Assuming that the icehouse-greenhouse cycle and major mass extinctions continue at the same pace as in the past (0.3 Gyr and 0.094 Gyr, respectively), we calculated future global surface temperatures (Figure 1).

3. Thermal Tolerance of Metazoans: The relationship between global average surface temperature and the local extinction temperature for superterranean (St), subterranean (underground) (U), surface-water (Sw), and deep-water (D) metazoans was examined (Figure 2). This relationship was used to estimate the global average surface temperature at which metazoan extinction occurs across low, mid, and high latitudes, as presented in Figure 1.

4. Oxygen Levels and Biodiversity Relationships: Using data from the Neoproterozoic and Paleozoic eras, we analyzed the relationship between atmospheric oxygen concentration and the diversity of marine metazoans, terrestrial tetrapods, insects, and terrestrial plants (Oxygen Concentration Data: Krause et al., 2018; Sperling et al., 2015; Biodiversity Data: Erwin et al., 1987; Labandeira and Sepkoski, 1993; Engel and Grimaldi, 2004; Benton, 2010) (Figure 3).

5. Future Metazoan Diversity Calculations: Based on a framework that includes temperature trends, oxygen levels, and $C_3$-$C_4$ plant crises (Figure 4), we predicted future metazoan biodiversity trends and illustrated them in Figures 1, 5, and 6.

We considered five past events, one anthropogenic crisis, and 16 future climate events in this study. The five past events correspond to the largest mass extinctions ("big five" mass extinctions, labeled as events -5 to -1), while event 0 represents the ongoing anthropogenic crisis. Events 1 to 16 account for abrupt large-scale climate changes related to major mass extinctions or subsequent climate shifts occurring after the complete extinction of metazoans.

Gradual extinctions, driven by long-term cyclic global warming, are projected to occur over extended periods, resulting in a continuous decline in biodiversity. This process is illustrated by the orange curve in Figure 1. The estimated duration of temperature anomalies and extinction-recovery events is 0 Gyr and 0.05 Gyr, respectively, based on geological data (Erwin et al., 1987).



## 2.2 Past records on climate cycles

Historical data on long-term cyclical icehouse and greenhouse phases were obtained from Vérard (2024) for the period between -1.0 and -0.6 Gyr and from Scotese et al. (2024) for the period between -0.6 and 0 Gyr (Fig. 1, Table A1). To estimate future abrupt climate changes and biotic crises, we selected the five largest mass extinctions in Earth's history. Global temperature anomalies associated with these extinction events were sourced from Kaiho (2022, 2024), who compiled data from Finnegan et al. (2011), Balter et al. (2008), Huang et al. (2018), Chen et al. (2016), Korte et al. (2009), and 95 Vellekoop et al. (2014) (Table A2). These data are applied in section 2.2.

## 2.3 Past records on oxygen levels and metazoan diversity

Projections of future atmospheric oxygen levels and marine metazoan diversity rates were derived from past records. The historical values include 0.01 PAL and a 0.04 marine metazoan diversity rate in the late Paleozoic marine diversity at the early Ediacaran (-0.6 Gyr), 0.10 PAL and 0.2 marine metazoan diversity rate at the late Ediacaran (-0.55 Gyr), 0.24 PAL and 100 0.4 marine metazoan diversity rate at the early Cambrian Explosion (-0.52 Gyr), 0.38 PAL and 1.0 marine metazoan diversity rate at the end-Ordovician (-0.44 Gyr), and 0.86 PAL and 1.0 marine metazoan diversity rate at the end-Silurian (-0.42 Gyr). These values are based on atmospheric oxygen reconstructions from Krause et al. (2018), supported by findings from Sperling et al. (2015) (Fig. 1, Table 1). The relationships between atmospheric oxygen levels and the diversity of marine metazoans, terrestrial tetrapods, insects, and terrestrial plants are illustrated in Figure 3.

This figure, based on fossil records and estimated oxygen levels, suggests that there is a positive relationship between atmospheric oxygen levels and biodiversity, with significant declines occurring when oxygen levels drop in both marine and terrestrial metazoans. Additionally, terrestrial metazoans appear to require higher oxygen levels, likely due to their reliance on an ozone layer for evolutionary adaptation. These interpretations are further applied in Section 2.4.

## 2.4 Past and future extinction percentages and the interval

Extinction percentages at the family level during major mass extinctions indicate significant biodiversity losses across various groups. Marine metazoan extinction percentages were recorded as 22% at the end-Ordovician, 21% during the Late Devonian, 50% at the end-Permian, 20% at the end-Triassic, and 15% at the end-Cretaceous (Sepkoski, 1982). Terrestrial tetrapods experienced extinction rates of 54% at the end-Permian, 21% at the end-Triassic, and 38% at the end-Cretaceous (Table A3; Benton, 2010). Insects faced extinction rates of 35% at the end-Permian, 14% at the end-Triassic, and 8% at the 115 end-Cretaceous (Table A4; Labandeira and Sepkoski, 1993).

      Averaging these extinction percentages, we calculated future diversity loss projections based on 26% for marine metazoans (0.74 survival rate), 37% for terrestrial tetrapods (0.63 survival rate) and 19% for insects (0.18 survival rate, Table 2). These estimates are applied in section 2.5 to model long-term biodiversity changes under future climate conditions.



## 2.5  Temperature estimation

To project future global temperatures, we obtained global average surface temperature data over a 2.5 Gyr timeline, encompassing both historical and projected changes (orange curve, and red and blue dots in Fig. 1). This analysis considers three primary physical factors influencing past and future temperatures: long-term warming trends, cyclical climate rhythms, and abrupt climate events.

### 2.5.1  Future global average surface temperature based on trends and cycles

The future long-term warming trend follows the projections of Mello and Friaça (2019), aligning with earlier studies by Franck et al. (2006) and Wolf et al. (2017). Earth's climate history has exhibited a cyclical pattern of approximately 0.3 Gyr since -1.0 Gyr (Fig. 1). This cycle consists of extended greenhouse phases lasting around 0.2 Gyr, followed by shorter icehouse phases of approximately 0.1 Gyr (Scotese et al., 2021; Torsvik et al., 2024; Vérard, 2024). To incorporate this climate cycle, we applied an 8°C temperature anomaly between icehouse and greenhouse periods (Bond and Grasby, 2017)

to Mello and Friaça's model. Starting from the current global average surface temperature of 14°C during an icehouse period, we projected temperature variations into the future (Table A1, red curve in Figure 1). The global average surface temperature during non-event periods was defined using the equation:

$$T_n = T_t + \Delta T_c \tag{1}$$

where $T_n$ is the global average surface temperature during non-events (orange curve in Figure 1), $T_t$ represents the climate

trend, and $\Delta T_c$ is the temperature anomaly associated with the climate cycle (8°C during greenhouse and 0 °C during icehouse periods). Data intervals were set at 0.05 Gyr in Figure 1.

### 2.5.2  Future global average surface temperature incorporating trends, cycles, and events

Short-term climate fluctuations (<1 million years) associated with mass extinction events, such as the Late Ordovician, Permian-Triassic, and Cretaceous-Paleogene extinctions, have significantly impacted global temperatures (Kaiho, 2022).

Global temperature anomalies for these events were derived from global sea surface temperature (SST) anomaly data as estimated by Kaiho (2022).

For future mass extinction events, temperature anomalies during cooling ($\Delta T_{ec}$) and warming ($\Delta T_{ew}$) phases were calculated using the following equations, with light blue and red dots in Figure 1 representing their respective values:

$$T_{ec} = T_t + \Delta T_c + \Delta T_{ec} \tag{2}$$

$$T_{ew} = T_t + \Delta T_c + \Delta T_{ew} \tag{3}$$



The values of $\Delta$Tec and $\Delta$Tew (light blue and red dots in Figure 1) were estimated based on the average temperature anomalies observed during the five largest mass extinctions, excluding the unknown cooling impact of the Permian-Triassic extinction. Although $CO_2$ emissions during these events are substantial, their influence on long-term temperature trends is minimal due to their relatively short duration (<10,000 years) compared to the multi-million-year timescale of global climate change.

### 2.5.3 Mantle temperature influence on $CO_2$ and $SO_2$ emissions

Future declines in mantle potential temperature, as modeled by Mello and Friaça (2019), were used to estimate $\Delta$Tec. Decreasing mantle temperatures are expected to reduce $CO_2$ and $SO_2$ emissions from sedimentary rocks in large igneous provinces (Kaiho, 2024). The temperature changes and emission rates were calculated using experimental data from Kaiho et al. (2022) (Table A2), with the following equations:

$$\Delta Tec = -8.9 - (-8.9 + 8.9 \times SR) \times 2/3 \tag{4}$$

$$\Delta Tew = 9.9 \times CR \tag{5}$$

$$SR = (0.00108Ts - 0.236)/42.25 \ (r = 0.97) \tag{6}$$

$$CR = \frac{(0.501 \times p)}{0.348}, \ p = e^{0.0077Ts} \ (r = 0.90) \tag{7}$$

$$Ts = (E \times Ti)/(E - Ti \times \ln t_i/t_s) \ \text{(from the Arrhenius equation)} \tag{8}$$

$$Ti = Tm \times 1423/1603 \tag{9}$$

where SR and CR represent the future rates of $SO_2$ and $CO_2$ emissions relative to present values, based on experimental heating data (Kaiho et al., 2022). Ts is the sill temperature after 100 years, Ti is the initial heating temperature of the sill, and $t_s$ and $t_i$ are actual and experimental heating durations, respectively. The activation energy E is 74 kcal/mol for $n$-$C_{16}$ alkane (a representative $CO_2$ source) and 67 kcal/mol for pyrite (a representative $SO_2$ source).

Using emission models from Black et al. (2018) for $CO_2$ and Schmidt et al. (2016) for $SO_2$, we estimated the impact of declining emissions on global temperature. A 50% reduction in $CO_2$ and a 67% reduction in $SO_2$ emissions were found to significantly decrease surface temperature anomalies over the next 1.5 Gyr.

### 2.5.4 Error calculation

The total error was estimated using the equation:

$$Er = SD + Er \ (\Delta Tc) \tag{10}$$





where SD is the standard deviation of ΔTec or ΔTew , and Er (ΔTc ) accounts for uncertainties in icehouse phase timing. While transitions between icehouse and greenhouse phases may vary, the overall error remains within 0.1 Gyr, which is less than the estimated ~0.1 Gyr duration of icehouse conditions.

## 2.6  Upper tolerance limit of temperature for metazoans

The upper thermal tolerance limit for terrestrial ectotherms (e.g., amphibians, reptiles, and arthropods) ranges from approximately 45–47°C in low to mid-latitudes (60°N–60°S) to 35–40°C in high latitudes (Sunday et al., 2011; Araújo et al., 2013). Marine ectotherms (e.g., mollusks, fish, and arthropods) exhibit upper tolerance limits spanning 40–45°C between 35°N and 50°S, decreasing to 10–20°C at higher latitudes (Sunday et al., 2011). While cold tolerance varies significantly

among species, thermal tolerance among ectotherms and endotherms (e.g., mammals and birds) is relatively conserved across lineages (Araújo et al., 2013).

As a result, the upper survival limits for metazoans are estimated at 45–47°C on land, 40–45°C in sea surface waters, and 42–46°C across both terrestrial and marine environments. At these temperatures, essential cellular functions—including metabolic activity and membrane integrity—are severely compromised, ultimately threatening metazoan survival (Somero,

1995; Pörtner, 2002).

Based on these thermal tolerance limits, extinction threshold temperatures for superterranean (St) metazoans, subterranean (underground) metazoans (Ut), surface-water (Sw) metazoans, and deep-water metazoans (Dw) across different latitudes were calculated using Figure 2. These thresholds provide critical insights into how rising global temperatures may impact metazoans across various ecological niches.

### 2.6.1  For superterranean (St) metazoans

The model for St metazoans in Figure 2a focuses on oceanic climate regions to assess conditions leading to their complete extinction. The Global Average surface Temperatures required for the Extinction (GATE) of St metazoans in Figure 1 are determined as follows (Fig. 2a):

1.  The upper tolerance limit of temperature for each metazoan group is set at 46 °C, representing the average

maximum daily temperature in the warmest month for 0°, 30°, 60°, and 90° latitudes (closed dots in Figure 2a).

2.  Thick oblique lines with a 15°C gradient between 0° and 90° latitudes are drawn for warm Earth conditions (Zhang et al., 2019; Burgener et al., 2023), including the late Paleocene-early Eocene and mid-Cretaceous warmest periods. These lines intersect the closed dots (thick oblique lines in Figure 2a).

3.  The oblique lines are adjusted to the Local Monthly Maximum Temperature (LMMT) by decreasing them by 5°C,

based on the Local Daily Maximum Temperature in oceanic climate regions during the warmest month (LDMT). Data from warm coastal cities, such as Shanghai (~30°N) and Singapore (~0°N), are used as modern Earth references (Weather Spark, https://weatherspark.com) (dashed oblique lines in Figure 2a).





4. The oblique lines are further adjusted to the Local Annual Temperatures (LAT) using a 1°C and 5°C difference between LMmT and LAT at ~0°N and ~60°N, respectively (Upchurch et al., 1998) (thin oblique lines in Figure 2a).

5. The GATE values for St, Sw, Ut, and Dw metazoans (GATES, GATEU, and GATED) are determined based on LAT at 37° latitude. Temperature horizontal lines for TES, TEU, and TED are then drawn in Figure 1. The GATE values for St and Sw metazoans are the same as described in Section 2.3.3 (open dots in Figure 2a).

The LAT for the extinction of St metazoans at 0°, 30°, 60°, and 90° latitudes is determined using the equation:

$$LAT = 46 – 5 – \Delta LT \ (°C) \tag{11}$$

### 2.6.2  For subterranean (underground terranean, Ut) metazoans

The GATEU values for subterranean metazoans were determined based on Figure 2b. While present-day subterranean organisms primarily inhabit depths of around 10 cm, future subterranean metazoans capable of survival are expected to retreat to depths of approximately 2.5 meters.

The temperature difference between the warmest month and the annual average at a depth of 2.5 m is set at 2°C (Singh and Sharma, 2017). Oblique lines were drawn crossing points located 2°C below the closed dots in Figure 2b.

### 2.6.3  For surface-water (Sw) metazoans

The difference between the Local Daily Maximum Temperature (LDMT) and the Local Monthly Maximum Temperature (LMMT) in the sea surface is less than 1°C. Sea Surface Temperature (SST) is approximately 5°C higher than terrestrial surface temperatures at all latitudes during the warmest periods (Zhang et al., 2019).

As a result, the LMMT of the sea surface is assumed to be equal to the LDMT of land. Similarly, LAT (Local Annual Temperature) and GAT (Global Annual Temperature) in the sea surface are considered equivalent to those in oceanic climate land regions (Upchurch et al., 1998; Fig. 2c). Therefore, GATES values are common for both St and Sw metazoans.

### 2.6.4  For deep-water (Dw) metazoans

Deep-sea water temperatures remain stable throughout the year. These temperatures are derived as the median annual surface temperature in both high- and low-latitude regions. The reference value is based on the purple closed dot (45° latitude, 46°C) in Figure 2d, which represents conditions during the warmest period of the Mesozoic–Cenozoic (mid-Cretaceous; Burgener et al., 2023).

The GATED value corresponds to a 48°C global surface temperature, represented by the purple open dot in Figure 2d.





## 2.7 Future metazoan diversity estimation

Future events influencing metazoan diversity include the anthropogenic crisis ("event 0"), mass extinction events (events 1–11), $C_3$ and $C_4$ plant crises, and gradual oxygen depletion (Table 2). The future diversity of metazoans before extinction events, immediately after events, and following recovery is calculated for insects, terrestrial tetrapods, and marine metazoans using the following equations:

Diversity due to event: $D_t = D_{t-1} \times SRC \times SRO$          (12)

Diversity due to recovery: $D_t = D_{t-1} + (D_{t-2} - D_{t-1}) \times RR \times SRO$     (13)

Diversity just before event: $D_t = D_{t-1} \times SR \times SRO$        (14)

where, $D_t$ represents the diversity of each metazoan group at time t, and $D_{t-1}$ is the diversity of the same group in the previous time step. SRC represents the Survival Rate associated with Climate change events, including mass extinctions and $C_3$–$C_4$ plant crises, while SRO denotes the Survival Rate in response to declining atmospheric Oxygen levels (Table 2). RR

refers to the Recovery Rate, and SR represents the Survival Rate for gradual environmental changes (underlined numbers in Table 2). The time step t consists of an extinction event, the subsequent recovery period, and the interval before the next extinction event, applied across each cycle from event 0 to event 16 (Table A5).

### 2.7.1 SRC

Before mass extinction events, SRC values follow gradual warming trends as described in Table 2. In the case of event 0,

which represents the Anthropogenic Crisis, the maximum SRC values are 0.95 for insects, 0.70 for terrestrial tetrapods, and 0.90 for marine metazoans. These values apply when a full-scale nuclear war occurs, accompanied by medium anthropogenic pollution, deforestation, and global warming (Kaiho, 2023). If no such nuclear conflict occurs, the maximum SRC remains at 1.

For events 1–4 and 7, the SRC values are 0.81 for insects, 0.63 for terrestrial tetrapods, and 0.74 for marine metazoans,

calculated from averaged extinction percentages in Table A3 and A4. In events 5 and 6, where global surface temperatures reach approximately 40°C, the SRC values decrease to 0.6 for insects, 0.4 for terrestrial tetrapods, and 0.5 for marine metazoans, based on estimated Survival Area Rate (SAR) in Figure 1.

The SRC (survival rate by climates) is calculated as:

$SRC_T = (StR \times SAR_S + UR \times SAR_U) \times FS$       (15)

$SRC_M = (SwR \times SAR_S + DR \times SAR_D) \times FS$       (16)



where, $SRC_T$ and $SRC_M$ are terrestrial SRC and marine SRC, respectively, StR and SwR are Superterranean and Surface-water Metazoan Rates, respectively, UR is Underground Metazoan Rate, DR is Deep-water Metazoan Rate, $SAR_S$ is Survival Area Rate for St and Sw metazoans, $SAR_U$ and $SAR_D$ are Survival Area Rate for U and D metazoans, respectively, and FS is food scarcity. The SAR is obtained from red dots with extinction thresholds shown in Figure 1 (see Table 2 caption). The $SAR_D$ is influenced not only by temperature but also by dissolved oxygen levels. Higher water temperatures lead to lower oxygen conditions in deep water, as seen during the end-Permian and end-Cenomanian anoxia-euxinia events, which coincided with the extinction of deep-water fauna despite high atmospheric oxygen levels (Sun et al., 2012; Kaiho et al., 2013; 2016a). Therefore, $SAR_D$ should approximate $SAR_S$. Accordingly, $SAR_D$ is set equal to $SAR_S$ for Events 5–16 and for non-events after Event 8, as these are characterized by temperatures comparable to or exceeding those of the end-Permian (Table 2 caption).

Beyond direct climate impacts, food shortages will exacerbate extinction pressures. As plants and primary producers vanish, only organisms feeding on bacteria or sedimentary organic matter (along with their predators) will persist. As a result, survival rates are expected to decline further, with an additional reduction of 0.1–0.5 due to food scarcity (FS) for abrupt events at Events 8–10 due to demise of SS metazoans (Fig. 1).

The UR is 0.05, based on mammalian lineage data (15 families out of a total of 315; Recknagel & Trontelj, 2021; Benton, 2010). The tetrapod family survival rate, estimated from survival areas on GATEU, is 0.32 (Fig. 1). For deep-sea species, 0.33 represents the proportion of deep-sea fish families among all fish families, derived from an estimated ~6% of teleost species being restricted to deep-sea habitats (depths >200 m; Miller et al., 2022). This proportion corresponds to a 0.94 extinction rate at the species level and a 0.67 extinction rate at the family level, as adapted from Kaiho et al. (2022). For Events 11–16, SRC values are calculated similarly and presented in Table 2.

The diversity of terrestrial tetrapods is expected to decline significantly during the $C_3$ plant crisis, projected to occur around $0.4 \pm 0.2$ Gyr. An estimated 50% reduction in tetrapod diversity is expected within tree-dependent ecosystems, based on mid-range estimates of the crisis (Figs. 5 and 6). This estimate is informed by bird lineage studies, where twenty orders rely on trees ($C_3$ plants) for survival, while another twenty orders do not (Braun and Kimball, 2021).

Approximately 40 million years after the extinction of $C_3$ plants, $C_4$ plants are expected to evolve into tree-like forms, leading to the diversification of metazoans that depend on these new plant species, similar to the Devonian terrestrial plants. These metazoans are projected to originate from species that previously relied on $C_3$ plants. The estimated recovery rate ranges between 0.3 and 0.8, reflecting the challenges of re-establishing tree-dependent ecosystems from $C_4$ plant species.

In marine environments, both $C_3$ and $C_4$ photosynthesis systems coexist (Reinfelder et al., 2000, 2004). To estimate the effects of the $C_3$ plant crisis on marine metazoan diversity, the same reduction and recovery percentages applied to terrestrial tetrapods are tentatively used in two outlined scenarios.

The timeline of the $C_4$ plant crisis is projected to align with the complete extinction of metazoans, expected to occur at $0.97 \pm 0.2$ Gyr, coinciding approximately with event 11. As a result, the extinction of metazoans is set at 0.97 Gyr.





Biodiversity loss due to the C₄ plant crisis is estimated at approximately 98%, affecting both SS and UD metazoans, as
the complete loss of plant life would lead to widespread food shortages. The remaining 2% of surviving metazoan families,
without recovery, are expected to include species independent of plant-based ecosystems, such as those that feed on bacteria
and detritus (Cosson and Soldati, 2008) or those that inhabit deep-sea hydrothermal vents (Miroshnichenko, 2004; Kelley et
al., 2005). The projected reduction is attributed to the reliance of most metazoans on plant-dependent food chains.

### 2.7.2 SRO

The Survival Rate in response to declining atmospheric oxygen levels (SRO):

For terrestrial tetrapod and insect diversities:

During the next 0.5 Gyr: SRO = 1.0                                              (17)

During the next 0.5–0.8 Gyr: SRO = 2.67 – 3.33T                                 (18)

During the next 0.8–1.0 Gyr: SRO = 0.0                                          (19)

For marine metazoan diversity:

During the next 0.5 Gyr: SRO = 1.0                                              (20)

During the next 0.5–0.8 Gyr: SRO = 2 – 2T                                       (21)

During the next 0.8–1.0 Gyr: SRO = 1.84 – 1.80T                                 (22)

where T is numerical age in Gyr (Table 2). These data are derived from Ozaki and Reinhard (2021) (Table 1). Additional
details are provided in SI Text 1.

Atmospheric oxygen levels are projected to decline over the next 1.1 Gyr. Starting from an atmospheric oxygen level of
1.0 PAL at -0.1 Gyr, levels are expected to gradually decrease to 0.5 PAL (median range: 0.3–0.7) by 0.5 Gyr, followed by a
further decline to 0.3 PAL (median range: 0.07–0.5) at 0.8 Gyr. A rapid decrease follows, with oxygen levels dropping to
0.01 PAL between 1.0 and 1.1 Gyr. These projections are based on the median atmospheric oxygen levels reported by Ozaki
and Reinhard (2021) (Table 1).

Using future oxygen level projections and interpretations of the relationships between oxygen levels and metazoan and
plant diversities (Fig. 3), we estimate the oxygen depletion effect on metazoan diversity.

### 2.7.3 RR and SR

The Recovery Rate (RR) varies depending on the event. For Event 0, RR is 1.0 in the case of maximum recovery but drops
to 0.0 if metazoans continue to decline due to human influence on the biosphere. During Events 1–7, RR remains at 1.0,
allowing full recovery. However, for Events 8–10, RR ranges between 0.01 and 0.3, indicating a significantly reduced
capacity for biodiversity restoration. The C₃ plant crisis results in an RR between 0.3 and 0.8, depending on the severity of
ecosystem disruption.





In contrast, the $C_4$ plant crisis, Events 11–13, and the oxygen depletion event lead to an RR of 0.01–0.1. Events 14–16
indicate no recovery, as biodiversity falls beyond the extinction threshold, making restoration impossible both during and
after these events.

At the family level, the evolutionary RR for tetrapods is expected to range from 0.01 to 0.3, constrained by the
adaptation and diversification challenges faced by subterranean metazoans. These organisms, which depend on bacterial and
sedimentary organic matter, are typically found at depths greater than 1 meter in complete darkness.

During events 8–10 and the aftermaths, the family-level RR for deep-water metazoans is also projected to be limited to
0.01–0.3. This limitation is due to large uninhabitable low-latitude regions during non-events (Fig. 1) and evolutionary
barriers affecting deep-water metazoans, which rely on bacteria and sedimentary organic matter at depths exceeding 400
meters. The minimal seasonal temperature variations (<2°C) and perpetual darkness at these depths further restrict their
diversification and recovery potential.

The Surviving Rate (SR) is influenced by gradual warming and is used to assess diversity just before each event. For
Events 0–7, SR remains at 1.0 but declines to 0.85 and 0.95 for terrestrial and marine metazoans, respectively, at Event 8. It
further decreases to 0.68 and 0.84 at Events 9 and 10. At Events 11–13, SR drops significantly to 0.003–0.015 and 0.025–
0.125 due to global surface temperature increases during non-extinction periods when SS metazoans are absent, as shown in
Figure 1. These values represent the proportion of the Earth's surface where temperatures exceed the upper tolerance limit
for metazoans (Fig. 1).

### 2.7.4 Variation of scenarios

The maximum and minimum diversity values for marine metazoans, terrestrial tetrapods, and insects are determined using
these equations and are presented in Table A5. These diversity ranges account for uncertainties in factors such as the
anthropogenic crisis, global surface temperature, survival rate, recovery rate, and oxygen level (Table 2).

Figures 5 and 6 illustrate the uncertainties in future metazoan diversity changes under different scenarios:

1.  No full-scale nuclear war, with minimum recovery rates after the $C_3$ plant crisis (Case 1 in Table A5, Fig. 5).
2.  Full-scale nuclear war, with an initial recovery rate of 1.0 followed by minimum recovery rates after the $C_3$ plant
    crisis (Case 2 in Table A5, Fig. 5).
3.  No full-scale nuclear war, with maximum recovery rates after the $C_3$ plant crisis (Case 3 in Table A5, Fig. 5).
4.  Full-scale nuclear war, with continued human influence on the biosphere leading to prolonged biodiversity decline
    (Case 4 in Table A5, Fig. 6).



## 3 Results

### 3.1 Projected changes in global average surface temperature

The orange curve in Figure 1 illustrates projected changes in global average surface temperature, incorporating long-term
trends and cyclical climate rhythms while excluding abrupt climate impacts associated with mass extinction events. The
timeline spans from -1.0 Gyr to 1.5 Gyr into the future, with data points plotted at 0.05 Gyr intervals.

During the Phanerozoic era (-0.54 Gyr to the present), global average surface temperatures fluctuated between 15 and
25°C, a range projected to persist until approximately 0.35 Gyr. Beyond this point, temperatures are expected to rise to 25–
35°C by 0.65 Gyr and further increase to 30–43°C between 0.65 and 0.95 Gyr. From 0.95 to 1.2 Gyr, global temperatures are
projected to reach 43–50°C, ultimately rising to 50–80°C by 1.5 Gyr.

In Figure 1, light blue and red dots represent cooling and subsequent warming phases associated with abrupt climate
changes during major mass extinction events. Events 5, 8, 11, and 14 indicate four distinct temperature surges occurring
within <0.1 million years during the 2.5-Gyr study period.

Up to 0.4 Gyr, climate patterns—ranging from approximately 12 to 32°C at their minima and maxima—are expected to
resemble those of the Phanerozoic era (designated as Climate Phase A). From 0.4 to 0.7 Gyr (Climate Phase B),
temperatures are projected to rise, with minima and maxima averaging 18 to 36°C. Subsequent phases include 29–46°C
between 0.7 and 1.0 Gyr (Climate Phase C), 37–53°C from 1.0 to 1.25 Gyr (Climate Phase D), and 52–68°C from 1.25 to 1.5
Gyr (Climate Phase E). These climate phases are characterized by their maximum temperatures (red dots).

Climate Phases B–E are expected to be triggered by abrupt climate events occurring during greenhouse periods. These
events highlight the potential for sudden and significant shifts in surface temperature, emphasizing the interplay between
greenhouse conditions and catastrophic climate perturbations.

### 3.2 Metazoan diversity change

Future metazoan diversity is estimated using anthropogenic crisis (Kaiho, 2023) surface temperature trends, the $C_3$ plant
crisis, the $C_4$ plant crisis, and oxygen depletion effects (Figs. 5 and 6). Metazoan diversity is influenced by surface
temperature changes, plant crises driven by global warming–weathering–$CO_2$-loss processes, and atmospheric oxygen
depletion resulting from the same processes.

The timeline between -1.0 and 1.5 Gyr is divided into five metazoan phases: Ancestor phase (Climate Phase A),
Evolution with mass extinctions (Climate Phase A), Decline with mass extinctions (Climate Phase B), Demise by mass
extinctions (Climate Phase C), and Aftermath (Climate Phases D and E) (Figs. 5 and 6).





### 3.2.1 Metazoan evolution with mass extinctions

During climate phase A, five potential future mass extinction events (Events 1–5) are projected to cause significant diversity losses among insects, terrestrial tetrapods, and marine metazoans. These events follow the pattern of the "Big Five" mass extinctions (Events -5 to -1) that occurred between -0.5 to 0 Gyr. However, similar to past mass extinction cycles, biodiversity is expected to recover to pre-event levels after each extinction. This period is defined as the "evolution with mass extinction" phase.

Metazoan phases are determined by the relationship between maximum global temperatures (red dots in Figure 1) and survival thresholds (yellow-orange regions in Figure 1). Large volcanic eruptions and asteroid impacts are expected to trigger extreme global cooling, comparable to past mass extinction events, followed by prolonged periods of global warming.

Event 0, representing a full-scale nuclear war, may lead to minor mass extinctions. If the recovery rate is 1.0, Events 1–4, occurring between 0.03 and 0.40 Gyr into the future, will maintain biodiversity at 610 insect families, 315 terrestrial tetrapod families, and 950 marine metazoan families. However, during each extinction event, these numbers will abruptly decline to 494, 198, and 703 families, respectively, before recovering (Fig. 5).

If the recovery rate of Event 0 is 0.0, leading to the continued decline of metazoans due to human influence on the biosphere, biodiversity loss will be more severe. In this scenario, during non-event periods, the estimated numbers of surviving families will be 580 for insects, 221 for terrestrial tetrapods, and 855 for marine metazoans (Fig. 6).

### 3.2.2 Metazoan decline with mass extinctions

Event 5 is expected to coincide with the $C_3$ plant crisis, leading to severe biodiversity loss. SS metazoans in low latitudes will be unable to survive due to extreme global surface temperatures averaging ~40°C annually. This high-temperature environment will also impact middle latitudes, increasing extinction rates beyond those observed in typical major mass extinctions (Fig. 1). The survival rates during this event (SRC) are estimated at 0.6 for insect families and 0.4 for terrestrial tetrapods. Due to the loss of $C_3$ plants, the recovery rate is expected to be moderate (0.3–0.8) because $C_4$ plants support the primally productivity.

By Event 7, oxygen depletion will begin driving a further decline in metazoan diversity. This decline will not be limited to mass extinction events alone but will also affect non-event periods. As a result, metazoan diversity will continue decreasing throughout climate phase B.

The $C_3$ plant crisis, Events 5–7, and a gradual oxygen decline starting from 0.5 Gyr will lead to significant biodiversity loss between 0.40 and 0.69 Gyr into the future. The number of insect families is projected to drop from 610 to 74–103, terrestrial tetrapod families from 315 to 37–47, and marine metazoan families from 950 to 189–358.





### 3.2.3 Metazoan demise

The final three mass extinction events (Events 8–10) are projected to occur at approximately 0.7, 0.8, and 0.9 Gyr during climate phase C. These extinctions, triggered by large-scale volcanism or meteorite impacts, will result in global surface temperatures reaching 44–46°C.

Events 8 and 9, along with the $C_4$ plant crisis and severe oxygen depletion between 0.69 and 0.88 Gyr, will drive the complete collapse of metazoan diversity. The number of insect families will decline from 74–103 to 0, terrestrial tetrapods from 37–47 to 0, and marine metazoans from 189–358 to 0–1 family. Any remaining metazoan family will disappear during Event 10, marking the complete extinction of all metazoans on Earth.

Event 8 is expected to lead to the extinction of all SS and UD metazoans due to global warming, oceanic anoxia-euxinia, the $C_3$ plant crisis, and atmospheric oxygen depletion.

### 3.2.4 Aftermath

During climate phase D, mass extinction events are projected to cause global warming, with temperatures reaching 51–54 °C—exceeding the survival threshold for metazoans, even in high-latitude underground and deep-sea environments. Simultaneously, atmospheric oxygen levels are expected to decline to less than 1% PAL. Although some highly specialized metazoans dependent on thermophilic bacteria may temporarily survive during background extinction periods, severe oxygen scarcity will ultimately render these adaptations unsustainable.

Between 1.25 and 1.5 Gyr from now (climate phase E), global average surface temperatures are projected to increase to 55–70°C during background periods and 65–70°C during extreme events. Even in high-latitude underground environments, temperatures are expected to remain elevated at 55–60°C, making survival increasingly unlikely. Under these extreme conditions, only thermophilic bacteria are expected to persist, thriving in an environment characterized by intense heat and critically low oxygen levels.

Temperatures will continue to rise, surpassing 100°C approximately 1.6 Gyr from now. This extreme heating will result in the complete evaporation of Earth's oceans, ultimately transforming the planet into a hot, arid landscape akin to present-day Venus.

## 4 Discussion

### 4.1 Validity of estimating of the future diversity of metazoans

Future surface temperature changes are influenced by three primary factors: (1) an increase in solar luminosity coupled with plant crises, (2) long-term (0.3 Gyr) hothouse-icehouse cycles, and (3) major mass extinction events. The reliability of these projections depends on four key considerations: (1) whether solar luminosity remains the dominant factor determining the





persistence or extinction of life, (2) whether long-term (0.3 Gyr) hothouse-icehouse cycles continue to occur, (3) whether
abrupt climate changes associated with major mass extinctions take place in the future, **and** (4) whether life can survive
beyond the current anthropogenic crisis, including $CO_2$-driven warming, pollution, and deforestation.

The first factor, solar luminosity increase, is modeled using a combination of terrestrial mantle convection models,
quasigrey atmosphere models, planetary albedo variations, effective solar flux estimates, surface temperature projections,
atmospheric water loss through photolysis, weathering models, and a simplified biosphere model. These models, developed
by Mello and Friaça (2019) and others, incorporate critical interactions between the atmosphere, oceans, mantle, chemical
weathering, and biosphere, ensuring a valid estimation of global surface temperatures.

The second factor, long-term hothouse-icehouse cycles, is derived from sea surface temperature (SST) data, based on
oxygen isotope records from fossil apatite and calcite, as well as evidence from glacial diamictite formations across different
latitudes in Earth's history. Over the past 1.0 Gyr, three full cycles of long-term 0.3 Gyr hothouse-icehouse transitions have
been identified, supporting the assumption that these cycles will persist in the future (Figs. 5 and 6). These cycles are
expected to continue because mantle temperatures will remain sufficiently high for large-scale mantle dynamics:
approximately 1350°C at present, 1250°C in 0.7 Gyr, and 1200°C in 1.0 Gyr (Mello and Friaça, 2019).

The third factor, abrupt climate changes linked to mass extinction events, is also based on oxygen isotope records from
fossil apatite and calcite, specifically at the five largest mass extinctions. Mantle temperature projections indicate that large
plume volcanism will still be possible in the future (Mello and Friaça, 2019). Additionally, asteroid impacts are expected to
occur regularly, as recorded in lunar and terrestrial impact histories (Glikson, 1999).

The fourth factor, the anthropogenic crisis, may trigger a minor mass extinction event that will peak in the late 21st
century (Kaiho, 2022b, 2023). Despite this, survival and recovery of metazoans remain possible if anthropogenic pollution,
deforestation, $CO_2$ emissions, and nuclear threats are mitigated. The final biodiversity outcome after the anthropogenic crisis
depends on human-driven environmental conservation efforts. This study assumes a 0.90 survival rate for terrestrial species,
0.95 for marine species, and an approximate 1.0 family survival rate for both terrestrial and marine metazoans. It also
considers two extreme scenarios: one where metazoans fully recover (RR = 1.0, Fig. 5) and another where no recovery
occurs (RR = 0.0, Fig. 6), leading to their final extinction.

Figures 5 and 6 cover all possible future diversity scenarios. In all cases, a decline in metazoan diversity occurs
between 0.4 and 0.7 Gyr, followed by their complete extinction at 0.7 Gyr into the future.

### 4.1.1  Future mass extinction events and anthropogenic crisis

The primary drivers of past major mass extinctions have been large-scale volcanism, such as Large Igneous Provinces
(LIPs), and asteroid impacts. Similar catastrophic events are expected to occur in the future due to ongoing geodynamic
evolution. Although mantle temperature is gradually decreasing, it is projected to remain high enough to sustain LIP
formation for at least the next 1.5 Gyr (see Methods).



Currently, human-induced environmental disturbances—including global warming, pollution, and deforestation—are accelerating extinction rates (Kaiho, 2023). Estimates suggest that terrestrial tetrapod extinction rates could reach approximately 10% between 2060 and 2080. However, beyond 2080, extinction rates are unlikely to increase further, as human population growth is expected to peak and then decline (Ritchie, 2023), potentially reducing pollution and

deforestation (Kaiho, 2022, 2023).

By 2100, mammalian species will face significant threats due to habitat destruction and climate change (Toussaint et al., 2021). Even though global warming will persist, driving continued species loss for the next few centuries (Kaiho, 2023), the overall impact on biodiversity may be moderate. Studies based on Paleocene-Eocene Thermal Maximum (PETM) data suggest that under temperature anomalies below 6.5°C, species loss remains under 5%, with family-level extinctions

approaching 0% (Kaiho, 2023).

However, despite these relatively low extinction percentages, significant declines in metazoan populations are still anticipated (Dirzo et al., 2014). Minor extinction events such as these are not expected to substantially alter the long-term diversity curve.

Extinction or survival rates in response to abrupt changes are crucial for certain species, such as *Homo sapiens*, and for

the final demise of each metazoan group. However, they are less relevant to long-term biodiversity trends. In contrast, recovery rates and survival rates for gradual changes play a significant role in shaping long-term diversity (underlined numbers in Table 2).

### 4.1.2 Thermal, CO₂ and oxygen constraints on metazoan survival

The absolute maximum temperature thresholds presented in this study apply only to a limited number of species. Many

terrestrial metazoans struggle to survive in environments exceeding 35°C, and it remains uncertain whether they can adapt to even higher temperatures (Sunday et al., 2011; Araújo et al., 2013). During past mass extinctions, both marine and terrestrial organisms responded to extreme warming by migrating away from high-temperature regions (Sun et al., 2012). This migration effect has already been incorporated into the mass extinction percentages derived from historical diversity records, which assume 100% post-extinction recovery rates (Table 2). The average extinction percentages from past events were used

to estimate future extinction probabilities (Table 2).

Declining atmospheric oxygen levels will further increase extinction risks, particularly for terrestrial organisms. As the ozone layer thins due to reduced oxygen concentrations, terrestrial metazoans will face greater exposure to ultraviolet radiation, which is more harmful than in marine environments. This pattern is already evident in historical diversity records, as indicated by the delayed appearance of terrestrial metazoans compared to marine metazoans (Fig. 3, Table 1).

Although future oxygen level projections contain some uncertainties, current models suggest a decline to 0.01 PAL by 1.0–1.1 Gyr (Figs. 5 and 6; Ozaki and Reinhard, 2021). This projection aligns with data showing a continuous downward trend in atmospheric oxygen levels from -0.1 Gyr to the present and extending into the future.





The combined effects of global warming, plant crises (resulting from $CO_2$ depletion due to increased weathering), and oxygen loss are expected to drive a substantial decline in metazoan diversity after 0.5 Gyr. This decline will not be limited to

mass extinction events but will also persist during extended non-event periods, ultimately leading to the complete extinction of metazoans at 0.7 Gyr into the future.

### 4.1.3 Future diversity of terrestrial invertebrates

Due to the lack of comprehensive historical diversity data for terrestrial invertebrates (excluding insects), Figures 5 and 6 illustrates only the diversity trends of tetrapods and insects in terrestrial environments. However, it is expected that other

terrestrial invertebrate groups will follow similar diversity trends, given their comparable upper thermal tolerance limits. These organisms typically reside in surface environments or within a few centimeters of soil, where temperature fluctuations are similar to those experienced by insects and small vertebrates. As a result, terrestrial invertebrate diversity trends are likely to mirror those of tetrapods and insects.

### 4.2 Future trends in metazoan diversity and extinction

Future metazoan diversity will be shaped by a combination of mass extinctions, plant crises, and atmospheric oxygen depletion. During climate phase A, global cooling caused by $SO_2$ or soot emissions (~10°C reduction in global surface average temperature) and warming due to $CO_2$ emissions (~30°C increase in global surface average temperature) will contribute to mass extinctions. In climate phase B, $CO_2$-driven warming will further raise global surface temperatures to 35–40°C, leading to additional extinctions, while temporary global cooling (15–20°C) may briefly mitigate these effects.

The impact of short-term $SO_2$, soot, and $CO_2$ release events on long-term warming trends is minimal. While these events can cause temperature fluctuations of ~10°C, their effects last less than a million years. Over longer geological timescales (100 million years or more), such variations become negligible due to natural recovery processes, including sulfur deposition and $CO_2$ absorption. As a result, the combined effects of long-term warming trends, cyclical climate fluctuations, and catastrophic events—as described in Equations 2 and 3 in the Methods—accurately represent global temperature trends

across different geological periods (Fig. 1).

The upper thermal tolerance limit for metazoans is highly conserved across lineages due to protein denaturation at high temperatures (Fig. 2). This biological constraint allows for reliable estimates of metazoan decline and extinction timelines, based on global warming projections.

The $C_3$ plant crisis is predicted to occur at $0.4 \pm 0.2$ Gyr, resulting in the widespread loss of trees and associated

metazoans (Figs. 5 and 6; Mello and Friaça, 2019; Ozaki and Reinhard, 2021). This crisis may drive the evolution and expansion of $C_4$ plants, which are more efficient at concentrating $CO_2$ for photosynthesis. The transition is expected to involve a major ecological shift from grass-dominated ecosystems to tree-based ecosystems, enabling partial metazoan





recovery. Recovery rates, depending on the likelihood of this evolutionary shift, are estimated to range between 0.3 and 0.8 (Figs. 5 and 6).

In marine environments, both $C_3$ and $C_4$ photosynthesis systems are expected to coexist, supporting biodiversity among marine metazoans (Reinfelder et al., 2000, 2004). Similar reduction and recovery trends observed in terrestrial tetrapods are tentatively extended to marine metazoans based on the two outlined scenarios. However, despite potential evolutionary adaptations, the $C_3$ plant crisis is still expected to cause a significant decline in overall metazoan diversity (Figs. 5 and 6).

Building on these considerations, Figures 5 and 6 illustrate trends in metazoan diversity over the past 1 Gyr, extending
1.5 Gyr into the future. While uncertainties exist, all models indicate a steady decline in metazoan diversity between 0.4 and 0.7 Gyr, followed by total metazoan extinction at 0.7 Gyr.

## 4.3 Survival strategies and the fate of intelligent life

Humankind—including Homo sapiens and potential future species or genera derived from it—may experience up to seven major extinction events (Events 1–7, Fig. 1). The most likely causes of these events include large-scale volcanic eruptions
and meteorite impacts (Bond and Grasby, 2017; Kaiho, 2024). These events could lead to reduced sunlight, climate disruption, and severe food shortages, with far-reaching consequences exceeding those of the current Anthropogenic crisis (Kaiho, 2022, 2023).

The final decline and extinction of metazoans is expected to occur at 0.7 Gyr from now, approximately 0.9 Gyr before Earth becomes completely dry and uninhabitable for most life forms (Fig. 6). Before reaching this point, highly intelligent
species may attempt to adapt to extreme environmental conditions through various strategies, including:

- Reducing human pressure on the biosphere to slow biodiversity loss
- Blocking sunlight using aerosols (Kaiho and Oshima, 2017)
- Seeking refuge in subterranean environments
- Altering Earth's orbit to mitigate extreme environmental conditions
- Migrating to extraterrestrial habitats, such as Mars or other habitable planets

At a global surface temperature of 45°C, superterranean and surface-water metazoans will be unable to survive. This threshold will be exceeded during a major mass extinction event around ~0.7 Gyr and in non-event periods by 1.0 Gyr (Fig. 1). However, at low latitudes on Mars, surface temperatures may remain habitable for metazoans around 0.7 Gyr.

The total lifespan of metazoans on Earth is estimated to be 1.4 billion years, accounting for approximately 12% of
Earth's 12-billion-year history (Fig. 7). Humanity currently exists at the midpoint (~50%) of this metazoan lifespan (Fig. 6). Preserving Earth's environmental stability is crucial for prolonging metazoan survival in terrestrial and marine ecosystems.

## 4.4 Incomplete recoveries in metazoan diversity

Four key factors will contribute to the incomplete recovery of metazoan diversity:





1. The Anthropogenic Crisis: The first factor causing incomplete recovery of metazoan diversity may occur due to human-driven environmental pressures, which will persist until the human impact on the biosphere declines.

2. The $C_3$ Plant Crisis: The second factor will occur due to primary productivity loss at around 0.4 Gyr, driven by a gradual decrease in $CO_2$. This will prevent full biodiversity recovery, as low primary productivity levels will limit species expansion.

3. Gradual Oxygen Depletion (0.6–1.1 Gyr): The third factor will occur due to continuous oxygen depletion, restricting metazoan recovery.

4. Rising Global Temperatures at Events 8–10 and Non-Event Periods (0.7–0.9 Gyr): The fourth factor will be driven by elevated global surface temperatures (44–46°C), leading to the extinction of metazoans. Even outside major extinction events, non-event period temperatures (36–38°C)—comparable to the most severe mass extinction events (e.g., end-Permian)—will persist, preventing the recovery of metazoans. Rising Global Temperatures at Events 8–10 and Non-Event Periods (0.7–0.9 Gyr): The fourth factor will be driven by elevated global surface temperatures (44–46°C), leading to the extinction of metazoans. Even outside major extinction events, non-event period temperatures (36–38°C)—comparable to the most severe mass extinction events (e.g., end-Permian)—will persist, preventing the recovery of metazoans. These high temperatures also cause deep-water anoxia-euxinia, leading to the deep-water metazoan extinction (See Methods and Table 2 caption).

The combined effects of factors 2–4 will contribute to the decline of metazoan diversity during Events 5–7 and ultimately lead to the extinction of metazoans during Events 8–10 (Fig. 5). Factor 1 may cause a reduction in diversity during the anthropogenic crisis, which will persist until human-induced environmental impacts subside. In the worst-case scenario—including a full-scale nuclear war—this crisis could extend until the complete demise of metazoans (Fig. 6).

## 5 Conclusions

This study incorporates metazoan temperature tolerance data along with other environmental factors to project future biodiversity trends and estimate extinction timelines.

Our findings indicate that metazoans will face complete extinction approximately 700 million years from now—300–400 million years earlier than previous estimates. Over the next 400 million years, biodiversity will repeatedly decline and recover following mass extinctions. However, rising solar luminosity and long-term climate shifts will progressively elevate peak global temperatures every 300 million years beyond 400 million years from now. Concurrent declines in atmospheric oxygen and carbon dioxide will impose increasing stress on metazoans, even outside extinction events, due to rising surface temperatures and the decline of $C_3$ plants. All metazoans are expected to disappear around 700 million years from now, likely due to volcanic activity or an impact event.

This scenario suggests that metazoan extinction will be complete within 0.7 Gyr. To mitigate biodiversity loss, advanced species may need to implement strategies such as solar radiation management (e.g., aerosol shielding), adaptation





to subterranean environments, altering Earth's orbit, or extraterrestrial colonization (e.g., Mars) within this timeframe.
Notably, this study is the first to reveal that humanity exists at the midpoint of metazoan lifespan, emphasizing the urgency
of proactive measures to ensure the long-term survival of complex life.

**Figure 1.** Global average surface temperature and atmospheric oxygen levels over the past and future 2.5 Gyr. Global average surface
temperature and atmospheric oxygen levels over the past 1.0 billion years (Gyr) (-1.0 Gyr) and projected for the next 1.5 Gyr (1.5 Gyr),
alongside metazoan extinction thresholds, climate phases, and metazoan phases. The black dashed line, adapted from Mello and Friaça
(2019), represents historical trends, while the orange curve illustrates long-term cyclical icehouse and greenhouse phases, with historical

data adapted from Scotese et al. (2021) and estimates for the period between -1.0 and -0.6 Gyr based on Vérard (2024) (Table A1). Future





projections for both lines begin at the green star, which marks the current average surface temperature of 14°C. Light blue and red dots represent average surface temperatures during mass extinction events, with light blue indicating cooling phases and red showing subsequent warming phases (Table A2). Green open diamond symbols denote temperatures leading up to mass extinctions. Thick vertical error bars indicate the standard deviation of temperature anomalies during major mass extinction events, while thin vertical error bars

represent uncertainty in the orange curve over these events (Table A2). Thin error bars are added to thick error bars. The yellow-orange shaded regions represent the upper temperature limits for superterranean and surface-water (SS) metazoan extinction at 30°, 60°, and 90° latitudes, while the gray shaded areas denote the upper temperature limits for subterranean (underground) (U) metazoan extinction at 60° and 90° latitudes. For deep-sea (D) metazoans, the thresholds are represented by the purple horizontal line (Fig. 2). The labels "eP" and "KPg" mark the end-Permian and Cretaceous-Paleogene boundary mass extinction events, respectively, while numbers 1, 5, 8, 11, and 14

correspond to event numbers. The predicted timing of surface oxygen depletion is based on Ozaki and Reinhard (2021). The gray gradient in the atmospheric oxygen level graph represents its impact on metazoans (see equations 16–21 in Section 2.6), with darker shades indicating a greater limitation on metazoan diversity. PAL refers to the Present Atmospheric Level. Past atmospheric oxygen data are sourced from Krause et al. (2018) and Sperling et al. (2015).





**Figure 2.** Surface temperatures across latitudes for the extinction of metazoans. (**a**, **c**) GATES: Global Average surface Temperatures required for the Extinction of Superterranean and Surface-water (SS) metazoans. Thick oblique lines represent Local Daytime Maximum average Temperatures during the warmest month (LDMT). Dashed oblique lines indicate Local Monthly Maximum Temperatures (LMMT), while thin oblique lines correspond to Local Average Temperatures (LAT). LT: Local Temperature. Latitude values 0°, 30°, 60°, and 90° are labeled as 00, 30, 60, and 90, respectively. The intersection of the vertical black line at 37° with an LAT oblique line

represents the global average surface temperature (TES00 to TES90). The yellow-shaded region highlights the surface temperature range that can cause metazoan extinction, which is common to both terrestrial and marine metazoans. Closed circle markers denote the maximum extinction temperatures (46 °C) for SS metazoans at 0°, 30°, 60°, and 90°. Open circle markers indicate the corresponding global average surface temperatures for these latitudes, color-matched to the closed circles. See Methods. (**b**) GATEU: Global Average surface Temperature required for the Extinction of subterranean (Underground) metazoans. (**d**) GATED: Global Average surface

Temperature required for the extinction of Deep-water metazoans. Open circle markers represent the maximum global average surface temperatures that lead to subterranean metazoan extinction, while the purple horizontal line denotes the extinction threshold for deep-sea metazoans (TED). See Methods section for details.

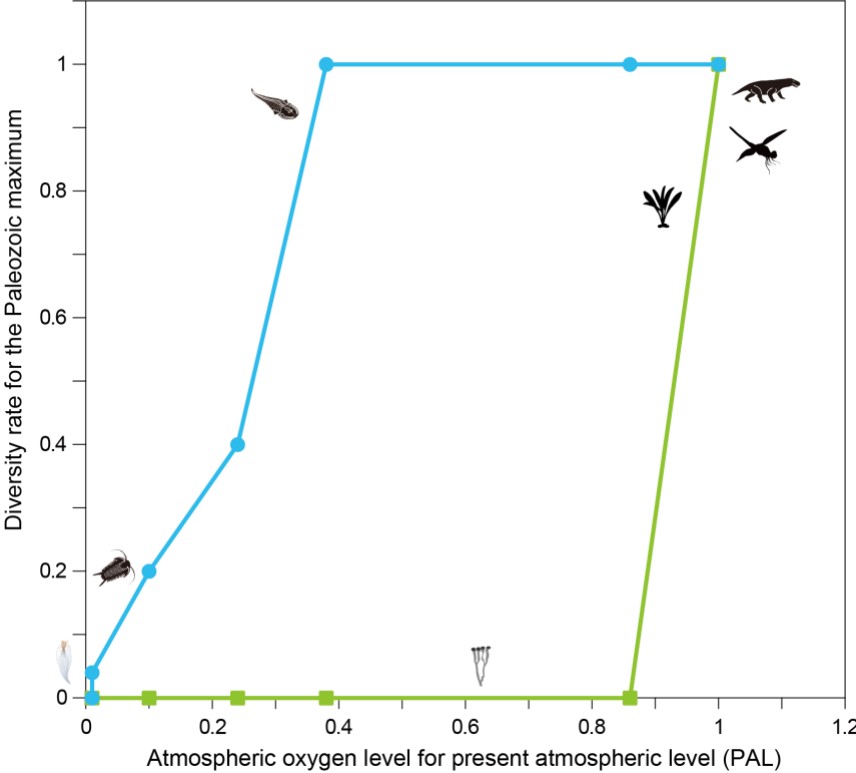

**Figure 3.** Relationship between atmospheric oxygen levels and past metazoan diversity. This figure illustrates the relationship between

atmospheric oxygen levels and the diversity of marine metazoans, terrestrial tetrapods, and insects. Silhouettes of terrestrial plants indicate their approximate diversity and corresponding oxygen levels. Blue lines with circular markers represent marine metazoans, while pale green lines with square markers represent terrestrial tetrapods and insects. These data are derived from records spanning the





Neoproterozoic and Paleozoic oxygen increase periods. Atmospheric oxygen level data are sourced from Krause et al. (2018) and Sperling et al. (2015), while diversity data are based on Erwin et al. (1987), Labandeira and Sepkoski (1993), Engel and Grimaldi (2004), and

Benton (2010).

**Figure 4.** Flowchart of methods for estimating metazoan diversity in the past and future to determine the lifespan of metazoans on Earth.

This figure presents a flowchart outlining the methods used to generate Figures 5 and 6. Different colors represent key influencing factors: green indicates diversity, red represents temperature, blue corresponds to atmospheric oxygen levels, and yellowish-green denotes local temperature conditions for metazoans.




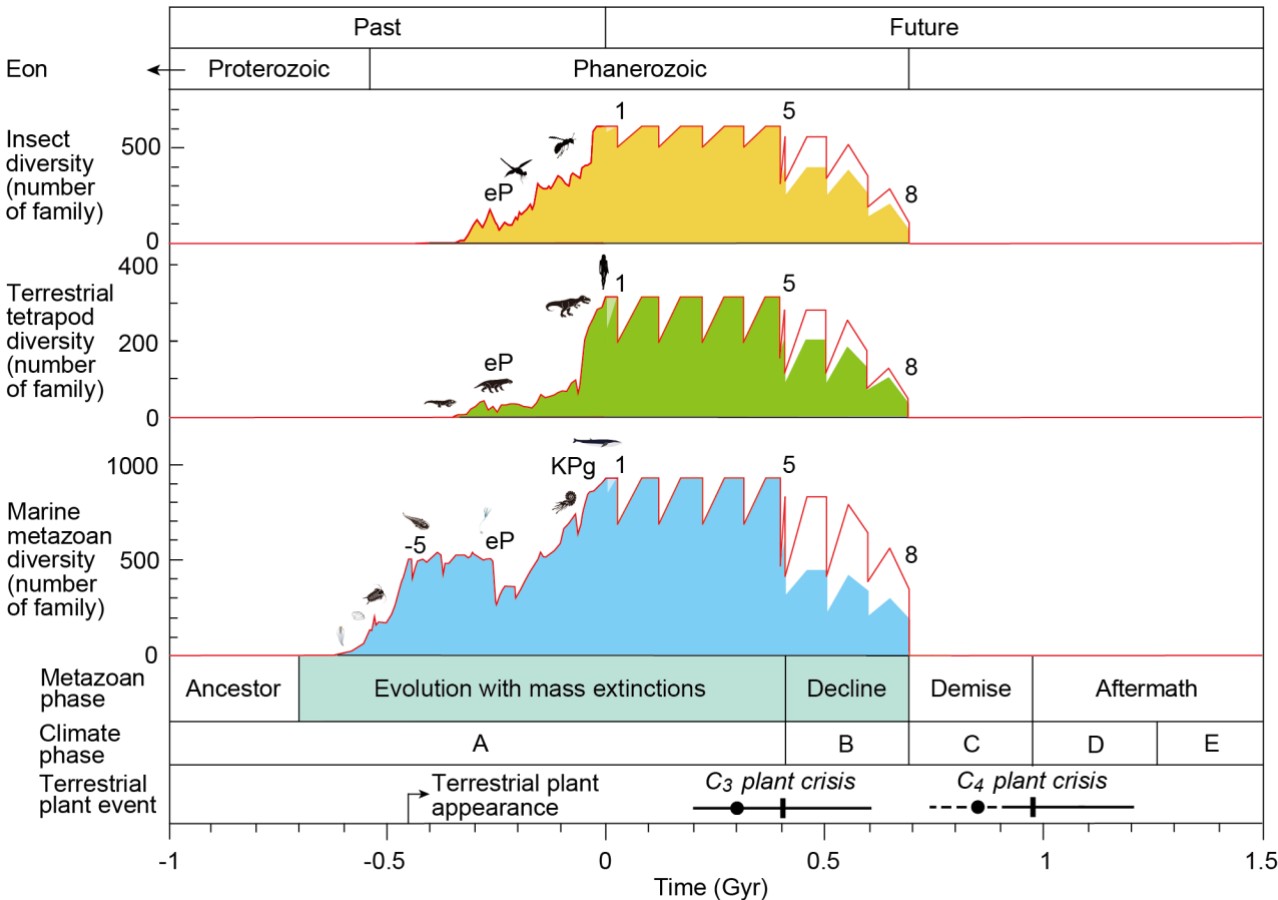

**Figure 5.** Changes of diversity of insects, tetrapods, and marine metazoan families. This figure illustrates the diversity trends of insects,
terrestrial tetrapods, and marine metazoan families over the past 1 billion years (Gyr) and projections for the next 1.5 Gyr, alongside key
terrestrial plant events. Diversity data through geological time are derived from fossil records documented in previous studies (Erwin et al.,
1987; Labandeira and Sepkoski, 1993; Engel and Grimaldi, 2004; Benton, 2010; see Tables A3 and A4). Future diversity projections are
based on extinction and recovery percentages outlined in Table 2 (see Table A5). Colored and white graphs illustrate the minimum and
maximum recovery scenarios following mass extinction events and the $C_3$ plant crisis (Table 2). Pale-colored sections between events 0
and 1 represent a scenario without full-scale nuclear war under a medium $CO_2$ emission scenario, whereas dark-colored sections between
events 0 and 1 indicate a scenario where a full-scale nuclear war occurred under the same $CO_2$ emission conditions (Kaiho, 2023). Each
metazoan group contains three independent graphs, rather than an integrated representation. Case 1 in Table A5 corresponds to both pale
and dark-colored sections, Case 2 corresponds to dark-colored sections, and Case 3 represents the maximum diversity curve shown by red
lines. Key transitions, including the $C_3$ and $C_4$ plant crises, are modeled using data from Mello and Friaça (2019) (depicted by dots and
dashed lines) and Ozaki and Reinhard (2021) (depicted by solid lines). Detailed methodologies for calculating future metazoan diversity
projections are available in Methods section. Abbreviations: eP refers to the end-Permian extinction, KPg denotes the Cretaceous-
Paleogene boundary, and -5, 1, 5, 8, and 9 represent event numbers.



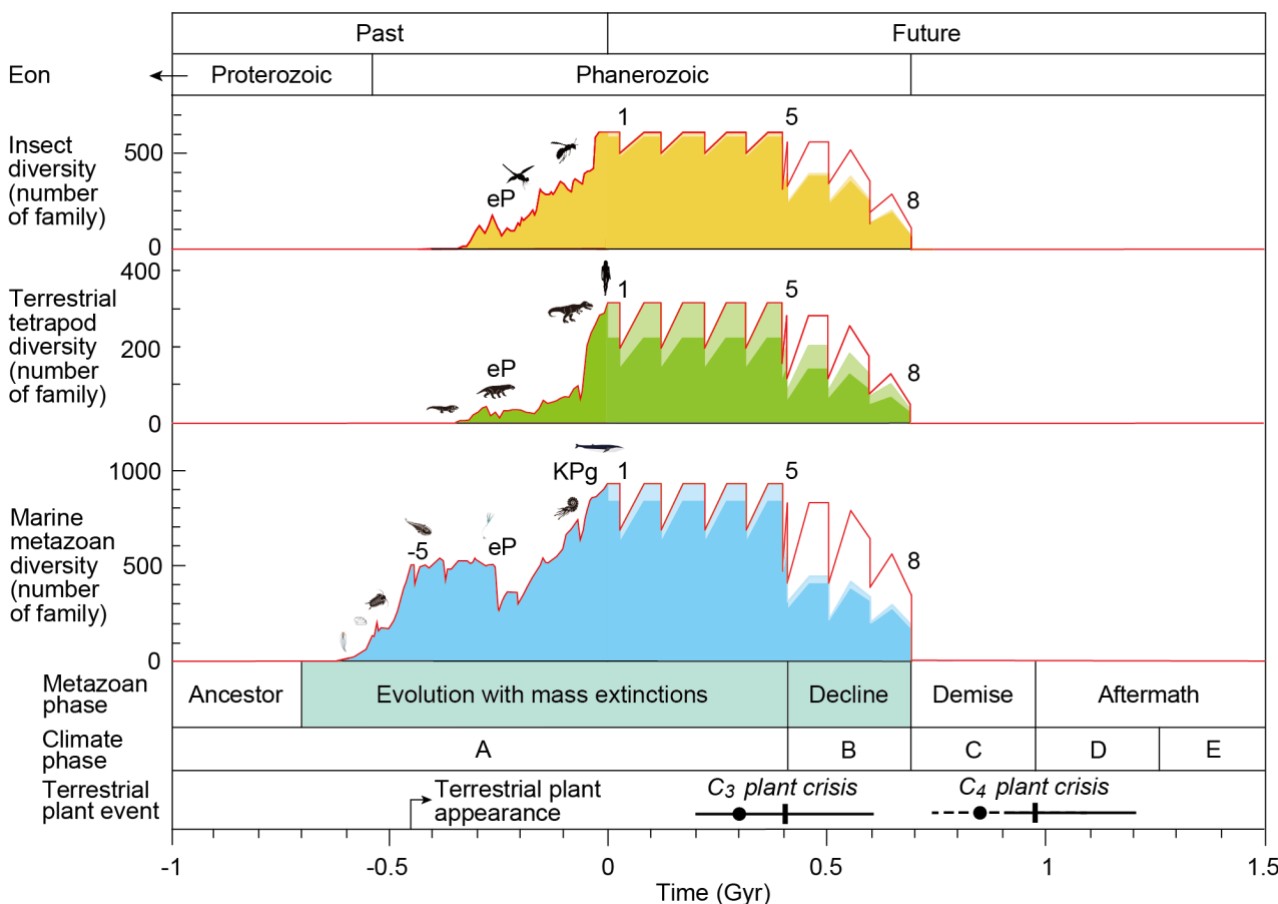

**Figure 6.** Diversity of insects, tetrapods, and marine metazoan families over the past and future 2.5 Gyr, under prolonged impact of the maximum anthropogenic crisis until metazoan demise. Pale-colored graphs depict a scenario without full-scale nuclear war under the medium $CO_2$ emission scenario, whereas colored graphs represent a scenario where full-scale nuclear war occurred under the medium $CO_2$ emission scenario and metazoan diversity remains suppressed by the maximum anthropogenic crisis until their extinction. Each metazoan group contains three independent graphs rather than an integrated representation. Case 4 in Table A5 corresponds to the dark-colored sections.





**Figure 7.** Lifespan of Prokaryotes, Eukaryotes, and Metazoans in Earth's History Alongside Metazoan Diversity. This figure presents the estimated lifespans of prokaryotes, eukaryotes, and metazoans in Earth's history, alongside metazoan diversity trends. The timeline of Earth's history, including the presence of oceans and the transition to a dry, uninhabitable state, is based on data from Mello and Friaça (2019) and referenced studies in the introduction section. The estimated lifespans of prokaryotes, eukaryotes, and metazoans are derived from Tashiro et al. (2017), Albani et al. (2010), and Yuan et al. (2011), respectively. Metazoan diversity graphs are sourced from Figure 5. The evolution of our Sun is based on Christensen-Dalsgaard (2021). The cooling of Earth's mantle and the eventual cessation of volcanism are based on research by Korenaga (2008a, b), Condie et al. (2016), and Palin et al. (2020).




**Table 1.** Atmospheric oxygen level and metazoan diversity rates controlled by oxygen during -1 to 1.5 Gyr

| Age | Time (Gyr) | Atmospheric Oxygen level for PAL | Surface-water metazoan diversity rate for Paleozoic maxima | Terrestrial tetrapod diversity rate for Paleozoic maxima |
|---|---|---|---|---|
| Future | 1.5 | 0.00 | 0 | 0 |
| | 1.0–1.1 | 0.01 | 0 | 0 |
| | 0.8 | 0.30 | 0.4 | 0 |
| | 0.5 | 0.50 | 1 | 1 |
| Mesozoic | -0.1 | 1.00 | 1 | 1 |
| Paleozoic | -0.27 | 1.00 | 1 | 1 |
| | -0.42 | 0.86 | 1 | 0 |
| | -0.44 | 0.38 | 1 | 0 |
| | -0.52 | 0.24 | 0.4 | 0 |
| Neoproterozoic | -0.55 | 0.10 | 0.2 | 0 |
| | -0.6 | 0.01 | 0.04 | 0 |
| | -0.7 | 0.01 | 0 | 0 |
| | -1 | 0.01 | 0 | 0 |

Future atmospheric oxygen levels for PAL are from Ozaki and Reinhard (2021) and the past atmospheric oxygen levels for PAL are from Krause et al. (2018) and Sperling et al. (2015). PAL: Present Atmospheric Level. Future diversity rates are estimated from the relationship between atmospheric oxygen levels and diversity rates in the past. *The superterranean metazoan diversity rate in future are estimated from the terrestrial plant diversity rate in the Silurian to Carboniferous because of delay of evolution of terrestrial tetrapods for terrestrial plants (Cascales-Miñana, 2016). To estimate the relationship between atmospheric oxygen level and metazoan diversity, the diversity of Paleozoic maxima to that of the present is tentatively set at 1.





**Table 2.** Extinction and recovery model for future projections, showing diversity change rates in family level

| Climate phase | Future event | Age (Gyr) | Survival Rate by Climate (SRC) | | | Recovery Rate (RR) for All | Survival Rate (SR) by gradual warming | | Survival Rate by O₂ (SRO) | |
|---|---|---|---|---|---|---|---|---|---|---|
| | | | Insect | Tetrapods | Marine animals | | Terrestrial | Marine | Terrestrial | Marine |
| E | 16 | 1.45 | 0.0 | 0.0 | 0.0 | 0.0 | 0.0 | 0.0 | 0.0 | 0.0 |
| E | 15 | 1.35 | 0.0 | 0.0 | 0.0 | 0.0 | 0.0 | 0.0 | 0.0 | 0.0 |
| E | 14 | 1.26 | 0.0 | 0.0 | 0.0 | 0.0 | 0.0 | 0.0 | 0.0 | 0.0 |
| D | 13 | 1.16 | 0.001 | 0.001 | 0.0 | 0.01–0.1 | * | ** | 0–0.001 | 0–0.01 |
| D | Oxygen | 1.07 | 0.001 | 0.001 | 0.001 | 0.01–0.1 | – | – | 0–0.001 | 0–0.01 |
| D | 12 | 1.07 | 0.001 | 0.001 | 0.0 | 0.01–0.1 | * | ** | 0–0.001 | 0–0.01 |
| D | 11 | 0.97 | 0.001 | 0.001 | 0.0 | 0.01–0.1 | * | ** | 0.001–0.01 | 0.09 |
| C | C₄ plant | 0.97 | 0.001 | 0.001 | 0.001 | 0.01–0.1 | – | – | 0.001–0.01 | 0.09 |
| C | 10 | 0.88 | 0.003–0.015 | 0.003–0.015 | 0.0 | 0.01–0.3 | 0.68 | 0.84 | 0.01–0.05 | 0.26 |
| C | 9 | 0.78 | 0.003–0.015 | 0.003–0.015 | 0.0 | 0.01–0.3 | 0.68 | 0.84 | 0.07 | 0.44 |
| C | 8 | 0.69 | 0.003–0.015 | 0.003–0.015 | 0.0 | 0.01–0.3 | 0.85 | 0.95 | 0.38 | 0.62 |
| B | 7 | 0.6 | 0.81 | 0.63 | 0.74 | 1 | 1 | 1 | 0.67 | 0.8 |
| B | 6 | 0.5 | 0.6 | 0.4 | 0.5 | 1 | 1 | 1 | 1 | 1 |
| B | 5 | 0.41 | 0.6 | 0.4 | 0.5 | 1 | 1 | 1 | 1 | 1 |
| A | C₃ plant | 0.4 | 0.5 | 0.5 | 0.5 | 0.3–0.8 | 1 | 1 | 1 | 1 |
| A | 4 | 0.31 | 0.81 | 0.63 | 0.74 | 1 | 1 | 1 | 1 | 1 |
| A | 3 | 0.22 | 0.81 | 0.63 | 0.74 | 1 | 1 | 1 | 1 | 1 |
| A | 2 | 0.12 | 0.81 | 0.63 | 0.74 | 1 | 1 | 1 | 1 | 1 |
| A | 1 | 0.03 | 0.81 | 0.63 | 0.74 | 1 | 1 | 1 | 1 | 1 |
| A | 0 | 0.00 | 0–0.95 | 0–0.7 | 0–0.9 | **0**-1 | 1 | 1 | 1 | 1 |

This table presents the percentages of family survival rate, recovery rate, and diversity rate under oxygen depletion. Survival rates of 0.63 and 0.81 are derived from Tables A3 and A4, while the other survival rates are sourced from Sections 2.3 and 2.6. Underlined values indicate factors contributing to the long-term decline of metazoan diversity. Event 0 represents the anthropogenic crisis, where a bold **0** signifies the continuation of zero recovery, ultimately leading to the complete extinction of metazoans (case 4 in Table A5). In Events 8–10, following the demise of SS metazoans, the minimum SRC for subterranean (U) metazoans is $0.05 \times 0.66 \times 0.1 = 0.003$, while the maximum SRC is $0.05 \times 0.66 \times 0.5 = 0.015$ (Equation 15). For deep-water (D) metazoans, the minimum SRC is $0.33 \times 0.76 \times 0.1 = 0.025$, whereas the maximum SRC is $0.33 \times 0.76 \times 0.5 = 0.125$ (Equation 16). The ranges are 0.003–0.015 and 0.025–0.125, respectively. However, SAR$_D$ is set equal to SAR$_S$ for Events 5–16 and for non-events after Event 8, as these are characterized by temperatures comparable to or exceeding those associated with end-Permian oceanic anoxia-euxinia (see Methods). Since SAR$_S$ values for Events 8–16 are 0.0, SRC$_M$ is also 0.0 for these events. *: 0.003–0.015. **: 0.025–0.125.

**Appendix A**





**Table A1.** Earth's average surface temperature from 0.7 billion years (Gyr) before the present to 1.5 Gyr into the future

| Age from present (Gyr) | Long-term Temperature (°C) | Long-term T + climate cycle (°C) | Age from present (Gyr) | Long-term Temperature (°C) | Long-term T + climate cycle (°C) |
|---|---|---|---|---|---|
| -0.7 | 0 | 0 | 0.45 | 21.5 | 29.5 |
| -0.65 | 0 | 0 | 0.5 | 22.7 | 30.7 |
| -0.6 | 14 | 14 | 0.55 | 23.8 | 23.8 |
| -0.55 | 20 | 20 | 0.6 | 25 | 25 |
| -0.5 | 24 | 24 | 0.65 | 26.2 | 34.2 |
| -0.45 | 18 | 18 | 0.7 | 27.6 | 35.6 |
| -0.4 | 20 | 20 | 0.75 | 28.9 | 36.9 |
| -0.35 | 19 | 19 | 0.8 | 30.5 | 38.5 |
| -0.3 | 14 | 14 | 0.85 | 31.9 | 31.9 |
| -0.25 | 22 | 22 | 0.9 | 33.6 | 41.6 |
| -0.2 | 22 | 22 | 0.95 | 35.1 | 43.1 |
| -0.15 | 21 | 21 | 1 | 36.8 | 44.8 |
| -0.1 | 22 | 22 | 1.05 | 38.5 | 46.5 |
| -0.05 | 22 | 22 | 1.1 | 40.5 | 48.5 |
| 0 | 14 | 14 | 1.15 | 42.6 | 42.6 |
| 0.05 | 14.4 | 22.4 | 1.2 | 45 | 53 |
| 0.1 | 15 | 23 | 1.25 | 47.6 | 55.6 |
| 0.15 | 15.8 | 23.8 | 1.3 | 51 | 59 |
| 0.2 | 16.7 | 24.7 | 1.35 | 54.8 | 62.8 |
| 0.25 | 17.5 | 17.5 | 1.4 | 59 | 59 |
| 0.3 | 18.4 | 18.4 | 1.45 | 64 | 64 |
| 0.35 | 19.4 | 27.4 | 1.5 | 71 | 79 |
| 0.4 | 20.5 | 28.5 | | | |

Long-term temperature: Data from Mello and Friaça (2020) corresponding to the black dushed curve in Figure 1. Climate cycle: Average temperature anomaly (+8°C in greenhouse periods and 0°C in icehouse periods) of the long-term climate cycle observed in the Phanerozoic (Scotese et al., 2021). 8°C and 0°C were added to the long-term temperature in greenhouse and icehouse periods, respectively.






**Table A2.** Earth's average surface temperature of long-term trend, long-term cycle, and short-term event with temperature anomaly and $CO_2$ and $SO_2$ decreasing rate due to the decrease in mantle potential temperature during major mass extinction event from 0.7 billion years (Gyr) before the present to 1.5 Gyr into the future

| Event | Event Age | Long-term trend | | Long-term cycle (> 1 m.y.) | Short-term event (< 0.1 m.y.) | | | | | | | |
| | | Mantle potential Temp. | Long-term temperature | Long-term temperature + climate cycle | Cooling anomaly considering mantle temp | Cooling Temp. SD | Aftermath warming considering mantle temp | Aftermath warming SD | $SO_2$ Rate | $CO_2$ Rate | Temp. during warming event |
| | (Gyr) | (°C) | (°C) | (°C) | (°C) | (°C) | (°C) | (°C) | | | |
|---|---|---|---|---|---|---|---|---|---|---|---|
| 16 | 1.446 | 1184 | 64 | 64 | -7.47 | 2.21 | 6.46 | 1.65 | 0.76 | 0.65 | 70 |
| 15 | 1.352 | 1192 | 55 | 62 | -7.56 | 2.24 | 6.52 | 1.67 | 0.77 | 0.66 | 69 |
| 14 | 1.257 | 1199 | 48 | 56 | -7.66 | 2.26 | 6.57 | 1.68 | 0.79 | 0.66 | 63 |
| 13 | 1.163 | 1207 | 43 | 45 | -7.75 | 2.29 | 6.62 | 1.69 | 0.81 | 0.67 | 52 |
| 12 | 1.068 | 1215 | 39 | 47 | -7.84 | 2.32 | 6.93 | 1.77 | 0.82 | 0.70 | 54 |
| 11 | 0.974 | 1223 | 36 | 44 | -7.94 | 2.35 | 7.27 | 1.86 | 0.84 | 0.73 | 51 |
| 10 | 0.879 | 1232 | 33 | 38 | -8.03 | 2.37 | 7.53 | 1.92 | 0.85 | 0.76 | 46 |
| 9 | 0.785 | 1242 | 30 | 38 | -8.12 | 2.40 | 7.78 | 1.99 | 0.87 | 0.79 | 46 |
| 8 | 0.690 | 1251 | 27 | 35 | -8.22 | 2.43 | 8.04 | 2.05 | 0.89 | 0.81 | 43 |
| 7 | 0.596 | 1260 | 25 | 26 | -8.31 | 2.46 | 8.29 | 2.12 | 0.90 | 0.84 | 34 |
| 6 | 0.501 | 1270 | 23 | 30 | -8.40 | 2.48 | 8.55 | 2.18 | 0.92 | 0.86 | 39 |
| 5 | 0.407 | 1281 | 21 | 29 | -8.50 | 2.51 | 8.80 | 2.25 | 0.93 | 0.89 | 38 |
| 4 | 0.312 | 1293 | 18 | 22 | -8.59 | 2.54 | 9.06 | 2.31 | 0.95 | 0.91 | 31 |
| 3 | 0.218 | 1304 | 17 | 22 | -8.68 | 2.57 | 9.31 | 2.38 | 0.96 | 0.94 | 31 |
| 2 | 0.123 | 1315 | 15 | 23 | -8.78 | 2.59 | 9.57 | 2.45 | 0.98 | 0.97 | 33 |
| 1 | 0.029 | 1327 | 14 | 20 | -8.87 | 2.62 | 9.82 | 2.51 | 1.00 | 0.99 | 30 |
| | 0.000 | 1330 | 14 | 14 | – | – | – | – | – | – | – |
| -1 | -0.066 | 1335 | 13 | 22 | -12 | 0.00 | 7 | 0.00 | 1.00 | 1.00 | 29 |
| -2 | -0.201 | 1346 | 12 | 22 | -8 | 0.00 | 7 | 0.00 | 1.00 | 1.00 | 29 |
| -3 | -0.252 | 1350 | 12 | 22 | 0 | 0.00 | 14 | 0.00 | 1.00 | 1.00 | 36 |
| -4 | -0.372 | 1370 | 10 | 19 | -8 | 0.00 | 11 | 0.00 | 1.00 | 1.00 | 30 |
| -5 | -0.444 | 1380 | 10 | 18 | -10 | 0.00 | 10 | 0.00 | 1.00 | 1.00 | 28 |

Light blue and red dots in Figure 1 are sourced from this Table. Gyr: giga (billion) years. temp.: temperature. Mantle potential temperature and long-term temperature: Data from Mello and Friaça (2020) corresponding to the black dushed curve in Figure 1. See Methods for calculations of the data in this table.





**Table A3.** Number of tetrapod families and extinction percentages over geological time based on Benton (2010)

| Event | Age (Myr) | Mammals | Birds | Reptiles | Amphibians | Total | Extinction% |
|---|---|---|---|---|---|---|---|
| | 0 | 125 | 130 | 40 | 20 | 315 | |
| | -10 | 150 | 80 | 40 | 20 | 290 | |
| | -20 | 150 | 75 | 40 | 20 | 285 | |
| | -30 | 130 | 70 | 40 | 20 | 260 | |
| | -40 | 130 | 50 | 40 | 17 | 237 | |
| | -50 | 110 | 35 | 40 | 18 | 203 | |
| | -60 | 35 | 5 | 30 | 12 | 82 | |
| K-Pg | -65 | 14 | 5 | 30 | 12 | 61 | 38 |
| | -70 | 20 | 8 | 60 | 12 | 98 | |
| | -80 | 23 | 6 | 50 | 10 | 89 | |
| | -90 | 14 | 3 | 40 | 10 | 67 | |
| | -100 | 12 | 2 | 45 | 10 | 69 | |
| | -110 | 13 | 2 | 40 | 10 | 65 | |
| | -120 | 11 | 2 | 37 | 8 | 58 | |
| | -130 | 11 | 2 | 33 | 8 | 54 | |
| | -140 | 11 | 2 | 31 | 7 | 51 | |
| | -150 | 10 | 3 | 40 | 6 | 59 | |
| | -160 | 7 | 0 | 25 | 5 | 37 | |
| | -170 | 6 | 0 | 15 | 3 | 24 | |
| | -180 | 5 | 0 | 18 | 3 | 26 | |
| | -190 | 6 | 0 | 18 | 3 | 27 | 21 |
| end-T | -200 | 5 | 0 | 23 | 5 | 33 | |
| | -210 | 4 | 0 | 23 | 7 | 34 | |
| | -220 | 0 | 0 | 25 | 9 | 34 | |
| | -230 | 0 | 0 | 23 | 8 | 31 | |
| | -240 | 0 | 0 | 24 | 8 | 32 | |
| P-T | -250 | 0 | 0 | 10 | 3 | 13 | 54 |
| | -260 | 0 | 0 | 20 | 8 | 28 | |
| | -270 | 0 | 0 | 9 | 9 | 18 | |
| | -280 | 0 | 0 | 12 | 30 | 42 | |
| | -290 | 0 | 0 | 8 | 30 | 38 | |
| | -300 | 0 | 0 | 5 | 23 | 28 | |
| | -310 | 0 | 0 | 2 | 18 | 20 | |
| | -320 | 0 | 0 | 0 | 7 | 7 | |
| | -330 | 0 | 0 | 0 | 6 | 6 | |
| | -340 | 0 | 0 | 0 | 5 | 5 | |
| | -350 | 0 | 0 | 0 | 2 | 2 | |
| | | | | | | Average | 37 |

"Myr" refers to million years. The value for mammals at -65 Ma is estimated based on land temperatures immediately following the
meteorite impact (Kaiho et al., 2016). **Underlined numbers** indicate values corresponding to **before and after extinction events**.





**Table A4.** Number of insect families and extinction percentages over geological time based on Labandeira and Sepkoski (1993)

| Event | Age (Myr) | Insect | Extinction% |
|---|---|---|---|
| | -5 | 610 | |
| | -10 | 610 | |
| | -15 | 605 | |
| | -20 | 600 | |
| | -25 | 590 | |
| | -30 | 580 | |
| | -35 | 420 | |
| | -40 | 410 | |
| | -45 | 400 | |
| | -55 | 390 | |
| | -60 | 330 | |
| K-Pg | -65 | 350 | 8 |
| | -75 | 360 | |
| | -80 | 350 | |
| | -85 | 290 | |
| | -90 | 310 | |
| | -100 | 340 | |
| | -110 | 350 | |
| | -120 | 300 | |
| | -125 | 280 | |
| | -130 | 290 | |
| | -135 | 285 | |
| | -140 | 285 | |
| | -150 | 300 | |
| | -155 | 310 | |
| | -160 | 200 | |
| | -165 | 175 | |
| | -170 | 200 | |
| | -180 | 180 | |
| | -190 | 150 | |
| | -195 | 155 | |
| end-T | -200 | 120 | 14 |
| | -205 | 140 | |
| | -210 | 95 | |
| | -220 | 90 | |
| | -230 | 105 | |
| | -235 | 80 | |
| | -240 | 70 | |
| | -245 | 90 | |
| P-T | -250 | 110 | 35 |
| | -260 | 170 | |
| | -270 | 120 | |
| | -280 | 80 | |
| | -290 | 120 | |
| | -300 | 90 | |
| | -310 | 40 | |



| | |
|---|---|
| -320 | 20 |
| -330 | 15 |
| -340 | 5 |
| -350 | 3 |
| -360 | 1 |
| -370 | 1 |
| -380 | 1 |
| -390 | 1 |
| -400 | 1 |
| -410 | 1 |
| Average | 19 |

Data for **-390 Myr** and **-410 Myr** are sourced from **Engel and Grimaldi (2004)**. **Underlined numbers** indicate values corresponding to

**before and after extinction events**.



**Table A5.** Projected numbers of insect, terrestrial tetrapod, and marine animal families in the future

| Event | Time t (Gyr) | Insect Case 1 | Tetrapod Case 1 | Marine animal Case 1 | Insect Case 2 | Tetrapod Case 2 | Marine animal Case 2 | Insect Case 3 | Tetrapod Case 3 | Marine animal Case 3 | Insect Case 4 | Tetrapod Case 4 | Marine animal Case 4 |
|---|---|---|---|---|---|---|---|---|---|---|---|---|---|
| | 1.496 | 0 | 0 | 0 | 0 | 0 | 0 | 0 | 0 | 0 | 0 | 0 | 0 |
| 16 | **1.446** | 0 | 0 | 0 | 0 | 0 | 0 | 0 | 0 | 0 | 0 | 0 | 0 |
| | 1.446 | 0 | 0 | 0 | 0 | 0 | 0 | 0 | 0 | 0 | 0 | 0 | 0 |
| | 1.402 | 0 | 0 | 0 | 0 | 0 | 0 | 0 | 0 | 0 | 0 | 0 | 0 |
| 15 | **1.352** | 0 | 0 | 0 | 0 | 0 | 0 | 0 | 0 | 0 | 0 | 0 | 0 |
| | 1.352 | 0 | 0 | 0 | 0 | 0 | 0 | 0 | 0 | 0 | 0 | 0 | 0 |
| | 1.307 | 0 | 0 | 0 | 0 | 0 | 0 | 0 | 0 | 0 | 0 | 0 | 0 |
| 14 | **1.257** | 0 | 0 | 0 | 0 | 0 | 0 | 0 | 0 | 0 | 0 | 0 | 0 |
| | 1.257 | 0 | 0 | 0 | 0 | 0 | 0 | 0 | 0 | 0 | 0 | 0 | 0 |
| | 1.213 | 0 | 0 | 0 | 0 | 0 | 0 | 0 | 0 | 0 | 0 | 0 | 0 |
| 13 | **1.163** | 0 | 0 | 0 | 0 | 0 | 0 | 0 | 0 | 0 | 0 | 0 | 0 |
| | 1.163 | 0 | 0 | 0 | 0 | 0 | 0 | 0 | 0 | 0 | 0 | 0 | 0 |
| | 1.118 | 0 | 0 | 0 | 0 | 0 | 0 | 0 | 0 | 0 | 0 | 0 | 0 |
| 12 | **1.068** | 0 | 0 | 0 | 0 | 0 | 0 | 0 | 0 | 0 | 0 | 0 | 0 |
| | 1.068 | 0 | 0 | 0 | 0 | 0 | 0 | 0 | 0 | 0 | 0 | 0 | 0 |
| | 1.024 | 0 | 0 | 0 | 0 | 0 | 0 | 0 | 0 | 0 | 0 | 0 | 0 |
| 11 | **0.974** | 0 | 0 | 0 | 0 | 0 | 0 | 0 | 0 | 0 | 0 | 0 | 0 |
| | 0.974 | 0 | 0 | 0 | 0 | 0 | 0 | 0 | 0 | 0 | 0 | 0 | 0 |
| | 0.929 | 0 | 0 | 0 | 0 | 0 | 0 | 0 | 0 | 0 | 0 | 0 | 0 |
| 10 | **0.879** | 0 | 0 | 0 | 0 | 0 | 0 | 0 | 0 | 0 | 0 | 0 | 0 |
| | 0.879 | 0 | 0 | 0 | 0 | 0 | 0 | 0 | 0 | 1 | 0 | 0 | 0 |
| | 0.835 | 0 | 0 | 0 | 0 | 0 | 0 | 0 | 0 | 5 | 0 | 0 | 0 |
| 9 | **0.785** | 0 | 0 | 0 | 0 | 0 | 0 | 0 | 0 | 2 | 0 | 0 | 0 |
| | 0.785 | 0 | 0 | 2 | 0 | 0 | 2 | 0 | 0 | 32 | 0 | 0 | 1 |
| | 0.740 | 0 | 0 | 4 | 0 | 0 | 4 | 7 | 3 | 79 | 0 | 0 | 4 |
| 8 | **0.690** | 0 | 0 | 3 | 0 | 0 | 3 | 1 | 0 | 28 | 0 | 0 | 3 |
| | 0.690 | 74 | 37 | 189 | 74 | 37 | 189 | 103 | 47 | 358 | 71 | 26 | 170 |
| | 0.646 | 200 | 100 | 305 | 200 | 100 | 305 | 277 | 127 | 577 | 190 | 70 | 274 |
| 7 | **0.596** | 141 | 70 | 206 | 141 | 70 | 206 | 195 | 76 | 391 | 134 | 49 | 186 |
| | 0.596 | 254 | 127 | 345 | 254 | 127 | 345 | 352 | 175 | 654 | 242 | 89 | 311 |
| | 0.551 | 371 | 185 | 427 | 371 | 185 | 427 | 514 | 256 | 810 | 352 | 129 | 385 |
| 6 | **0.501** | 239 | 82 | 225 | 239 | 82 | 225 | 331 | 114 | 426 | 227 | 58 | 202 |
| | 0.501 | 397 | 205 | 450 | 397 | 205 | 450 | 550 | 284 | 853 | 377 | 144 | 405 |
| | 0.457 | 397 | 205 | 451 | 397 | 205 | 451 | 549 | 284 | 855 | 377 | 143 | 406 |
| 5 | **0.407** | 238 | 82 | 309 | 238 | 82 | 309 | 329 | 113 | 428 | 226 | 57 | 278 |



| | $t$ | | | | | | | | | | | | |
|---|---|---|---|---|---|---|---|---|---|---|---|---|---|
| | 0.407 | 397 | 205 | 618 | 397 | 205 | 618 | 549 | 284 | 855 | 377 | 143 | 556 |
| | 0.407 | 397 | 205 | 618 | 397 | 205 | 618 | 549 | 284 | 855 | 377 | 143 | 556 |
| C3 | ***0.4*** | 305 | 158 | 475 | 305 | 158 | 475 | 305 | 158 | 475 | 290 | 110 | 428 |
| | 0.4 | 610 | 315 | 950 | 610 | 315 | 950 | 610 | 315 | 950 | 580 | 221 | 855 |
| | 0.362 | 610 | 315 | 950 | 610 | 315 | 950 | 610 | 315 | 950 | 580 | 221 | 855 |
| 4 | **0.312** | 494 | 192 | 703 | 494 | 192 | 703 | 494 | 198 | 703 | 469 | 139 | 633 |
| | 0.312 | 610 | 315 | 950 | 610 | 315 | 950 | 610 | 315 | 950 | 580 | 221 | 855 |
| | 0.268 | 610 | 315 | 950 | 610 | 315 | 950 | 610 | 315 | 950 | 580 | 221 | 855 |
| 3 | **0.218** | 494 | 198 | 703 | 494 | 198 | 703 | 494 | 198 | 703 | 469 | 139 | 633 |
| | 0.218 | 610 | 315 | 950 | 610 | 315 | 950 | 610 | 315 | 950 | 580 | 221 | 855 |
| | 0.173 | 610 | 315 | 950 | 610 | 315 | 950 | 610 | 315 | 950 | 580 | 221 | 855 |
| 2 | **0.123** | 494 | 198 | 703 | 494 | 198 | 703 | 494 | 198 | 703 | 469 | 139 | 633 |
| | 0.123 | 610 | 315 | 950 | 610 | 315 | 950 | 610 | 315 | 950 | 580 | 221 | 855 |
| | 0.079 | 610 | 315 | 950 | 610 | 315 | 950 | 610 | 315 | 950 | 580 | 221 | 855 |
| 1 | **0.029** | 494 | 198 | 703 | 494 | 198 | 703 | 494 | 198 | 703 | 469 | 139 | 633 |
| | 0.029 | 610 | 315 | 950 | 610 | 315 | 950 | 610 | 315 | 950 | 580 | 221 | 855 |
| 0 | **0** | 610 | 315 | 950 | 580 | 221 | 855 | 610 | 315 | 950 | 580 | 221 | 855 |
| | 0 | 610 | 315 | 950 | 610 | 315 | 950 | 610 | 315 | 950 | 610 | 315 | 950 |

Time $t$: Bold numbers indicate ages corresponding to **major mass extinction events**, while underlined numbers represent **periods of recovery following these extinctions**. Normal numbers correspond to the periods just before extinction events. Explanations of **cases 1–4** are provided in **Methods section**.


**Acknowledgments**

I thank anonymous referees for their valuable comments.

**Authorship contributions**

Kunio Kaiho: Conceptualization, Formal analysis, Investigation, Methodology, Writing – original draft, Writing – review & editing.

**Competing interest declaration**

The author declares no competing interests.


**Data availability statement**



Data is provided within the manuscript or supplementary information files.

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
