# Peer review of "Future diversity and lifespan of metazoans under global warming and oxygen depletion"

_EGUsphere, 2025_

## Referee Comment (RC1)

**Review of „Future diversity and lifespan of metazoans under global warming and oxygen depletion" by Kunio Kaiho**

**Summary**

In this research article, Kunio Kaiho presents novel findings on the future development of metazoan diversity in superterranean, subterranean, surface-water, and deep-water habitats based on diversity changes in the past. By incorporating seven different environmental drivers, the author projects the complete extinction of metazoans within the next 700 million years, which is 300–400 million years earlier than previously estimated.

**General comments**

Overall, the manuscript is well written and provides novel insights into an important field of research. The language is almost perfect, clear, and easy to follow. However, there are a few general points that should be addressed before final publication of the article.

Neither the Introduction nor the Discussion provides much context regarding previous research efforts. While the Introduction nicely explains the different environmental drivers incorporated into the current study, it is unclear what previous research entailed and what the current study adds to it. These aspects should be included in the revised manuscript. Similarly, the Discussion repeats the major results of the current study without discussing them in the context of previous findings. For example, it is repeatedly mentioned throughout the manuscript that the current study projects metazoan extinction to occur 300–400 million years earlier than previous estimates, but these previous estimates are not further specified. What differences between previous studies and the current study may cause these different results? Why are the results of the current study more/similarly realistic? These questions should be addressed in the Discussion.

In addition, I think that some parts of the Methods section are difficult to follow. Firstly, this section uses many abbreviations, but not all of them are defined in the text itself, only in figure/table captions (e.g., PAL is only defined in the caption of Fig. 1). Secondly, many terms are unclear to the reader and require further explanation (e.g., what exactly are diversity rates and what is the difference between survival rates and survival area rates?). Thirdly, the argumentation is partly difficult to follow since the required explanations are either insufficient or provided later in the Results or Discussion section. I recommend adding further explanations and revising the structure of the manuscript where necessary. I give specific examples in the "Specific comments" section.

**Specific comments**

- L. 29: I would replace "all known forms of life" by "**almost** all known forms of life", e.g., tardigrades can survive temperatures higher than 100°C.
- L. 32: Can you shortly explain what $C_3$ and $C_4$ plants are?
- L. 52-53: This sentence disrupts the flow of the text. Since the corresponding information was just mentioned a few paragraphs earlier, the sentence is not necessary in my opinion.
- L. 95: "These data are applied in section 2.2." – The current section is 2.2, so I do not understand this sentence.

- Section 2.3: Are only records of marine metazoans available? If yes, the possible impacts of this limitation should be discussed.
- L. 97: What exactly is meant by "diversity rates"?
- L. 98: What is PAL?
- L 106-107: I think you mean that oxygen levels drop in the habitats of metazoans and not in metazoans themselves, right?
- L. 108: Why do terrestrial metazoans require an ozone layer for evolutionary adaptation? (This is explained in l. 491-493, but I would already explain it here).
- L. 110-115: What were the main reasons for mass extinction during these events?
- L. 131: There is no red curve in Fig. 1. Do you mean the orange curve?
- L. 147: Which were the five largest mass extinction events?
- L. 152: I thought $\Delta Tec$ was estimated using SST data as stated in the previous section?
- L. 163: What is sill?
- L. 167-168: How are long-term changes in $CO_2$ and $SO_2$ emissions related to short-term temperature anomalies?
- L. 191: Why do you use regions with oceanic climate?
- L. 196: Does a gradient of 15°C only apply to warm conditions or why do you explicitly mention warm conditions here?
- L. 199: Where exactly do the 5°C come from?
- L. 201: Same as l. 191 and l. 196: Why do you use data from warm coastal cities?
- L. 207: There is no section 2.3.3. Do you mean 2.6.3?
- Equation 11: What is $\Delta LT$? (LT is only defined in the caption of Fig. 2)
- Sect. 2.7.1: This section is quite hard to follow since many abbreviations are used. Maybe it would help to spell out the abbreviations from time to time.
- L. 244-248: I think it should already be mentioned here that different scenarios are analyzed.
- L. 249-252: I think it would be helpful to provide a brief description of the different events. Some description is given in Sect. 3.2, but I believe that including such a description earlier on would give the reader a better understanding.
- L. 252: What exactly is the survival area rate and what is the difference to the survival rate?
- L. 256-257: What exactly do the rates StR, SwR, UR, and DR describe?
- L. 259: Could you explain more clearly how the SAR is calculated?
- L. 263: I cannot follow the argumentation here. Why should $SAR_D$ approximate $SAR_S$?
- L. 268: How did you determine the impact of food scarcity on survival rates?
- L. 275: And the other events?
- L. 285-286: Is this reasonable? The limitations of this assumption should be discussed.
- L. 287-293: I cannot follow here. Why are metazoans extinct at 0.97 Gyr if 2% remain? And don't you state in other parts of the manuscript (e.g., the Abstract, l. 460, and the Conclusions) that according to your calculations, metazoans go extinct at 0.7 and not 0.97 Gyr?
- L. 304: What exactly is numerical age?
- Sect. 3.1: I think this description would have been more helpful in the Methods section somewhere between Sects. 2.5 and 2.7. Then the reader could better understand the different survival rates etc.
- L. 364: What exactly do you mean by abrupt climate events? Are you referring to volcanic eruptions and meteorite impacts? If yes, I recommend stating this here again.
- L. 385-388: Could you also give current numbers for comparison?

- L. 592: Are you sure that your study is the first to reveal that?
- Fig. 1:
    - I do not understand what the green open diamond symbols denote exactly. Can you maybe explain again in other words?
    - Would it be possible to add some sort of legend to the atmospheric oxygen level graph that specifies the impact on metazoans?
- Fig. 2 (l. 630): Not only silhouettes of terrestrial plants are shown, so maybe write "silhouettes of metazoans and terrestrial plants"?
- Fig. 3: The yellowish-green is hard to distinguish from the green, so maybe use a different color?
- Table A2: What is SD? What does aftermath warming mean?
- L. 720 and 725: before or after?
- L. 727-728: Why do the underlined numbers occur just before major mass extinction events if they represent periods of recovery? Something seems wrong here.

**Technical corrections**
- L. 43: have  triggered historical mass extinctions
- L. 61: This manuscript was written by only one author, right? I would use "I" instead of "we"; there are other occurrences throughout the manuscript.
- L. 89 and 96: I think it should be "Past records **of**"
- L. 204: LMMT
- L. 213: 2.5 m
- L. 222: GATES values are  equal
- L. 237: where $D_t$ represents
- L. 324: 1 m
- L. 378: "However" does not seem to fit here.
- L. 398: primary productivity?
- L. 430: estimating  the future diversity
- L. 504: illustrate
- L. 569-573: This is a repetition of l. 566-569 and should be deleted.
- Fig. 2:
    - Atmospheric oxygen level relative to  present atmospheric level
    - Diversity rate relative to  the Paleozoic maximum
- L. 659 and 729: the Methods section
- L. 676: The capitalization in this sentence seems odd.
- L. 686: rate in the future
- L. 687-688:  compared to terrestrial plants
- Table 2:  at the family level
- Table A2: Earth's average surface temperature  including the long-term trend, long-term cycle, and short-term event with temperature anomalies and decreasing $CO_2$ and $SO_2$ emis-sions  due to the decrease in mantle potential temperature during major mass extinction event from 0.7 billion years (Gyr) before the present to 1.5 Gyr into the future

---

## Referee Comment (RC2)

**Kaiho "Future diversity and lifespan of metazoans under global warming and oxygen depletion"**

**General comments**

This manuscript presents a novel and comprehensive model that predicts the lifespan of metazoans on Earth over the next 1.5 billion years. It combines a wide range of geological, climate, and biological data to create a compelling narrative about the long-term future of complex life. The main conclusion—that metazoans will go extinct in approximately 700 million years, much earlier than previous estimates—is a significant and provocative contribution. However, given the immense timescale and the complexity of the integrated model, it is essential to carefully present assumptions, uncertainties, and limitations to address skeptical readers convincingly.

This is a potentially high-impact manuscript that aligns well with the scope of this journal. Its bold projections are its main strength. To maximize its impact and increase the chances of publication, the authors need to strengthen the presentation regarding the treatment of uncertainty and provide more robust justifications for the key parameters that drive the model. By doing so, they can turn a compelling thought experiment into a foundational and highly cited piece of future Earth system science. I put specific comments about each section below.

**Specific comments**

**[introduction]**

While the author currently identifies several individual gaps, such as cyclical rhythms "(L. 40), have not yet been fully incorporated, and abrupt events "(L. 46) have not yet been factored in," this can feel somewhat fragmented; they could be woven together to create a single, compelling argument for why your study is necessary. The author can reframe the problem to highlight the interaction of multi-timescale forcings as the central, unexplored challenge. I recommend adding a concise, overarching problem statement just before your final thesis paragraph (L. 48). Moreover, the final paragraph should then directly answer this problem statement. The list of seven critical factors is comprehensive, but the paragraph's impact will be greater if it highlights the integrative model itself as the core novelty.

The transitions between the main ideas, such as long-term trends, cycles, and abrupt events, can be made smoother, and the link between physical forcings and biological impacts can be clarified. A key conceptual point is the difference between predictable cycles and unpredictable events. Strengthen this transition by adding a sentence that clearly contrasts their timescales. For example, at the end of the cycles paragraph: "...have not yet been fully incorporated into projections of future surface temperatures. Beyond these predictable, multi-million-year cycles, Earth's climate is also punctuated by unpredictable, abrupt events..." Then, the author briefly explains how physical drivers influence biological outcomes modeled in the system. Regarding Tectonic Cycles, the author adds a phrase on how these cycles might influence plant crises or metazoan survival (e.g., through changes in continental configuration that affect weathering rates or create or eliminate refugia). Additionally, regarding abrupt events, the author can include a sentence explaining that these events are modeled as drivers of "step-changes" in biodiversity, which can reset recovery trajectories.

The author can enhance clarity and scientific rigor by refining specific sentences. The sentence "Global warming will accelerate terrestrial weathering..." serves as a key link. The author should add a few words to clarify how the mechanism works, which would add depth. In the final paragraph, the sentence "Projections are based on temperature modeling, thermal tolerance limits..." appears to preview the Methods section. Therefore, the author can rephrase it as, "Our model projections combine future scenarios of temperature and oxygen levels with established data on metazoan thermal tolerance and diversity trends."

The final two paragraphs contain some repetitive information and could be merged to create a more powerful and concise conclusion to the introduction. The author can integrate the description of metazoan evolution and thermal tolerance into the core thesis paragraph. This creates a single, strong paragraph that states what you did, what you based it on, and the scope of your analysis.

**[Methods: 2.1]**

The statement "Assuming that the icehouse-greenhouse cycle and major mass extinctions continue at the same pace as in the past" (L. 66) is a significant assumption that is central to your model. This requires a brief justification. Is there a reference supporting the consistency of these cycles over billion-year timescales? A sentence citing relevant geological timescale studies would greatly strengthen this.

The method assumes that the relationships between temperature/O2 and biodiversity observed in the deep past will also apply to the entire future of complex life. This core assumption should be clearly acknowledged as a potential limitation or justified with a solid rationale.

Although the author mentions specific groups later, the term "metazoan extinction" (L. 72) in point 3 is quite broad. It would be helpful to clarify early on that your study focuses on the specific groups listed —marine metazoans, terrestrial tetrapods, and insects —as proxies for overall metazoan diversity.

The nature of these abrupt large-scale future climate events (Events 1-16; L. 81 and 83) remains unclear. Are they modeled as analogues to the Big Five? Are they stochastic events? A brief explanation of how these were defined and selected would be very helpful.

The phrase "framework that includes temperature trends, oxygen levels, and C3-C4 plant crises" (L. 78-79) is too general. The author needs to describe the actual model. Is it a statistical correlation? A dynamic system model? A set of conditional rules? Then, C3-C4 plant crises (L. 79) are mentioned but are not introduced earlier. The author should briefly explain what this crisis involves and why it is a factor in your model.

Regarding methodological precision, the author used "compiled records (L. 63)" and "analyzed the relationship (L. 64)," but should specify the particular statistical methods, such as correlation analysis and regression modeling. At point 3 (L. 69~), the transition from "local extinction temperature" to "global average surface temperature" at which extinction occurs is a crucial scaling step. This process should be clearly explained in the detailed methods (2.6), and the summary should hint at its complexity (e.g., "...was scaled to a global average surface temperature using latitudinal gradients"). Finally, in the last paragraph (L. 85~), the sentences about gradual extinctions (orange curve) and the estimated durations (0 Gyr and 0.05 Gyr) seem somewhat out of place in the method summary. They are better suited for the Results section or a dedicated part of the detailed methods. The "0 Gyr" duration for temperature anomalies is confusing and needs clarification.

**Revised first paragraph:**

To project the future lifespan of metazoans on Earth, we developed a multi-step model that integrates relationships derived from past climate and biodiversity dynamics. Our core assumption is that the pacing of icehouse-greenhouse cycles (~0.3 Gyr) and major mass extinctions (~0.094 Gyr), along with the physiological constraints on metazoans, will remain consistent in the future [references, if possible]. The analysis proceeded as follows:

- 1. Past Climate and Diversity Baselines: -- correspond to subsection 2.2
- 2. Future Temperature Projections: -- correspond to subsection 2.3
- 3. Metazoan Thermal Tolerance Limits: -- correspond to subsection 2.4
- 4. Oxygen-Biodiversity Relationship: -- corresponds to subsection 2.5
- 5. Integrated Future Diversity Model: -- corresponds to subsection 2.6

The paragraph about five past events, 16 events, and gradual extinctions (L. 81-88) could be moved or integrated into points 2 and 5 above for better flow.

To maintain consistency across subsections, the author should verify that the titles of subsections 2.2 through 2.6 directly correspond to points 1 through 5 in the Method summary and expand on them. For example, 2.2 should include a detailed methodology for point 1, "Past Records." Therefore, if possible, the author should either revise the outline based on your Method summary or revise the Method summary itself.

The subsections (2.2 to 2.6 (or 2.7)) must flesh out the details summarized in 2.1. For each step, the author needs to specify: About data sources, the author mentions specific databases or publications, and provides tables (e.g., Tables A1 and A2). About analytical techniques, how did the author analyze the relationship? Was it a linear regression? A non-linear model? Specify the statistical tests and the software/tools used. About Quantitative Definitions, what are the numerical thresholds for "low, mid, and high latitudes"? What defines a "C3-C4 plant crisis" in your model? Define these operationally. Model Parameters: The values "0.3 Gyr" and "0.094 Gyr" are key model inputs. Justify these choices with references beyond the general (Erwin et al., 1987). How sensitive are your results to these specific values?

**[Methods: 2.2]**

The issue with this section is the jump from describing the data to stating the objective. The sentence "To estimate future abrupt climate changes and biotic crises, we selected the five largest mass extinctions..." (L. 91-92) is an objective, not a method. Thus, the author should restructure the text to first present the data and then explain the analytical step.

The author also needs to provide a more detailed explanation of how they selected and used these "five major mass extinctions." This clarifies the reason for choosing specific events and helps ensure the methodology can be reproduced.

The author mentions two different types of data: 1) long-term climate cycles (icehouse-greenhouse) and 2) short-term temperature anomalies during extinction events, but the connection between them is not explained. Please briefly explain the role of each dataset in your overall model.

The final sentence (L. 95), "These data are applied in section 2.2," is redundant as it is already within section 2.2.

**[Methods: 2.3]**

The paragraph (L. 97-L. 104) states that "marine metazoan diversity rates" were "derived from past records," but it does not provide a source for these diversity rates, only for the oxygen levels. This creates a notable gap in reproducibility. The author must provide the reference(s) for your biodiversity data.

What does a "diversity rate" of 0.04 or 1.0 actually mean? Is it a count of families or genera normalized to a maximum value? Is it an estimated measure of richness? Without a clear operational definition, the metric remains ambiguous. Please provide a brief explanation of how the calculation is performed.

The second paragraph (L. 105-108) is mostly interpretative. Phrases such as "suggests that there is a positive relationship," "significant declines," and "appear to require higher oxygen levels" are conclusions based on the data. Remove the entire second paragraph from the Methods section. These should be moved to your Results section (to show the relationship) and the Discussion section (to discuss the possible reasons).

The final sentence (L.108), "These interpretations are further applied in Section 2.4," should be rephrased to emphasize the data and relationship, not the interpretation. The author should rephrase it to something like, "The quantitative relationship between O2 and biodiversity established here (Figure 3) is used in Section 2.4 to describe the specific action, such as constraining future diversity projections based on predicted oxygen levels."

**[Methods: 2.4]**

Averaging percentages from different extinction events, which had vastly different causes and selective pressures, into a single "future diversity loss" value is a highly simplified modeling choice. The author should justify why this is a valid approach for long-term projection. Please add a sentence or two explaining the rationale. The author should also acknowledge the limitations of this approach.

The author lists five events for marine metazoans but only three for tetrapods and insects. Why is there a discrepancy? Is the author averaging five events for the marine group and three for the terrestrial group? This needs to be clarified.

The subsection is titled "Past and future extinction percentages and the interval," but the content does not mention any time interval. The 0.094 Gyr extinction cycle from your initial summary is a crucial parameter. This is where to define it. The author includes the methodology for the interval here.

**[Methods: 2.5]**

The first paragraph (L. 120-123) doesn't serve well as an introduction to this section because the framework for temperature estimation is unclear. Additionally, the section reads more like a collection of individual sub-models rather than a coherent methodology. The most significant gap is clarifying how the mass extinction events are timed and situated within the 2.5 Gyr timeline. Are they periodic, random, or triggered by a specific threshold? This is a key factor affecting the results and remains undefined.

Several critical numbers and equations in this Section are presented without explanation, making it impossible for a reader to evaluate or reproduce your work:

- 1) Why is there an 8°C temperature difference between the icehouse and greenhouse periods? (L. 129) Bond and Grasby (2017) are probably insufficient because their study focuses on specific events. Using a single value over 2.5 Gyr is a big oversimplification that needs strong justification. Therefore, the author should provide a brief context.
- 2) These equations (L.156-161) are the core of your event-based temperature calculation, but they are presented as a "black box." Where do the constants, such as -8.9, 9.9, 2/3, 9.9, 0.00108 ..., originate from? The author cites Kaiho et al. (2022), but it is unclear whether these are empirically derived relationships from that paper or if the author has manipulated their data. This is the most critical part of the section to expand on.
- 3) "A 50% reduction in CO2 and a 67% reduction in SO2 emissions" (L. 167) are striking figures. How were they calculated? Are these the results of your model or the inputs into the model?
- 4) The author mentions the basis for applying "the 0.3 Gyr climate cycle to the next billion years" (L. 126-127), but further explanation is needed. What is the physical mechanism? Additionally, why do its period and amplitude remain constant over billions of years, despite geological changes such as increased solar activity and/or mantle cooling?
- 5) The decision to model the impact of declining mantle temperatures only for the next 1.5 Gyr (L. 168) seems arbitrary. Why not consider the entire 2.5 Gyr?
- 6) The term "sill" (Ts, Ti) is used without being defined for a non-geology specialist. A brief explanation is necessary.

- 7) The formulation of Equation (10),  $Er = SD + Er(\Delta Tc)$ , is problematic. It defines the total error, Er, in terms of itself, creating circular logic. Furthermore, it adds a standard deviation (SD), a statistical measure of dispersion, to an error term,  $Er(\Delta Tc)$ , which likely represents a potential range. The description implies these components are being combined, but the equation's additive form is not logically justified. Clarification is required on the functional form of  $Er(\Delta Tc)$ —is it a root-sum-square, a linear addition, or another method?
- 8) Finally, the claim that the error is "within 0.1 Gyr" is ambiguous; it needs clarification on whether this refers to an uncertainty in age or temperature.
- 9) Concerning Equation 8, the derivation from the Arrhenius equation should be briefly explained or referenced.

**[**Methods**: 2.6]**

The section outlines a logical and organized approach to determining critical thermal thresholds for metazoans, which is a key and valuable contribution. The method of converting physiological limits into the global average surface temperature required for extinction (GATE) is ambitious and clearly segmented by habitat. However, the description currently needs significant clarification and justification to meet the standards of reproducibility:

- Regarding the 5°C adjustment from the Local Monthly Maximum Temperature (LMMT) to the Local Daily Maximum Temperature (LDMT) in Section 2.6.1, point 3, why 5°C? Is this a global average? Is it based on the Weather Spark data cited? This needs a clear justification.
- Regarding the 1°C and 5°C Local Annual Temperature (LAT) adjustments in Section 2.6.1, point 4, the cited reference (Upchurch et al., 1998) is somewhat outdated. Is this still the most relevant source? Please briefly explain the climatic reasons behind this latitudinal gradient in temperature differences.
- The selection of 37° latitude for determining GATE in Section 2.6.1, point 5, appears arbitrary as stated. Why 37°? Is this based on a global average temperature weighting? This step is crucial in determining your final GATE value and must be clearly explained.
- The assumed future burrowing depth of 2.5 meters in Section 2.6.2 is a significant biological assumption. Why choose 2.5 m instead of 1 m or 5 m? Is there evidence that metazoans can or will burrow to this depth for thermoregulation? This requires ecological justification.

The step-by-step process in Section 2.6.1 can be confusing because it mixes the description of what is shown in the figure with the actual methodological steps. It would be clearer to organize it as a standalone explanation. If possible, the author should rewrite 2.6.1 as a numbered or bulleted list that is independent of the figure: 1) Define the upper thermal tolerance (46°C) as the average maximum daily temperature (...) at key latitudes (0°, 30°, 60°, 90°). 2) Establish the warm Earth latitudinal temperature gradient (15°C from 0° to 90°). 3) Adjust this gradient for LMMT by applying a -5°C correction. 4) Adjust LMMT to LAT using a latitudinally variable correction ( $\Delta$ LT). 5) Calculate the GATE from the LAT at a representative latitude (37°N/S).

The text in Section 2.6.1, point 5 states, "The GATE values for St and Sw metazoans are the same as described in Section 2.3.3." However, since your structure indicates that 2.3 is about oxygen levels, this likely refers to a different part of your manuscript, which could cause confusion. All necessary information for understanding the calculation should be in this section or clearly cross-referenced.

The frequent and inconsistent use of acronyms makes the text very difficult to understand. Readers must constantly refer back to remember what they stand for. Therefore, if possible, please consider adding a table: a summary that lists each metazoan group, its defined upper thermal limit, all applied corrections, and the final derived GATE value would greatly enhance clarity and reproducibility.

**[**Methods: 2.7**]**

The three main equations of the core model are shown, but their logic and connections are not clearly explained. Equation (13) presents a problem in this recovery model. Dt-2 refers to diversity two time steps earlier, implying that recovery aims to reach a diversity level similar to what was before the last extinction event, which may not be ecologically realistic. A more common approach would be to target a recovery toward a carrying capacity or a pre-event level. The reasoning behind this needs further clarification. Equation (14) appears to apply to gradual diversity loss preceding an event. It's unclear why this is a separate equation from equation (12) and how the "Survival Rate for gradual changes" (SR) differs from SRC in practice. The model's flow between these equations is not described.

This is a critical issue that significantly hampers readability. The text uses over 20 different acronyms (SRC, SRO, RR, SR, SAR, SARS, SARU, SARD, SRCT, SRCM, StR, SwR, UR, DR, FS, GATE, etc.). Many are non-intuitive, and some are confusingly similar (e.g., SR vs. SRC vs. SRO; SARS as "Survival Area Rate" is an unfortunate choice). The author should create a nomenclature table before Section 2.7, including a list of all acronyms, their full names, and brief definitions. Otherwise, please avoid creating an acronym for every term. Use them sparingly for the most frequently used concepts.

Many of the numerical values used in the model seem arbitrary or lack a clear, reproducible source. This presents the biggest risk to the model's credibility.

Regarding SRC for Event 0 (Anthropocene), the values of 0.95, 0.70, and 0.90 for a "full-scale nuclear war" are very precise. What is the basis for these exact numbers? The citation (Kaiho, 2023) must specifically provide these values or the model that produced them.

Regarding Food Scarcity (FS), a reduction of "0.1–0.5" is a very broad range. How is a specific value selected for a particular calculation? This requires a clear rationale.

Concerning "Underground Rate (UR=0.05) and Deep-water Rate (DR=0.33)," the logic for converting modern lineage proportions (e.g., 15 out of 315 mammalian families) into a future survival rate is not clearly explained. This is a major assumption that requires a strong ecological justification.

For "Recovery Rates (RR)," many RR values (e.g., 0.01-0.3 for events 8-10) are given without a clear, quantitative link to environmental conditions. The justification is qualitative ("reduced capacity," "adaptation challenges"), but the output is a precise number. There needs to be a transparent method for converting the severity of an event into a recovery rate.

Regarding the "C4 Plant Crisis" (L. 287), the timeline is set to "coincide approximately with event 11," and the extinction of metazoans is then "set at 0.97 Gyr." This appears to be a circular argument where the model is adjusted to fit a pre-selected outcome rather than the outcome emerging naturally from the model's mechanics. Is it true?

The Methods section should describe how you calculated diversity, not what the results are. This section often overlaps with presenting results and speculative explanations, such as ... expected to decline (L. 276), ... are expected to evolve (L. 280), and ... are expected to include (L. 290). The author should rephrase these statements to describe the model's rules.

**[Results: 3.1]**

The most significant issue is the lack of a methodological link between your methods and your primary result. The "Results" section should present findings derived from the previously described methodology; however, the description of the temperature curve reads more like an input scenario than a calculated outcome. The core problem is that the reader is told *what* the temperature curve is, but not *how* it was generated. While Section 2.6 details a model for calculating extinction thresholds, it does not explain the foundational climate model that produced the temperature projection itself. The

author must explicitly state the model or data source used to generate the orange curve in Figure 1. Is it an output from a climate model run under specific CO2 scenarios? An extrapolation from past climate data? Or is it derived from astronomical solutions, such as Milankovitch cycles? Crucially, you must identify the primary forcing driver (e.g., increasing solar luminosity, greenhouse gas concentrations) to provide the necessary context.

The same time periods and temperature ranges are described multiple times in slightly different ways (e.g., 0.65-0.95 Gyr vs. 0.7-1.0 Gyr). This repetition is redundant and causes confusion. Paragraph 2 (L. 352-355) describes the trend with specific future time points (0.35, 0.65, 0.95 Gyr), while Paragraph 4 (L. 359-363) restates the trend by re-binning the timeline into Phases A-E. Therefore, the author should combine these into a single, clear description, using the climate phases as the main structural framework and avoiding disconnected time points.

The nature of the abrupt climate changes described in Events 5, 8, 11, and 14 in paragraph 3 (L. 356-358) remains unclear. What physical process causes these "temperature surges" of more than 10°C in "<0.1 million years"? Are they inputs to the model (prescribed forcing) or results (emergent behavior)? The text currently treats them are given. We need to determine whether these are hypothetical events, representations of volcanic or tectonic activity, or outcomes from climate tipping points.

The use of "expected" and "projected" is inconsistent. Since this is a model result, "projected" is more appropriate.

The final paragraph states that Phases B-E are "triggered by" abrupt events (L. 364). This creates a causality dilemma. Is the long-term trend the primary driver, or are the abrupt events the triggers for new phases? The current description makes it seem like both, which is confusing. The author must clarify the relationship between the gradual trend and the abrupt events.

**[Results: 3.2]**

The first paragraph states diversity is "estimated using" (L. 368) several factors but provides no explanation of how this estimation is performed. What is the model? Is it a statistical correlation, a dynamic ecosystem model, or a set of threshold rules? What are the "survival thresholds" (mentioned in 3.2.1) and how are they determined? The reader cannot assess the results without understanding the fundamental rules of the model.

Several claims are presented as facts without justification, moving from scientific projection to pure speculation. "Large volcanic eruptions and asteroid impacts are expected to trigger..." (L. 382-383) The timing and occurrence of these specific events cannot be projected. This must be framed as a scenario or sensitivity test like "In scenarios where large volcanic eruptions occur...".

The line between results and interpretation is often unclear. A Results section should present the data (or model outputs), while the Discussion should explain what they mean. In this section, phrases like "This high-temperature environment will also impact..." or "will drive the complete collapse..." are interpretive. The text often states what will happen as if it is certain, based on the model, rather than just presenting the model's output. Therefore, the author should reframe to simply present the model's findings.

The author provides us with specific numbers (e.g., insect families dropping from 610 to 74–103), but does not inform us how these numbers were generated. This breaks the chain of reproducibility. Therefore, when presenting a key result, briefly link it back to the methodological framework.

The text in this section mentions ranges (e.g., 74–103 insect families) and different scenarios (Figs. 5 and 6), but it doesn't explain the *causes* of these ranges. For a projection of this magnitude, a thorough

exploration of uncertainty is crucial. The author needs to state which parameters are responsible for the ranges explicitly.

Given the highly speculative nature of billion-year projections, using definitive language like "will," "are expected to," or "lead to" is too strong for a Results section. The author should adopt more tentative and precise phrasing that reflects the model-dependent nature of the findings. For example, the author can use phrases such as: "The model projects...", "Under the defined scenarios...", "Simulation results indicate...", and "Our findings suggest...".

The "Aftermath" section (L. 415-428) extends beyond metazoan diversity to explore the fate of all life and the planet. While a compelling conclusion, it exceeds the stated scope of "metazoan diversity change." The author should consider whether some of this material, especially the comparison to Venus, might be better suited for the Discussion.

**[Discussion: 4.]**

This section currently reads more like a summary and extension of the results than a critical discussion. To meet the journal's standards, the work needs to be more thoroughly addressed, including the limitations, uncertainties, and broader implications, shifting from "what we found" to "what it means and how reliable it is."

Section 4.1 (L. 430-460) mainly defends the assumptions rather than thoroughly examining their uncertainties. A robust discussion must openly acknowledge and analyze the weaknesses. Some specific gaps include: 1) The model incorporates highly uncertain components (billion-year climate, biodiversity, oxygen, plant evolution). The combined effect of these uncertainties is not addressed. How reliable is the 0.7 Gyr extinction date, considering this? 2) The model assumes fixed thermal tolerances. The discussion should explicitly consider the possibility of evolutionary adaptation over millions of years, even if the conclusion suggests it might be limited due to fundamental physiological reasons. 3) The survival and recovery rates are key but poorly constrained. A discussion on how sensitive the main conclusion (extinction at ~0.7 Gyr) is to these parameters is essential. Is the outcome unavoidable across a wide range of plausible values? The author can integrate a dedicated subsection, like "4.5 Model Limitations and Uncertainties," that systematically addresses these points. This demonstrates scholarly rigor and strengthens the paper.

The author often restates results (e.g., "The combined effects... will drive a substantial decline") and ventures into highly speculative territory (e.g., Section 4.3 on intelligent life strategies and Mars) without a clear framing. For example, Section 4.3, while interesting, is tangential to the core scientific findings about metazoan diversity. It risks detracting from the paper's focus and lacks the scientific support found in other sections. Thus, the author reframes result restatements as a setup for interpretation. For speculative sections, clearly label them as such and connect them directly to the model's projections.

While citations are used, the discussion often *uses* them to support the model's assumptions rather than to contrast *or* synthesize the model's findings with other published work. Therefore, the author should actively engage with alternative viewpoints or models, positioning their work within the ongoing scientific conversation.

The central message that metazoans will decline after 0.4 Gyr and go extinct at 0.7 Gyr due to combined temperature, O2, and plant crises is repeated multiple times (e.g., in 4.1, 4.1.2, 4.2, 4.4). This repetition reduces its impact. The author should simplify the main narrative, state the key conclusion clearly once, and then use the following sections to discuss different aspects of it (limitations, mechanisms, implications) without restating the conclusion verbatim.

There is a clear copy-paste error in Section 4.4 (L. 566-574), where a whole paragraph is duplicated ("Rising Global Temperatures at Events 8–10...").

**[Conclusion: 5.]**

The second half of the conclusion goes far beyond the paper's scientific findings into policy, ethics, and futurology. The author's statements, such as "To mitigate biodiversity loss, advanced species may need to implement strategies..." and "emphasizing the urgency of proactive measures," are not conclusions derived from your model. While interesting, the author should phrase it more neutrally. The author should focus the conclusion on the scientific findings. The speculative strategies (aerosol shielding, space colonization) can be mentioned as potential consequences of the findings, but not as prescriptive "needs." Avoid language that tells the reader what is "urgent."

The entire study relies on a model with significant uncertainties (as noted in comments for the discussion). The conclusion states the 0.7 Gyr timeline as a definitive result without any qualification. The author should consider adding a sentence that acknowledges these inherent uncertainties.

The phrase "this study is the first to reveal that humanity exists at the midpoint of metazoan lifespan" makes a very strong claim in the conclusion. While it might be true, it sounds self-promoting. The most suitable place to highlight this novelty is in the Introduction and Discussion sections. The author could rephrase this to emphasize the finding itself. For example, "An interesting corollary of this timeline is that humanity appears near the midpoint of Earth's metazoan history."

The main point, extinction at 0.7 Gyr, is repeated three times in a very brief text. While repetition can emphasize a point, here it limits the development of a more nuanced final message. The author can condense the key finding into one strong statement at the beginning and use the remaining space to discuss its causes and implications.

---

## Author Comment (AC1)

Dear editor, this document contains a point by point reply to the issues raised by the reviewers (in red); changes made in the MS are indicated in blue. All original comments by the reviewers are left in black font.

**Referee comment from Anonymous Referee #1**

**Summary**

In this research article, Kunio Kaiho presents novel findings on the future development of metazoan diversity in superterranean, subterranean, surface-water, and deep-water habitats based on diversity changes in the past. By incorporating seven different environmental drivers, the author projects the com-plete extinction of metazoans within the next 700 million years, which is 300–400 million years earlier than previously estimated.

**General comments**

Overall, the manuscript is well written and provides novel insights into an important field of research. The language is almost perfect, clear, and easy to follow. However, there are a few general points that should be addressed before final publication of the article.

Neither the Introduction nor the Discussion provides much context regarding previous research efforts. While the Introduction nicely explains the different environmental drivers incorporated into the current study, it is unclear what previous research entailed and what the current study adds to it. These aspects should be included in the revised manuscript.

Author Reply: I agree with the comments. I revised words marked in blue in the attached manuscript (Kaiho Revise Marked 1).

Added lines 38-43 in the Introduction. Estimates for the end of Earth's biosphere published after 2010 vary widely. Projections based on surface temperature range from 1.0 to 5.0 Gyr (O'Malley-James et al., 2012; Rushby, 2013; Leconte et al., 2013; Wolf and Toon, 2015), while scenarios based on $CO_2$ depletion yield estimates of 0.84–1.08 Gyr (Rushby, 2015; Ozaki and Reinhard, 2021). Mello and Friaça (2019) suggest that biosphere collapse is unlikely before 1.5 Gyr based on thermal constraints. However, a decline in atmospheric oxygen to 1% PAL within $1.08 \pm 0.14$ Gyr ($1\sigma$) as predicted by Ozaki and Reinhard (2021) may lead to the earlier extinction of metazoan life.

Added lines 62-63 in the Introduction. This study builds upon previous models by integrating anthropogenic crises (Waters et al., 2011; Ceballos et al., 2015; Waters et al., 2016; Kaiho, 2022, 2023), cyclical climate rhythms, and abrupt climate events. It considers seven key factors—anthropogenic crises, long-term warming, cyclical climate rhythms, abrupt events, $C_3$ plant collapse, $C_4$ plant decline, and oxygen depletion—to project metazoan extinction over the next 1.5 Gyr. Projections are grounded in temperature and oxygen modeling, thermal tolerance limits, and observed metazoan diversity trends.

Similarly, the Discussion repeats the major results of the current study without discussing them in the context of previous findings. For example, it is repeatedly mentioned throughout the manuscript that the current study projects metazoan extinction to occur 300–400 million years earlier than previous estimates, but these previous estimates are not further specified. What differences between previous studies and the current study may cause these different results? Why are the results of the current study more/similarly realistic? These questions should be addressed in the Discussion.

Author Reply: Added 3.2 (Results) and 4.1 (Discussion) sections.

**3.2 Metazoan lifetime estimation under four scenarios**

When only the long-term warming trend driven by the gradual increase in solar luminosity is considered (Mello and Friaça, 2019), metazoans are projected to go extinct at approximately 1.3 Gyr, based on the intersection of the black dashed line with the upper boundary of GATEU90 in Figure 1. Under this scenario, surface-dwelling metazoans are expected to go extinct slightly earlier, at 1.2 Gyr, based on the same trend intersecting the upper boundary of GATES90.

When long-term cyclical fluctuations between icehouse and greenhouse phases are incorporated into the model, extinction is projected to occur at 1.2 Gyr, corresponding to the intersection of the orange line with the top of GATEU90. In this case, surface metazoans are expected to disappear by 1.0 Gyr, as indicated by the intersection with GATES90.

Incorporating average surface temperature anomalies associated with past mass extinction events further lowers the projected extinction time to 1.0 Gyr, based on the red circle's intersection with the upper boundary of GATEU90. However, actual complete extinction is expected to occur earlier, between 0.7 and 0.8 Gyr, due to compounded survival rate reductions (0.01–0.1) caused by food scarcity (FS) and oceanic anoxia. These stressors are expected to be triggered by elevated temperatures during abrupt extinction events (Events 8–10), which involve the collapse of surface metazoan populations and severe reductions in primary productivity caused by global-scale extreme warming (see Table A5).

By 0.7 Gyr, atmospheric oxygen and $CO_2$ levels are not anticipated to be the dominant extinction drivers. Instead, the primary cause of complete extinction is projected to be extreme surface warming, resulting from the combined effects of increased solar luminosity, long-term climatic oscillations, and large-scale volcanic activity. Therefore, the final extinction of all metazoan life is projected to occur at $0.7 \pm 0.05$ Gyr.

**4 Discussion**

**4.1 Lifetime estimation**

Previous studies have proposed varying estimates for the remaining lifespan of metazoan life on Earth. A 1.2 Gyr estimate has been cited as a plausible median value (Jebari and Sandberg,

2022), while 1.3 Gyr is projected based solely on the long-term warming trend driven by increasing solar luminosity (Fig. 1), and 1.1 Gyr corresponds to the point at which atmospheric oxygen is predicted to decline to 1% PAL (Ozaki and Reinhard, 2021).

In contrast, the present study estimates the total remaining metazoan lifespan at approximately 0.7–0.8 Gyr from now, which is 300 to 600 million years earlier than previous estimates. This discrepancy arises because earlier models do not account for the compounding effects of long-term cyclical icehouse–greenhouse climate phases, average surface temperature anomalies during mass extinction events, and cooling effects linked to mantle temperature decline.

The estimates from the current study are considered more realistic, as they incorporate all major drivers of surface temperature variability—including long-term trends, cyclical oscillations, and abrupt catastrophic events—providing a more comprehensive projection of the environmental conditions leading to metazoan extinction.

Author Reply: Revised to "Ultimately, a final extinction event—likely initiated by large-scale volcanism—will be primarily driven by extreme global warming." in lines 16-17 (Abstract). Added ", primarily driven by global warming" in line 704-705 (Conclusions). The sentence is "This scenario indicates that complete metazoan extinction is expected to occur within 0.7–0.8 Gyr, primarily driven by global warming."

In addition, I think that some parts of the Methods section are difficult to follow. Firstly, this section uses many abbreviations, but not all of them are defined in the text itself, only in figure/table captions (e.g., PAL is only defined in the caption of Fig. 1). Secondly, many terms are unclear to the reader and require further explanation (e.g., what exactly are diversity rates and what is the difference between survival rates and survival area rates?). Thirdly, the argumentation is partly difficult to follow since the required explanations are either insufficient or provided later in the Results or Discussion section. I recommend adding further explanations and revising the structure of the manuscript where necessary. I give specific examples in the "Specific comments" section.

Author Reply: I agree with the comments. Revised the 2.7 section in Methods: Future metazoan diversity estimation as the following section (revised parts are marked in blue). Also revised Figures 5–7 based on the revised 2.7 section.

2.7 Future metazoan diversity estimation

Future changes in metazoan diversity are influenced by the ongoing anthropogenic crisis ("event 0"), subsequent mass extinction events (events 1–11), $C_3$ and $C_4$ plant crises, and gradual oxygen depletion (see Table 2). The projected diversity of insects, terrestrial tetrapods, and marine metazoans is estimated before extinction events, immediately after, and following recovery using the equations below:

Diversity loss due to extinction event:

$$D_t = D_{t-1} \times SRC \times FSR \qquad (12)$$

Diversity following recovery (after 50 Myr from the extinction event to before the next event):

$$D_{t+1} = D_t + (D_{t-1} - D_t) \times RR \qquad (13)$$

Recovery Rate (RR):

$$RR = RRW \times RRO \times RRP \qquad (14)$$

In these equations, $D_t$ represents metazoan diversity at time step $t$, corresponding to the level immediately following an extinction event. $D_{t-1}$ denotes the diversity prior to the extinction event, while $D_{t+1}$ reflects the diversity after the recovery phase, measured at the midpoint between extinction events. SRC is the Survival Rate associated with climate-driven crises, including mass extinctions and $C_3$–$C_4$ plant collapses. *FSR* is the Food Scarcity Rate, reflecting the impact of the collapse of plants and primary producers. The total Recovery Rate (RR) is calculated as the product of three components: RRW (recovery from gradual warming), RRO (recovery from progressive oxygen decline), and RRP (recovery from decreased primary productivity due to $CO_2$ reduction).

These equations are applied sequentially across time steps from event 0 through event 16, encompassing extinction episodes, recovery phases that conclude at the midpoint between events, and the subsequent interval leading up to the next extinction event (see Table A5).

**2.7.1 Survival Rate associated with Climate change (SRC)**

Survival Area Rate (SAR) is rate of land and ocean area where metazoans survive in all land and ocean area (km2/km2). When extinction occurred in 0–10, 0–20, 0–30, 0–40, 0–50, 0–60, 0–70, 0–80, and 0–90° latitudes by warming, SAR values are defined as 0.83, 0.66, 0.50, 0.36, 0.24, 0.14, 0.06, 0.02, 0.00, respectively, under the same rate of land and ocean in those latitudes. The rates SAR are decided by only temperature 46 °C using Figure 2. $SAR_S$ is Survival Area Rate for St and Sw metazoans, $SAR_U$ and $SAR_D$ are Survival Area Rate for U and D metazoans, respectively. These SAR are obtained from GATES, GATEU, and GATED in Figure 2.

The SRC (survival rate by climates) is calculated as:

$$SRC_T = 0.95 \times SAR_S + 0.05 \times SAR_U \qquad (15)$$

$$SRC_M = 0.67 \times SAR_S + 0.33 \times SAR_D \qquad (16)$$

Here, $SRC_T$ and $SRC_M$ represent the terrestrial and marine SRC, respectively. The coefficient 0.05 corresponds to the proportion of subterranean metazoan families among all terrestrial metazoan families (15 out of 315), based on mammalian lineage data (Recknagel & Trontelj, 2021; Benton, 2010). The remaining 0.95 represents superterranean (surface-dwelling) taxa. The coefficient 0.33 reflects the proportion of deep-sea fish families among all marine fish families, based on an estimated 6% of teleost species restricted to depths >200 m (Miller et al., 2022), with scaling applied via the species-genus-family extinction relationship from Kaiho (2022). The remaining 0.67 applies to surface-dwelling marine taxa. Equations (15) and (16) are used for modeling extinction scenarios during Events 5–16.

SAR$_D$ is influenced by both temperature and dissolved oxygen levels. Elevated surface temperatures reduce oxygen concentrations in deep water, a process linked to deep-sea extinctions during the end-Permian and end-Cenomanian anoxia–euxinia events, despite elevated atmospheric O₂ (e.g., Sun et al., 2012; Kaiho et al., 2013, 2016a). High surface temperatures can cause extinction in both surface and deep-water taxa. Although deep water temperatures are lower than those at the surface, the greatest thermal anomalies occur in surface waters, while deep-water temperatures remain relatively constant throughout the water column. Consequently, SAR$_D$ is assumed to approximate SAR$_S$.

Thus, for Events 5–6, 8–16, and all non-events after Event 8 (excluding the interval between Events 9 and 10), SAR$_D$ is set equal to SAR$_S$, as these intervals are characterized by global surface temperatures comparable to or exceeding those of the end-Permian.

The dominant climate driver for mass extinction varies by event. Events –5 to 4 involve both warming and cooling phases, while Events 5–16 are exclusively warming-driven, corresponding to the yellow–orange shaded zone in Figure 1.

For Event 0 (the Anthropogenic Crisis), the maximum SRC values are set at 0.95 for insects, 0.70 for terrestrial tetrapods, and 0.90 for marine metazoans. These values reflect a worst-case scenario involving full-scale nuclear war, combined with moderate anthropogenic pollution, deforestation, and global warming (Kaiho, 2023). In the absence of nuclear conflict, SRC is assumed to be 1.0 for all groups. For Events 1–4 and 7, SRC values are 0.81 for insects, 0.63 for terrestrial tetrapods, and 0.74 for marine metazoans, based on average extinction percentages reported in Tables A3 and A4.

In events 5 and 6 where global average surface temperatures reach 38–39 °C in Figures 2a and 2c. The 38 °C and 39 °C correspond to GATES40 and 50 (SAR$_S$: 0.36 and 0.24) in Figures 2a and 2c, and GATEU00 and GATEU10 (SAR$_U$: 1.00 and 0.83) in Figure 2b. The both temperatures are lower than GATED showing 48°C (Fig 2d). Using equations 15 and 16, the SRC values are:

In Events 5 and 6, global mean surface temperatures reach 38–39 °C (Figures 2a, 2c). These correspond to GATES40 and GATES50 (SARS: 0.36 and 0.24), and to GATEU00 and GATEU10 (SARU: 1.00 and 0.83) in Figure 2b. Both values are below the stable GATED threshold of 48 °C (Figure 2d). Applying equations (15) and (16):

Event 5:

$$SRC_T = 0.95 \times 0.36 + 0.05 \times 1.00 = 0.39$$
$$SRC_M = 0.67 \times 0.36 + 0.33 \times 1.00 = 0.57$$

Event 6:

$$SRC_T = 0.95 \times 0.24 + 0.05 \times 0.83 = 0.27$$
$$SRC_M = 0.67 \times 0.24 + 0.33 \times 1.00 = 0.49$$

These values are listed in Table 2.

In Events 8–10, global surface temperatures rise to 43–45 °C (Figures 2a, 2c). At 43 °C, GATES90 is reached, resulting in complete extinction of surface-dwelling metazoans (SARS = 0). The corresponding

SARU values, 0.50 and 0.30, are based on GATEU30 and GATEU45 (Figure 2b). GATED remains at 48 °C, so SARD = 0. Using equations (15) and (16):

Event 8:

$$SRC_T = 0.95 \times 0 + 0.05 \times 0.50 = 0.025$$

Events 9 and 10:

$$SRC_T = 0.95 \times 0 + 0.05 \times 0.30 = 0.015$$

Events 8–10:

$$SRC_M = 0.67 \times 0 + 0.33 \times 0 = 0$$

These SRC values are also summarized in Table 2.

**2.7.2 Food Scarcity Rate (FSR)**

In addition to direct climatic impacts, food scarcity significantly contributes to extinction risk. As plants and primary producers collapse, only organisms capable of surviving on bacterial biomass or sedimentary organic matter—along with their predators—will remain. Consequently, an additional decline in survival rate, quantified as the Food Scarcity Rate (FSR), is expected during abrupt extinction events such as Events 8–10.

These events are characterized by the extinction of superterranean and surface-water (SS) metazoans and severe reductions in primary productivity due to sunlight loss—conditions common in major mass extinction scenarios (see Fig. 1). A low FSR value of 0.01 reflects survival through alternative nutritional pathways, including hydrothermal vent ecosystems and bacterial-based underground food sources (Cosson and Soldati, 2008; Miroshnichenko, 2004; Kelley et al., 2005).

Conversely, a high FSR value of 0.1 represents an optimistic estimate, assuming evolutionary adaptation of primary producers to extreme temperatures, allowing some limited ecosystem function to persist. This range (0.01–0.1) is applied to adjust survival estimates in scenarios where abrupt collapse of food webs occurs due to light inhibition and temperature stress.

**2.7.3   Recovery Rate by Warming (RRW)**

The Recovery Rate by gradual Warming (RRW), applied outside of abrupt extinction events, is calculated using the same structure as Equations (15) and (16), but based on temperature anomalies associated with gradual climate changes. These temperature anomalies are derived from Figure 2.

The RRW is calculated separately for terrestrial and marine metazoans using the following equations:

$$RRW_T = 0.95 \times SAR_S + 0.05 \times SAR_U \tag{17}$$

$$RRW_M = 0.67 \times SAR_S + 0.33 \times SAR_D \tag{18}$$

Here, $RRW_T$ and $RRW_M$ denote the recovery rates for terrestrial and marine metazoans, respectively. The coefficients reflect the relative contributions of surface and subsurface habitats, consistent with SRC calculations in earlier sections.

**2.7.4 Recovery Rate by Oxygen decline (RRO)**

The Recovery Rate by Oxygen decline (RRO) is calculated separately for terrestrial and marine metazoans, reflecting projected atmospheric $O_2$ levels over the next 1.0 Gyr. The RRO varies across time intervals and is defined by the following equations:

For terrestrial metazoans ($RRO_T$):

$$\text{During the next 0.5 Gyr: } RRO_T = 1.0 \tag{19}$$

$$\text{During the next 0.5–0.8 Gyr: } RRO_T = 2.67 - 3.33T \tag{20}$$

$$\text{During the next 0.8–1.0 Gyr: } RRO_T = 0.0 \tag{21}$$

For marine metazoans ($RRO_M$):

$$\text{During the next 0.5 Gyr: } RRO_M = 1.0 \tag{22}$$

$$\text{During the next 0.5–0.8 Gyr: } RRO_M = 2 - 2T \tag{23}$$

$$\text{During the next 0.8–1.0 Gyr: } RRO_M = 1.84 - 1.80T \tag{24}$$

In these equations, $T$ is the numerical time variable in Gyr, ranging from 0.5 to 1.0. $RRO_T$ and $RRO_M$ represent the recovery rates for terrestrial and marine metazoans, respectively. These values are derived from oxygen level projections by Ozaki and Reinhard (2021).

Atmospheric oxygen levels are projected to steadily decline over the next 1.1 Gyr. Beginning at approximately 1.0 PAL around –0.1 Gyr (i.e., present time), $O_2$ levels are expected to drop to ~0.5 PAL (median: 0.3–0.7) by 0.5 Gyr. This decline continues to ~0.3 PAL (median: 0.07–0.5) at 0.8 Gyr, followed by a rapid collapse to ~0.01 PAL between 1.0 and 1.1 Gyr.

These projections inform the modeled RRO values and are used in conjunction with observed relationships between atmospheric $O_2$ levels and metazoan/plant diversity (Fig. 3) to assess the impact of oxygen depletion on biodiversity trajectories throughout Earth's future.

**2.7.5 Recovery Rate by Primary productivity (RRP)**

A molecular-level investigation of a $C_3$ plant's response to low $CO_2$ concentrations (100 ppm compared to the typical 380 ppm) revealed that reduced $CO_2$ levels lead to a significant decline in biomass productivity (Li et al., 2014). In the future, such low $CO_2$ conditions are projected to occur at approximately 0.5 Gyr (ranging from 0.4 to 0.65 Gyr), despite rising surface temperatures driven by increasing solar luminosity. This decline in atmospheric $CO_2$ is expected to cause a gradual reduction in net primary productivity (NPP), ultimately contributing to long-term decreases in metazoan diversity.

Approximately 40 million years after the extinction of $C_3$ plants, $C_4$ plants are expected to evolve into tree-like forms. This evolutionary transition is anticipated to support the diversification of metazoans that rely on such vegetation, paralleling the Devonian rise of terrestrial plant ecosystems. These recovering metazoan groups are expected to originate from species formerly dependent on $C_3$ plants. The Recovery Rate by primary Productivity (RRP) in this context is estimated to range from 0.3 to 0.8, reflecting the evolutionary and ecological challenges in re-establishing complex, tree-supported food

webs from $C_4$ vegetation. This recovery event is centered at 0.4 Gyr, based on a temporal uncertainty of ±0.2 Gyr (Table 2).

Although oceanic primary producers are predominantly phytoplankton, both $C_3$ and $C_4$ photosynthetic pathways coexist in marine environments (Reinfelder et al., 2000, 2004). To approximate the effect of terrestrial plant crises on marine metazoan diversity, equivalent reduction and recovery values are provisionally applied to marine systems, mirroring those used for terrestrial tetrapods under two modeled scenarios.

For Events 8–10, which involve abrupt primary productivity collapse due to sunlight reduction, RRP is estimated between 0.1 and 0.3, representing an intermediate range between $C_3$ and $C_4$ plant crises.

The $C_4$ plant crisis is expected to occur at approximately 0.97 ± 0.2 Gyr, aligning closely with Event 11. At this stage and beyond, RRP values decline to between 0.01 and 0.1, as NPP is assumed to approach zero. Under such conditions, only UD metazoans—those subsisting on bacteria, detritus, or residing in deep-sea hydrothermal ecosystems—would persist (Cosson and Soldati, 2008; Miroshnichenko, 2004; Kelley et al., 2005). This sharp reduction in RRP reflects the critical dependence of most metazoans on photosynthetically sustained food webs.

**Specific comments**

• _L. 29: I would replace "all known forms of life" by "**almost** all known forms of life", e.g., tar-digrades can survive temperatures higher than 100°C.

Author Reply: Done

• _L. 32: Can you shortly explain what $C_3$ and $C_4$ plants are?

Author Reply: Revised to "The reduction in $CO_2$ will affect the photosynthesis of two major plant groups—$C_3$ and $C_4$ plants—differently, due to their distinct photosynthetic pathways. $C_4$ plants generally exhibit higher photosynthetic efficiency under hot and dry conditions, making them more resilient to low $CO_2$ levels than $C_3$ plants."

• _L. 52-53: This sentence disrupts the flow of the text. Since the corresponding information was just mentioned a few paragraphs earlier, the sentence is not necessary in my opinion.

Author Reply: Removed the sentence:

• _L. 95: "These data are applied in section 2.2." – The current section is 2.2, so I do not understand this sentence.

Author Reply: Revised to section 2.5.2.

• _Section 2.3: Are only records of marine metazoans available? If yes, the possible impacts of this limitation should be discussed.

Author Reply: Added "and terrestrial".

• _L. 97: What exactly is meant by "diversity rates"?

Author Reply: Revised to "Projections of future atmospheric oxygen levels and marine and terrestrial metazoan diversity were informed by records from the Paleozoic biodiversity maximum.".

• _L. 98: What is PAL?

Author Reply: The phrase "present atmospheric levels (PAL)" has already written in the first paragraph of Introduction section.

• _L 106-107: I think you mean that oxygen levels drop in the habitats of metazoans and not in metazoans themselves, right?

Author Reply: Removed ", with significant declines occurring when oxygen levels drop in both marine and terrestrial metazoans"

• _L. 108: Why do terrestrial metazoans require an ozone layer for evolutionary adaptation? (This is explained in l. 491-493, but I would already explain it here).

Author Reply: Revised the sentence to: Moreover, the emergence of terrestrial metazoans likely required elevated oxygen levels due to their dependence on a protective ozone layer that shields the surface from harmful short-wavelength ultraviolet radiation. These insights form the basis for further discussion in Section 2.4.

• _L. 110-115: What were the main reasons for mass extinction during these events? Author Reply: Added the following sentence: The primary drivers of these mass extinctions were large-scale volcanic events, with the notable exception of the end-Cretaceous extinction, which was caused by a meteoroid impact (Kaiho, 2025).

• _L. 131: There is no red curve in Fig. 1. Do you mean the orange curve?

Author Reply: Yes, I revised it to orange.

• _L. 147: Which were the five largest mass extinction events?

Author Reply: Revised to "five major mass extinctions" in line 169. Five major mass extinctions were defined in Introduction section.

• _L. 152: I thought ΔTec was estimated using SST data as stated in the previous section?

Author Reply: Yes, it was.

• _L. 163: What is sill?

Author Reply: Added "(a tabular sheet intrusion from magma that has intruded between older layers of sedimentary rocks)" in line 185.

• _L. 167-168: How are long-term changes in CO2 and SO2 emissions related to short-term temperature anomalies?

Author Reply: This sentence does not refer to long-term trends, but rather to abrupt changes that trigger major mass extinctions. I applied a gradual decrease in mantle temperature to estimate future $SO_2$ and $CO_2$ emission rates relative to present-day levels. These rates were then used to calculate extinction percentages.

• _L. 191: Why do you use regions with oceanic climate?

Author Reply: Revised to "The model presented in Figure 2a estimates extinction thresholds for St metazoans (i.e., surface-dwelling terrestrial animals) based on oceanic climate regions on land, as these regions generally experience milder climates compared to continental interiors."

• _L. 196: Does a gradient of 15°C only apply to warm conditions or why do you explicitly mention warm conditions here?

Author Reply: Revised to "Baseline warm climate gradient: Oblique lines with a 15 °C gradient from equator to pole are drawn to represent warm Earth conditions (e.g., the late Paleocene–early Eocene and mid-Cretaceous hothouse periods), which are relevant for modeling extinction under future warming scenarios (Zhang et al., 2019; Burgener et al., 2023). These lines intersect the 46 °C tolerance points (thick oblique lines in Figure 2a)."

• _L. 199: Where exactly do the 5°C come from?

Author Reply: This adjustment uses modern reference data from warm coastal cities such as Shanghai (~30°N) and Singapore (~0°N) (Weather Spark, https://weatherspark.com) (dashed oblique lines in Figure 2a).

• _L. 201: Same as l. 191 and l. 196: Why do you use data from warm coastal cities? Author Reply: Revised to "This adjustment uses modern reference data from warm coastal cities such as Shanghai (~30°N) and Singapore (~0°N) (Weather Spark, https://weatherspark.com), as indicated by the dashed oblique lines in Figure 2a, to determine the extinction threshold."

• _L. 207: There is no section 2.3.3. Do you mean 2.6.3?

Author Reply: Section 2.6.3 in line 267.

• _Equation 11: What is ΔLT? (LT is only defined in the caption of Fig. 2)

Author Reply: Revised to "where LT (Local Temperature) is the latitude-dependent adjustment from LMMT to LAT." in line 260.

• _Sect. 2.7.1: This section is quite hard to follow since many abbreviations are used. Maybe it would help to spell out the abbreviations from time to time.

Author Reply: Revised this section. Also revised the title to Survival Rate associated with Climate change (SRC).

• _L. 244-248: I think it should already be mentioned here that different scenarios are analyzed.

Author Reply: Removed "Before mass extinction events, SRC events values follow gradual warming trends as described in Table 2.".

• _L. 249-252: I think it would be helpful to provide a brief description of the different events. Some description is given in Sect. 3.2, but I believe that including such a description earlier on would give the reader a better understanding.

Author Reply: The different SRC in events 5 and 6 and event 7 is due to a long-term greenhouse period and a long-term ice house period, respectively. Added the calculation in lines 343-347.

• _L. 252: What exactly is the survival area rate and what is the difference to the survival rate?

Author Reply: SAR is rate of land area where metazoans survive in all land area ($km^2/km^2$). Added "Survival Area Rate (SAR) is rate of land and ocean area where metazoans survive in all land and ocean area ($km^2/km^2$). When extinction occurred in 0–10, 0–20, 0–30, 0–40, 0–50, 0–60, 0–70, 0–80, and 0–90° latitudes by warming, SAR values are defined as 0.83, 0.66, 0.50, 0.36, 0.24, 0.14, 0.06, 0.02, 0.00, respectively, under the same rate of land and ocean in those latitudes. The rates SAR are decided by only temperature 46 °C using Figure 2." in line 302-305.

• _L. 256-257: What exactly do the rates StR, SwR, UR, and DR describe?

Author Reply: I wrote coefficient 0.95, 0.67, 0.05, 0.33 instead of these abbreviation in these equations (15, 16) in lines 309-310. The following paragraph explains these coefficients.

• _L. 259: Could you explain more clearly how the SAR is calculated?

Author Reply: Added "Survival Area Rate (SAR) is rate of land and ocean area where metazoans survive in all land and ocean area ($km^2/km^2$). When extinction occurred in 0–10, 0–20, 0–30, 0–40, 0–50, 0–60, 0–70, 0–80, and 0–90° latitudes by warming, SAR values are defined as 0.83, 0.66, 0.50, 0.36, 0.24, 0.14, 0.06, 0.02, 0.00, respectively, under the same rate of land and ocean in those latitudes. The rates SAR are decided by only temperature 46 °C using Figure 2." in line 302-305.

• _L. 263: I cannot follow the argumentation here. Why should SARD approximate SARS?

Author Reply: Added "High surface temperatures can cause extinction in both surface and deep-water taxa. Although deep water temperatures are lower than those at the surface, the greatest thermal anomalies occur in surface waters, while deep-water temperatures remain relatively constant throughout the water column." in lines 320-323.

• _L. 268: How did you determine the impact of food scarcity on survival rates?

Author Reply: Revised to "FSR is the Food Scarcity Rate, reflecting the impact of the collapse of plants and primary producers." in lines 295-296.

Added the following explanation in lines 364-371.

These events are characterized by the extinction of superterranean and surface-water (SS) metazoans and severe reductions in primary productivity due to sunlight loss—conditions common in major mass extinction scenarios (see Fig. 1). A low FSR value of 0.01 reflects survival through alternative nutritional pathways, including hydrothermal vent ecosystems and bacterial-based underground food sources (Cosson and Soldati, 2008; Miroshnichenko, 2004; Kelley et al., 2005).

Conversely, a high FSR value of 0.1 represents an optimistic estimate, assuming evolutionary adaptation of primary producers to extreme temperatures, allowing some limited ecosystem function to persist. This range (0.01–0.1) is applied to adjust survival estimates in

scenarios where abrupt collapse of food webs occurs due to light inhibition and temperature stress.

• _L. 275: And the other events?

Author Reply: Added the followings in lines 335-358.

In events 5 and 6 where global average surface temperatures reach 38–39 °C in Figures 2a and 2c. The 38 °C and 39 °C correspond to GATES40 and 50 (SAR$_S$: 0.36 and 0.24) in Figures 2a and 2c, and GATEU00 and GATEU10 (SAR$_U$: 1.00 and 0.83) in Figure 2b. The both temperatures are lower than GATED showing 48°C (Fig 2d). Using equations 15 and 16, the SRC values are:

In Events 5 and 6, global mean surface temperatures reach 38–39 °C (Figures 2a, 2c). These correspond to GATES40 and GATES50 (SARS: 0.36 and 0.24), and to GATEU00 and GATEU10 (SARU: 1.00 and 0.83) in Figure 2b. Both values are below the stable GATED threshold of 48 °C (Figure 2d). Applying equations (15) and (16):

Event 5:

$SRC_T = 0.95 \times 0.36 + 0.05 \times 1.00 = 0.39$

$SRC_M = 0.67 \times 0.36 + 0.33 \times 1.00 = 0.57$

Event 6:

$SRC_T = 0.95 \times 0.24 + 0.05 \times 0.83 = 0.27$

$SRC_M = 0.67 \times 0.24 + 0.33 \times 1.00 = 0.49$

These values are listed in Table 2.

In Events 8–10, global surface temperatures rise to 43–45 °C (Figures 2a, 2c). At 43 °C, GATES90 is reached, resulting in complete extinction of surface-dwelling metazoans (SARS = 0). The corresponding SARU values, 0.50 and 0.30, are based on GATEU30 and GATEU45 (Figure 2b). GATED remains at 48 °C, so SARD = 0. Using equations (15) and (16):

Event 8:

$SRC_T = 0.95 \times 0 + 0.05 \times 0.50 = 0.025$

Events 9 and 10:

$SRC_T = 0.95 \times 0 + 0.05 \times 0.30 = 0.015$

Events 8–10:

$SRC_M = 0.67 \times 0 + 0.33 \times 0 = 0$

These SRC values are also summarized in Table 2.

• _L. 285-286: Is this reasonable? The limitations of this assumption should be discussed.

Author Reply: Revised to "Although oceanic primary producers are predominantly phytoplankton, both $C_3$ and $C_4$ photosynthetic pathways coexist in marine environments (Reinfelder et al., 2000, 2004)." in lines 414-415.

• _L. 287-293: I cannot follow here. Why are metazoans extinct at 0.97 Gyr if 2% remain? And

don't you state in other parts of the manuscript (e.g., the Abstract, l. 460, and the Conclusions) that according to your calculations, metazoans go extinct at 0.7 and not 0.97 Gyr? Author Reply: Revised to "The $C_4$ plant crisis is expected to occur at approximately $0.97 \pm 0.2$ Gyr, aligning closely with Event 11. At this stage and beyond, RRP values decline to between 0.01 and 0.1, as NPP is assumed to approach zero. Under such conditions, only UD metazoans— those subsisting on bacteria, detritus, or residing in deep-sea hydrothermal ecosystems—would persist (Cosson and Soldati, 2008; Miroshnichenko, 2004; Kelley et al., 2005). This sharp reduction in RRP reflects the critical dependence of most metazoans on photosynthetically sustained food webs." in lines 420-424.

• _L. 304: What exactly is numerical age?

Author Reply: Revised to "In these equations, T is the numerical time variable in Gyr, ranging from 0.5 to 1.0."

• _Sect. 3.1: I think this description would have been more helpful in the Methods section some-where between Sects. 2.5 and 2.7. Then the reader could better understand the different survival rates etc.

Author Reply: Moved to section 2.5.5.

• _L. 364: What exactly do you mean by abrupt climate events? Are you referring to volcanic eruptions and meteorite impacts? If yes, I recommend stating this here again.

Author Reply: Revised to "Climate Phases B–E are likely to be initiated by abrupt climate disturbances caused by large volcanic eruptions or meteorite impacts occurring during greenhouse intervals." In lines 214-215."

• _L. 385-388: Could you also give current numbers for comparison?

Author Reply: Revised to "In both the Min and Max Cases, during Events 1–4 (occurring between 0.03 and 0.40 Gyr into the future), biodiversity will decline sharply from current levels of 610 insect families, 315 terrestrial tetrapod families, and 950 marine metazoan families to 494, 198, and 703 families, respectively. These losses are followed by full recovery to the present number of families (Fig. 5).

In the NC Min Case, Event 0 (the Anthropogenic Crisis) results in reduced survival, with 580 insect families, 221 terrestrial tetrapod families, and 855 marine metazoan families remaining, followed by recovery to current diversity levels (Fig. 5).

However, if the recovery rate after Event 0 is zero—as modeled in the NCC Min Case— biodiversity continues to decline due to sustained anthropogenic impacts on the biosphere. In this scenario, even during non-extinction intervals, the surviving numbers of families are projected to remain at 580 for insects, 221 for terrestrial tetrapods, and 855 for marine metazoans (Fig. 6). The NCC Max Case also results in equivalent proportional reductions compared to the Max Case.""

- _L. 592: Are you sure that your study is the first to reveal that?

Author Reply: Removed "is the first to".

- _Fig. 1:

o I do not understand what the green open diamond symbols denote exactly. Can you maybe explain again in other words?

Author Reply: Revised to "Green open diamond symbols denote temperatures before mass extinctions."

o Would it be possible to add some sort of legend to the atmospheric oxygen level graph that specifies the impact on metazoans?

Author Reply: Added "Impact on metazoans" in Figure 1.

- Fig. 2 (l. 630): Not only silhouettes of terrestrial plants are shown, so maybe write "silhouettes of metazoans and terrestrial plants"?

Author Reply: Revised to "Silhouettes of metazoans indicate their approximate diversity and corresponding oxygen levels, whereas silhouettes of terrestrial plants indicate only corresponding oxygen levels."

- Fig. 3: The yellowish-green is hard to distinguish from the green, so maybe use a different color?

Author Reply: Revised it to yellow.

- Table A2: What is SD? What does aftermath warming mean?

Author Reply: Revised SD to standard deviation. Revised Aftermath warming to Warming.

- _L. 720 and 725: before or after?

Author Reply: Revised to "Underlined and double underlined numbers indicate values corresponding to before and after extinction events, respectively."

- _L. 727-728: Why do the underlined numbers occur just before major mass extinction events if they represent periods of recovery? Something seems wrong here.

Author Reply: No, the underlined numbers occur just after major mass extinction events.

**Technical corrections**

- _L. 43: have  triggered historical mass extinctions

Author Reply: Revised to "have triggered past mass extinctions"

- _L. 61: This manuscript was written by only one author, right? I would use "I" instead of "we"; there are other occurrences throughout the manuscript.

Author Reply: Revised to "I" for all.

- _L. 89 and 96: I think it should be "Past records of"

Done

• _L. 204: LMMT

Done

• _L. 213: 2.5 m

Done

• _L. 222: GATES values are  equal

Done

• _L. 237: where $D_t$ represents

Done

• _L. 324: 1 m

Done

• _L. 378: "However" does not seem to fit here.

Deleted "However".

• _L. 398: primary productivity?

Yes, done

• _L. 430: estimating  the future diversity

Done

• _L. 504: illustrate

Done

• _L. 569-573: This is a repetition of l. 566-569 and should be deleted.

Done

• _Fig. 2:

o  Atmospheric oxygen level relative to  present atmospheric level

o  Diversity rate relative to  the Paleozoic maximum

Done

• _L. 659 and 729: the Methods section

Done

• _L. 676: The capitalization in this sentence seems odd.

Done

• _L. 686: rates in the future

Done

• _L. 687-688:  compared to terrestrial plants

Done

• _Table 2:  at the family level

Done

• _Table A2: Earth's average surface temperature  including the long-term trend, long-term

cy-cle, and short-term events with temperature anomalies and decreasing CO2 and SO2 emissions due to the decrease in mantle potential temperature during major mass extinction events from 0.7 billion years (Gyr) before the present to 1.5 Gyr into the future

Done

---

## Author Comment (AC2)

**Author replies for reviewer comments for MS No.: egusphere-2025-1853**

Title: Future diversity and lifespan of metazoans under global warming and oxygen depletion

Author(s): Kunio Kaiho

MS No.: egusphere-2025-1853

MS type: Research article

1. Author replies for comments from Anonymous Referee #2

2. Author replies for comments from Anonymous Referee #1

This document contains a point by point reply to the issues raised by the reviewers (in red); changes made in the MS are indicated in blue. All original comments by the reviewers are left in black font.

**1. Author replies for comments from Anonymous Referee #2**

**General comments**

This manuscript presents a novel and comprehensive model that predicts the lifespan of metazoans on Earth over the next 1.5 billion years. It combines a wide range of geological, climate, and biological data to create a compelling narrative about the long-term future of complex life. The main conclusion—that metazoans will go extinct in approximately 700 million years, much earlier than previous estimates—is a significant and provocative contribution. However, given the immense timescale and the complexity of the integrated model, it is essential to carefully present assumptions, uncertainties, and limitations to address skeptical readers convincingly. This is a potentially high-impact manuscript that aligns well with the scope of this journal. Its bold projections are its main strength. To maximize its impact and increase the chances of publication, the authors need to strengthen the presentation regarding the treatment of uncertainty and provide more robust justifications for the key parameters that drive the model. By doing so, they can turn a compelling thought experiment into a foundational and highly cited piece of future Earth system science. I put specific comments about each section below.

**Specific comments**

**[introduction]**

While the author currently identifies several individual gaps, such as cyclical rhythms "(L. 40), have not yet been fully incorporated, and abrupt events "(L. 46) have not yet been factored in," this can feel somewhat fragmented; they could be woven together to create a single, compelling argument for why your study is necessary. The author can reframe the problem to highlight the interaction of

multitimescale forcings as the central, unexplored challenge. I recommend adding a concise, overarching problem statement just before your final thesis paragraph (L. 48). Moreover, the final paragraph should then directly answer this problem statement. The list of seven critical factors is comprehensive, but the paragraph's impact will be greater if it highlights the integrative model itself as the core novelty.

The transitions between the main ideas, such as long-term trends, cycles, and abrupt events, can be made smoother, and the link between physical forcings and biological impacts can be clarified. A key conceptual point is the difference between predictable cycles and unpredictable events. Strengthen this transition by adding a sentence that clearly contrasts their timescales. For example, at the end of the cycles paragraph: "...have not yet been fully incorporated into projections of future surface temperatures. Beyond these predictable, multi-million-year cycles, Earth's climate is also punctuated by unpredictable, abrupt events..." Then, the author briefly explains how physical drivers influence biological outcomes modeled in the system. Regarding Tectonic Cycles, the author adds a phrase on how these cycles might influence plant crises or metazoan survival (e.g., through changes in continental configuration that affect weathering rates or create or eliminate refugia). Additionally, regarding abrupt events, the author can include a sentence explaining that these events are modeled as drivers of "step-changes" in biodiversity, which can reset recovery trajectories.

The author can enhance clarity and scientific rigor by refining specific sentences. The sentence "Global warming will accelerate terrestrial weathering..." serves as a key link. The author should add a few words to clarify how the mechanism works, which would add depth. In the final paragraph, the sentence "Projections are based on temperature modeling, thermal tolerance limits..." appears to preview the Methods section. Therefore, the author can rephrase it as, "Our model projections combine future scenarios of temperature and oxygen levels with established data on metazoan thermal tolerance and diversity trends."

The final two paragraphs contain some repetitive information and could be merged to create a more powerful and concise conclusion to the introduction. The author can integrate the description of metazoan evolution and thermal tolerance into the core thesis paragraph. This creates a single, strong paragraph that states what you did, what you based it on, and the scope of your analysis.

Author reply: In response to the comments regarding the introduction section, I revised the opening

to begin with: "Metazoans, which first appeared and diversified around 700–500 million years ago, evolved from simple cnidarian-like organisms into more complex forms such as arthropods and vertebrates during an oxygen increase associated with climate change from icehouse to greenhouse states (Erwin, 2015; Kaiho et al., 2024). Following their transition to land around 400 million years ago, metazoans eventually gave rise to humans (genus Homo) approximately 3 million years ago. However, the trajectory of future metazoan diversity remains unclear."

I then summarized previous studies projecting future life and biodiversity on Earth, explained the underlying causes and mechanisms influencing long-term biodiversity trends, and finally added the following new paragraphs to clarify how these factors are addressed in the present study.

Despite their significance, both cyclical and abrupt events remain largely absent from existing future climate models, and the long-term trajectory of biodiversity on Earth remains poorly constrained. In this study, these cyclic and abrupt events are incorporated as drivers producing "step changes" in biodiversity to evaluate how long animal life may continue to persist on Earth.

This study evaluates extinction dynamics across superterranean, surface-water, subterranean, and deep-sea habitats. Our model projections integrate future scenarios of temperature and oxygen levels with established data on metazoan thermal tolerance and diversity trends. The model incorporates seven key factors—anthropogenic crises, long-term warming, cyclical climate rhythms, abrupt events, C3 plant collapse, C4 plant decline, and oxygen depletion—to project metazoan extinction over the next 1.5 billion years (Gyr).

**Methods:**

**[Methods: 2.1]**

The statement "Assuming that the icehouse-greenhouse cycle and major mass extinctions continue at the same pace as in the past" (L. 66) is a significant assumption that is central to your model. This requires a brief justification. Is there a reference supporting the consistency of these cycles over billionyear timescales? A sentence citing relevant geological timescale studies would greatly strengthen this.

Author reply: Revised to: "To incorporate these cycles, I applied a temperature anomaly of 8 °C between icehouse and greenhouse intervals, based on Phanerozoic temperature reconstructions by Scotese et al. (2021), to the long-term thermal evolution model of Mello and Friaça (2019). However, this temperature anomaly is expected to gradually decrease as atmospheric CO2 declines due to enhanced continental weathering. This follows the well-known climate model result that a doubling of CO2 produces a 2.3 °C increase in surface temperature (Manabe, 1996)."

The revised orange curve in Figure 2 reflects a reduced temperature anomaly, shifting the projected timing of metazoan disappearance from ~0.7 Gyr to ~0.9 Gyr.

**Figure 2.** Global average surface temperature and atmospheric CO2 and oxygen levels over the past and future 2.5 Gyr.

The method assumes that the relationships between temperature/ $O_2$  and biodiversity observed in the deep past will also apply to the entire future of complex life. This core assumption should be clearly acknowledged as a potential limitation or justified with a solid rationale.

Author reply: Please see the revised paragraph provided in the response below.

Although the author mentions specific groups later, the term "metazoan extinction" (L. 72) in point 3

is quite broad. It would be helpful to clarify early on that your study focuses on the specific groups listed—marine metazoans, terrestrial tetrapods, and insects —as proxies for overall metazoan diversity.

Author reply: Added "insects, terrestrial tetrapods, and marine metazoans" in the final sentence in the introduction section.

The nature of these abrupt large-scale future climate events (Events 1-16; L. 81 and 83) remains unclear. Are they modeled as analogues to the Big Five? Are they stochastic events? A brief explanation of how these were defined and selected would be very helpful.

Author reply: Added (Events 1–16, established as analogs to the "Big Five" mass extinctions).

The phrase "framework that includes temperature trends, oxygen levels, and C3-C4 plant crises" (L. 78-79) is too general. The author needs to describe the actual model. Is it a statistical correlation? A dynamic system model? A set of conditional rules? Then, C3-C4 plant crises (L. 79) are mentioned but are not introduced earlier. The author should briefly explain what this crisis involves and why it is a factor in your model.

Author reply: Removed the sentence, which is due to the recommended simplified first paragraph shown below. This part corresponds to "5. Integrated Future Diversity Model: -- corresponds to subsection 2.6". The subsection explains the model.

I explain C3–C4 plant crises in the introduction section "The reduction in  $CO_2$  will differentially affect the two major plant groups— $C_3$  and  $C_4$  plants—because of their distinct photosynthetic pathways.  $C_4$  plants generally exhibit higher photosynthetic efficiency under hot and dry conditions, making them more resilient to low  $CO_2$  levels than  $C_3$  plants. Consequently,  $C_3$  plants, which include most trees, are expected to decline first ( $C_3$  plant crisis), followed by a  $C_4$  plant crisis, ultimately leading to a collapse in metazoan diversity (Reinfelder et al., 2000, 2004; Mello and Friaça, 2019; Ozaki and Reinhard, 2021)."

Regarding methodological precision, the author used "compiled records (L. 63)" and "analyzed the relationship (L. 64)," but should specify the particular statistical methods, such as correlation analysis and regression modeling. At point 3 (L. 69~), the transition from "local extinction temperature" to "global average surface temperature" at which extinction occurs is a crucial scaling step. This process should be clearly explained in the detailed methods (2.6), and the summary should hint at its complexity (e.g., "...was scaled to a global average surface temperature using latitudinal gradients"). Finally, in the last paragraph (L. 85~), the sentences about gradual extinctions (orange curve) and the estimated durations (0 Gyr and 0.05 Gyr) seem somewhat out of place in the method summary. They are better suited for the Results section or a dedicated part of

the detailed methods. The "0 Gyr" duration for temperature anomalies is confusing and needs clarification.

Author reply: Explained the methods for L. 63 and 64 in the section 2.2 Past climate and diversity baselines.

Added "Local extinction temperatures were scaled to a global average surface temperature using latitudinal gradients.".

Revised to "The estimated durations of temperature anomalies-extinction and recovery events are 0.00 Gyr (years to less 0.1 m.y.) and 0.05 Gyr, respectively, based on geological data (Erwin et al., 1987)." in the new section 2.3.4 Events.

**Revised first paragraph:**

To project the future lifespan of metazoans on Earth, we developed a multi-step model that integrates relationships derived from past climate and biodiversity dynamics. Our core assumption is that the pacing of icehouse-greenhouse cycles (~0.3 Gyr) and major mass extinctions (~0.094 Gyr), along with the physiological constraints on metazoans, will remain consistent in the future [references, if possible]. The analysis proceeded as follows:

- 1. Past Climate and Diversity Baselines: -- correspond to subsection 2.2
- 2. Future Temperature Projections: -- correspond to subsection 2.3
- 3. Metazoan Thermal Tolerance Limits: -- correspond to subsection 2.4
- 4. Oxygen-Biodiversity Relationship: -- corresponds to subsection 2.5
- 5. Integrated Future Diversity Model: -- corresponds to subsection 2.6

The paragraph about five past events, 16 events, and gradual extinctions (L. 81-88) could be moved or integrated into points 2 and 5 above for better flow.

Author reply: Simplified the first paragraph: To project the future lifespan of metazoans on Earth, I developed a multi-step model that integrates relationships derived from past climate patterns and biodiversity dynamics (Fig. 1). The core assumption is that the amplitude of the icehouse—greenhouse cycles—occurring at 0.35–0.30 Gyr intervals (Scotese et al., 2021; Torsvik et al., 2024; Vérard, 2024)—will gradually decrease due to declining atmospheric CO2 driven by continental weathering. In contrast, major abrupt climatic perturbations leading to mass extinctions will continue to recur at approximately 0.094 Gyr intervals, based on the mean ages reported by Kaiho (2025).

Abrupt events are assumed to be independent of long-term trends, as their short duration prevents significant modulation by gradual climate evolution. Together with physiological constraints on metazoans, these cyclic and abrupt processes are expected to remain consistent in the future. This framework accounts for the projected long-term decline in mantle potential temperature (see

Section 2.3.3), since mantle degassing governs both climatic cyclicity and large-scale extinction events.

An additional assumption is that progressive CO2 drawdown, intensified by enhanced continental weathering, will trigger recurrent plant crises, whereas the concurrent decrease in atmospheric oxygen will ultimately lead to the extinction of metazoan life.

The analysis proceeded as follows:

- 1. Past climate and diversity baselines corresponds to subsection 2.2
- 2. **Future temperature projections** corresponds to subsection 2.3
- Metazoan thermal tolerance limits local extinction temperatures scaled to global mean surface temperature using latitudinal gradients – corresponds to subsection 2.4
- 4. Oxygen-biodiversity relationship corresponds to subsection 2.5
- Integrated future diversity model incorporating components 1–4, atmospheric CO2 trends, and food scarcity effects corresponds to subsection 2.6

To maintain consistency across subsections, the author should verify that the titles of subsections 2.2 through 2.6 directly correspond to points 1 through 5 in the Method summary and expand on them. For example, 2.2 should include a detailed methodology for point 1, "Past Records." Therefore, if possible, the author should either revise the outline based on your Method summary or revise the Method summary itself.

Author reply: Revised these subsections to fit the above 1 to 5.

Added "To estimate future metazoan diversity, I compiled past records of global surface temperatures and biodiversity. Earth's climate history has exhibited a cyclical pattern of 0.35–0.30 Gyr since -1.0 Gyr (Fig. 2). This cycle consists of extended greenhouse phases lasting around 0.2 Gyr, followed by shorter icehouse phases of approximately 0.1 Gyr (Scotese et al., 2021; Torsvik et al., 2024; Vérard, 2024). To incorporate this climate cycle, I applied an 8°C temperature anomaly between icehouse and greenhouse periods (Bond and Grasby, 2017) to Mello and Friaça's model (Fig. 2, Table A1)." and "Climate cycles operate over long timescales, whereas mass extinction events occur over much shorter durations. Both processes contribute to rising temperatures and act to shorten the time required for the eventual extinction of metazoans in the future." in subsection 2.2.

The subsections (2.2 to 2.6 (or 2.7)) must flesh out the details summarized in 2.1. For each step, the author needs to specify: About data sources, the author mentions specific databases or publications, and provides tables (e.g., Tables A1 and A2).

Author reply: For 2.3 Future temperature projections, for example, I revised "A coherent

methodology for projecting future temperatures is illustrated in Figure 1 (highlighted in pale red and yellow). This figure shows that surface temperatures were calculated based on long-term warming trends, climatic cycles, and abrupt events (Events 1–16, established as analogs to the "Big Five" mass extinctions) using Equations 1–10 presented in this section. Mantle temperature is treated as a controlling factor for abrupt events (Equations 4–9). The same pacing of icehouse–greenhouse cycles (0.3 Gyr intervals) and major abrupt mass extinctions (0.0945 Gyr intervals) is applied to future temperature projections. The uncertainties associated with these cycles are used to estimate the timing of the complete extinction of metazoans, with error margins of approximately ±0.1 Gyr for icehouse–greenhouse cycles and ±0.03 Gyr (1 SD) for mass extinction timing." Done for the other subsections and data sources.

About analytical techniques, how did the author analyze the relationship? Was it a linear regression? A non-linear model? Specify the statistical tests and the software/tools used.

Author reply: Added "New datasets were generated using the equations and baseline data presented in subsection 2.2 (Tables 2, A1, A3, and A4). All calculations were performed in Microsoft Excel to produce Tables 2–3 and A1–A5. Detailed methodologies are provided in the Methods section and described in the corresponding table captions." In subsection 2.1.

About Quantitative Definitions, what are the numerical thresholds for "low, mid, and high latitudes"? Author reply: Revised to "in low to mid-latitudes (0°-30° to 30°-60°) to 35-40°C in high latitudes (60°-90°)" in line 195."

What defines a "C3-C4 plant crisis" in your model? Define these operationally.

Author reply: Revised to "---the C3 plant crisis defined by 0.15 mbar CO2 partial pressure (Lovelock and Whitfield 1982; Caldeira and Kasting 1992; Lenton and von Bloh 2001;---" and "---the C4 plant crisis defined by 0.01 mbar CO2 partial pressure (Caldeira and Kasting 1992; Franck et al. 2000a, b, 2006),---"

Model Parameters: The values "0.3 Gyr" and "0.094 Gyr" are key model inputs. Justify these choices with references beyond the general (Erwin et al., 1987). How sensitive are your results to these specific values?

Author reply: Added "Scotese et al., 2021; Torsvik et al., 2024; Vérard, 2024)" for 0.3 Gyr. Revised ~0.3 to 0.35–0.30 Gyr. Added data in the 0.25, 0.35, 0.40 Gyr cases to show that "metazoan extinction will be complete within 0.7–0.8 Gyr" in all cases in Supplementary Figures.

Added ": average of the ages from Kaiho, 2025" for 0.094.

**[Methods: 2.2]**

The issue with this section is the jump from describing the data to stating the objective. The sentence "To estimate future abrupt climate changes and biotic crises, we selected the five largest mass extinctions..." (L. 91-92) is an objective, not a method. Thus, the author should restructure the text to first present the data and then explain the analytical step.

Author reply: This section has been revised as shown above.

The author also needs to provide a more detailed explanation of how they selected and used these "five major mass extinctions." This clarifies the reason for choosing specific events and helps ensure the methodology can be reproduced.

Author reply: Revised to "I selected the five major mass extinctions marked by abrupt event and more than 35% marine genera extinctions corresponding to more than 60% marine species extinctions in Earth's history based on Bambach (2006) and Kaiho (2022). I used family-level extinction percentages during those major mass extinction events reflect severe biodivesity losses across multiple taxonomic groups.".

The author mentions two different types of data: 1) long-term climate cycles (icehouse-greenhouse) and 2) short-term temperature anomalies during extinction events, but the connection between them is not explained. Please briefly explain the role of each dataset in your overall model.

Author reply: Added "Abrupt events are assumed to be independent of long-term trends, as their short duration prevents significant modulation by gradual climate evolution." in subsection 2.1.

The final sentence (L. 95), "These data are applied in section 2.2," is redundant as it is already within section 2.2.

Author reply: Revised to "These data are applied in section 2.5.2.".

**[Methods: 2.3]**

The paragraph (L. 97-L. 104) states that "marine metazoan diversity rates" were "derived from past records," but it does not provide a source for these diversity rates, only for the oxygen levels. This creates a notable gap in reproducibility. The author must provide the reference(s) for your biodiversity data.

Author reply: I selected the five major mass extinctions marked by abrupt event and more than 35% marine genera extinctions corresponding to more than 60% marine species extinctions in Earth's history based on Bambach (2006) and Kaiho (2022).

What does a "diversity rate" of 0.04 or 1.0 actually mean? Is it a count of families or genera normalized to a maximum value? Is it an estimated measure of richness? Without a clear

operational definition, the metric remains ambiguous. Please provide a brief explanation of how the calculation is performed.

Author reply: Revised to "Historical data show that during the early Ediacaran (~0.6 Gyr ago), atmospheric O2 was around 0.01 PAL, with a marine metazoan diversity rate for Paleozoic biodiversity maximum in family level of 0.04 (Fig. 4; Erwin et al., 1987).". Moved to section 2.5.1.

The second paragraph (L. 105-108) is mostly interpretative. Phrases such as "suggests that there is a positive relationship," "significant declines," and "appear to require higher oxygen levels" are conclusions based on the data. Remove the entire second paragraph from the Methods section. These should be moved to your Results section (to show the relationship) and the Discussion section (to discuss the possible reasons).

Author reply: Revised to "Next section 2.5.2 uses the positive relationship between atmospheric oxygen levels and biodiversity, emphasizing that the emergence and diversification of terrestrial metazoans occurred during intervals of elevated oxygen concentration (Fig. 4). This relationship is applied to constrain future diversity projections based on predicted oxygen levels." Moved the interpretation to the discussion section.

The final sentence (L.108), "These interpretations are further applied in Section 2.4," should be rephrased to emphasize the data and relationship, not the interpretation. The author should rephrase it to something like, "The quantitative relationship between O2 and biodiversity established here (Figure 3) is used in Section 2.4 to describe the specific action, such as constraining future diversity projections based on predicted oxygen levels."

Author reply: Revisd to "Next section 2.5.2 uses the positive relationship between atmospheric oxygen levels and biodiversity, emphasizing that the emergence and diversification of terrestrial metazoans occurred during intervals of elevated oxygen concentration (Fig. 4). This relationship is applied to constrain future diversity projections based on predicted oxygen levels.", because this is needed here. Moved the interpretation on ozone to the discussion section: Figure 4 illustrates a positive relationship between oxygen levels and biodiversity. Moreover, the emergence of terrestrial metazoans likely required elevated oxygen concentrations, as their survival depended on the formation of a protective ozone layer that shields the surface from harmful short-wavelength ultraviolet radiation. This dependency may explain why terrestrial insects and tetrapods exhibit lower biodiversity compared to marine metazoans.

**[Methods: 2.4]**

Averaging percentages from different extinction events, which had vastly different causes and

selective pressures, into a single "future diversity loss" value is a highly simplified modeling choice. The author should justify why this is a valid approach for long-term projection. Please add a sentence or two explaining the rationale. The author should also acknowledge the limitations of this approach.

Author reply: Added "Averaged percentages from different extinction events were used only for diversity estimates in Events 1–4 and 7 as shown in Figures 5–7 and Table 2. These averaged values do not influence the main conclusions regarding the timing of the diversity decline (~4 Gyr from now) or the complete metazoan extinction (9–10 Gyr from now), because full recovery took place prior to the onset of the diversity decline phase. The principal causes of the diversity declines are attributed to the C3 plant crisis, decreased oxygen levels, and local temperatures exceeding thermal tolerance limits." In new subsection 2.3.4 Events.

The author lists five events for marine metazoans but only three for tetrapods and insects. Why is there a discrepancy? Is the author averaging five events for the marine group and three for the terrestrial group? This needs to be clarified.

Author reply: Added "Because terrestrial tetrapods and insects appeared shortly before the second mass extinction, no terrestrial diversity data exist for the first and second events. Therefore, average extinction percentages from the five major marine and three terrestrial events were used." The subsection is titled "Past and future extinction percentages and the interval," but the content does not mention any time interval. The 0.094 Gyr extinction cycle from your initial summary is a crucial parameter. This is where to define it. The author includes the methodology for the interval here.

Author reply: Revised to "Past climate and diversity baselines". Added to "Because terrestrial tetrapods and insects appeared shortly before the second mass extinction, no terrestrial diversity data exist for the first and second events. Therefore, average extinction percentages from the five major marine and three terrestrial events were used. The time interval for mass extinctions, 0.094 Gyr, corresponds to the mean ages of the five major extinctions reported by Kaiho (2025).".

**[Methods: 2.5]**

The first paragraph (L. 120-123) doesn't serve well as an introduction to this section because the framework for temperature estimation is unclear. Additionally, the section reads more like a collection of individual sub-models rather than a coherent methodology.

Author reply: Added "A coherent methodology for projecting future temperatures is illustrated in Figure 1 (highlighted in pale red and yellow). This figure shows that surface temperatures were

calculated based on long-term warming trends, climatic cycles, and abrupt events (Events 1–16, established as analogs to the "Big Five" mass extinctions) using Equations 1–10 presented in this section. Mantle temperature is treated as a controlling factor for abrupt events (Equations 4–9).".

The most significant gap is clarifying how the mass extinction events are timed and situated within the 2.5 Gyr timeline. Are they periodic, random, or triggered by a specific threshold? This is a key factor affecting the results and remains undefined.

Author reply: Added "The same pacing of icehouse–greenhouse cycles (0.3 Gyr intervals) and major abrupt mass extinctions (0.094 Gyr intervals) is applied to future temperature projections. The uncertainties associated with these cycles are used to estimate the timing of the complete extinction of metazoans, with error margins of approximately ±0.1 Gyr for icehouse–greenhouse cycles and ±0.03 Gyr (1 SD) for mass extinction timing.".

Several critical numbers and equations in this Section are presented without explanation, making it impossible for a reader to evaluate or reproduce your work:

1) Why is there an 8 °C temperature difference between the icehouse and greenhouse periods? (L. 129) Bond and Grasby (2017) are probably insufficient because their study focuses on specific events. Using a single value over 2.5 Gyr is a big oversimplification that needs strong justification. Therefore, the author should provide a brief context.

Author reply: Revised to "To incorporate these cycles, I applied a temperature anomaly of 8 °C between icehouse and greenhouse intervals, based on Phanerozoic temperature reconstructions by Scotese et al. (2021), to the long-term thermal evolution model of Mello and Friaça (2019). However, this temperature anomaly is expected to gradually decrease as atmospheric CO2 declines due to enhanced continental weathering. This follows the well-known climate model result that a doubling of CO2 produces a 2.3 °C increase in surface temperature (Manabe and Wetherald, 1967).".

Manabe and Wetherald, 1967, Thermal Equilibrium of the Atmosphere with a Given Distribution of Relative Humidity. J. Atmospheric Sci. 24, 241-259.

2) These equations (L.156-161) are the core of your event-based temperature calculation, but they are presented as a "black box." Where do the constants, such as -8.9, 9.9, 2/3, 9.9, 0.00108 ..., originate from? The author cites Kaiho et al. (2022), but it is unclear whether these are empirically derived relationships from that paper or if the author has manipulated their data. This is the most critical part of the section to expand on.

Author reply: Added "-8.9 and 9.9 represent the average cooling and post-event warming associated with the five major mass extinctions, respectively.", "Equations (6) and (7) represent

best-fit curves derived from these experiments." Tm is the mantle temperature for each geologic age", "The modern mantle temperature is 1603°K (Mello and Friaça, 2019), and the average initial sill temperature is 1423°K (Aarnes et al., 2010); the 180°K difference represents average cooling during sill formation.".

- 3) "A 50% reduction in  $CO_2$  and a 67% reduction in  $SO_2$  emissions" (L. 167) are striking figures. How were they calculated? Are these the results of your model or the inputs into the model? Author reply: Revised to "A 65% reduction in  $CO_2$  and a 76% reduction in  $SO_2$  emissions are projected to significantly lower surface temperature anomalies over the next 1.5 Gyr, corresponding to  $SR \times 100$  and  $CR \times 100$  at 1.5 Gyr from now (Table A2). Using the emission models of Black et al. (2018) for  $CO_2$  and Schmidt et al. (2016) for  $SO_2$ , the impacts of declining emissions were integrated to estimate their effects on future global temperature evolution."
- 4) The author mentions the basis for applying "the 0.3 Gyr climate cycle to the next billion years" (L.126-127), but further explanation is needed. What is the physical mechanism? Additionally, why do its period and amplitude remain constant over billions of years, despite geological changes such as increased solar activity and/or mantle cooling?

Author reply: Added "As the 0.35–0.30 Gyr interval may vary in the future due to mantle cooling and increased solar activity, these variations produce a maximum difference of 0.1 Gyr in the timing of each metazoan diversity phase. This uncertainty was used to constrain the onset of the decline phase and the final demise phase. The 0.1 Gyr difference arises from the relatively short duration of individual icehouse phases, approximately 0.1 Gyr."

5) The decision to model the impact of declining mantle temperatures only for the next 1.5 Gyr (L. 168) seems arbitrary. Why not consider the entire 2.5 Gyr?

Author reply: The 2.5 Gyr timeline includes 1.0 Gyr of past data and 1.5 Gyr representing the future period analyzed. Mantle temperature was used to estimate future surface temperatures. Because surface temperature data for the past are already available, calculations based on mantle temperature were applied only to future projections.

Revised to "A 65% reduction in  $CO_2$  and a 76% reduction in  $SO_2$  emissions are projected to significantly lower surface temperature anomalies over the next 1.5 Gyr encompassing the entire period analyzed. These reductions correspond to  $SR \times 100$  and  $CR \times 100$  at 1.5 Gyr from now (Table A2)."

6) The term "sill" (Ts, Ti) is used without being defined for a non-geology specialist. A brief explanation is necessary.

Author reply: Revised to "Ts is the sill (a tabular sheet intrusion from magma)".

7) The formulation of Equation (10),  $Er = SD + Er(\Delta Tc)$ , is problematic. It defines the total error, Er, in terms of itself, creating circular logic. Furthermore, it adds a standard deviation (SD), a statistical measure of dispersion, to an error term,  $Er(\Delta Tc)$ , which likely represents a potential range. The description implies these components are being combined, but the equation's additive form is not logically justified. Clarification is required on the functional form of  $Er(\Delta Tc)$ —is it a root-sum-square, a linear addition, or another method?

Author reply: This section and the thin error bars in Figure 2 have been deleted because  $Er(\Delta Tc)$  decreases as the temperature anomaly diminishes through the long-term climate cycle, approaching 2°C at the time of metazoan disappearance. Added "The model uncertainty for the black dushed line ranges between -4°C and +1 (Mello and Friaça, 2019). The model uncertainty for the orange line is 2°C at maximum during the Climate Phase C depending on timing of icehouse. These uncertainties can lead to a range of 0.8–1.2 Gyr for complete extinction of metazoans for most possible projection of 0.9–1.0 Gyr." in the subsection 4.5 Model limitations and uncertainties. "Including uncertainties in climate projections, the timing of total metazoan extinction is estimated at 0.9–1.0 billion years, with a broader uncertainty range of 0.8–1.2 billion years." in the abstract.

Author reply: Added "The 0.1 Gyr difference arises from the relatively short duration of individual icehouse phases, approximately 0.1 Gyr." in 2.3.1 section.

this refers to an uncertainty in age or temperature.

9) Concerning Equation 8, the derivation from the Arrhenius equation should be briefly explained or referenced.

Author reply: Added "Equation (8) is modified from the Arrhenius equation on page 174 of PAC (1996) (Supplementary Information).".

Added the two references: "The activation energies (E) are 74 kcal/mol for *n*-C16 alkane (representing CO2 release; Jackson et al., 1995) and 67 kcal/mol for pyrite (representing SO2 release; Concer et al., 2017)."

**[Methods: 2.6]**

The section outlines a logical and organized approach to determining critical thermal thresholds for metazoans, which is a key and valuable contribution. The method of converting physiological limits into the global average surface temperature required for extinction (GATE) is ambitious and clearly segmented by habitat. However, the description currently needs significant clarification and justification to meet the standards of reproducibility:

Regarding the 5 °C adjustment from the Local Monthly Maximum Temperature (LMMT) to the Local Daily Maximum Temperature (LDMT) in Section 2.6.1, point 3, why 5 °C? Is this a global average? Is it based on the Weather Spark data cited? This needs a clear justification.

Author reply: Revised to "These lines are shifted downward by 6 °C to reflect the average Local Daily Maximum Temperature (LDMT) in oceanic climate zones during the warmest month (6 °C represents the typical difference between the daily maximum temperature and the monthly average temperature during the warmest month, based on data from coastal cities such as Singapore (~0° N), Shanghai (~30° N), and Helsinki (~60° N) (Weather Spark, <a href="https://weatherspark.com">https://weatherspark.com</a>) (dashed oblique lines in Figure 3).". Revised 5 °C to 6 °C in related figures and tables.

Regarding the 1 °C and 5 °C Local Annual Temperature (LAT) adjustments in Section 2.6.1, point 4, the cited reference (Upchurch et al., 1998) is somewhat outdated. Is this still the most relevant source? Please briefly explain the climatic reasons behind this latitudinal gradient in temperature differences.

Author reply: Added "a rare reference showing monthly temperature difference, Upchurch et al. (1998)". Added "Latitudinal temperature gradient is due to changes in the Earth's axial tilt and solar incidence angle.".

The selection of 37° latitude for determining GATE in Section 2.6.1, point 5, appears arbitrary as stated. Why 37°? Is this based on a global average temperature weighting? This step is crucial in determining your final GATE value and must be clearly explained.

Author reply: Added "because GAT equals LAT at 37°N latitude in oceanic climate regions of the modern Earth (14°C GAT and 14°C LAT at Iwaki and San Francisco (Weather Spark, https://weatherspark.com)"

The assumed future burrowing depth of 2.5 meters in Section 2.6.2 is a significant biological assumption. Why choose 2.5 m instead of 1 m or 5 m? Is there evidence that metazoans can or will burrow to this depth for thermoregulation? This requires ecological justification.

Author reply: Revised to "At this depth, the temperature difference between the warmest month and the annual mean is about 2 °C—sufficiently low for survival—whereas at a depth of 1 m, the difference reaches approximately 8 °C, which is too high for survival (Singh and Sharma, 2017). At depths of 4 m or more, soil temperature becomes nearly constant.".

The step-by-step process in Section 2.6.1 can be confusing because it mixes the description of what is shown in the figure with the actual methodological steps. It would be clearer to organize it as a standalone explanation. If possible, the author should rewrite 2.6.1 as a numbered or bulleted

list that is independent of the figure: 1) Define the upper thermal tolerance (46ÅãC) as the average maximum daily temperature (...) at key latitudes (0°, 30°, 60°, 90°). 2) Establish the warm Earth latitudinal temperature gradient (15 °C from 0° to 90°). 3) Adjust this gradient for LMMT by applying a -5 °C correction. 4) Adjust LMMT to LAT using a latitudinally variable correction (ΔLT). 5) Calculate the GATE from the LAT at a representative latitude (3°N/S). Author reply: Added "The Global Average surface Temperatures required for the Extinction of St metazoans (GATES), as shown in Figure 1, were determined through the following steps:

- 1. Define the upper thermal tolerance (46 °C) as the average Local Daily Maximum Temperature (LDMT) in oceanic climate zones during the warmest month at key latitudes (0°, 30°, 60°, 90°).
- 2. Establish the warm Earth latitudinal temperature gradient (15 °C from 0° to 90°).
- 3. Adjust this gradient for Local Monthly Maximum Temperature (LMMT) by applying a -6 °C correction.
- 4. Adjust LMMT to Local Annual Temperature (LAT) using a latitudinally variable correction (ΔLT).
- 5. Calculate the GATES from the LAT at a representative latitude (37°NS)."

The text in Section 2.6.1, point 5 states, "The GATE values for St and Sw metazoans are the same as described in Section 2.3.3." However, since your structure indicates that 2.3 is about oxygen levels, this likely refers to a different part of your manuscript, which could cause confusion. All necessary information for understanding the calculation should be in this section or clearly cross-referenced.

Author reply: Removed this sentence, because the new sentence is above 5. Calculate the GATES from the LAT at a representative latitude (37°NS).

The frequent and inconsistent use of acronyms makes the text very difficult to understand. Readers must constantly refer back to remember what they stand for. Therefore, if possible, please consider adding a table: a summary that lists each metazoan group, its defined upper thermal limit, all applied corrections, and the final derived GATE value would greatly enhance clarity and reproducibility.

**Author reply:**

1. Added the following table.

**Table 1.** Method summary on surface temperatures required for extinctions of four metazoan groups

| Metazoan | Upper thermal | Applied     | Representative | GATE00, 30, 60, | 0 41    |
|----------|---------------|-------------|----------------|-----------------|---------|
| group    | limit (°C)    | corrections | latitude for   | 90 (°C)         | Section |

|                |    |                  | GATE |                |       |
|----------------|----|------------------|------|----------------|-------|
| Superterranean | 46 | LDMT-            | 27°N | 22 26 20 42    | 0.4.4 |
| (St)           | 46 | 6°C−ΔLT          | 37°N | 33, 36, 39, 42 | 2.4.1 |
| Subterranean   | 46 | depth correction | 37°N | 38, 43, 48, 53 | 2.4.2 |
| (U)            | 40 | depth correction | 37 N | 30, 43, 40, 33 |       |
| Surface water  | 46 | LDMT-            | 37°N | 22 26 20 42    | 2.4.3 |
| (Sw)           | 40 | 6°C−ΔLT          | 37 N | 33, 36, 39, 42 |       |
| Doop water (D) | 46 | 46°C at 45°      | 37°N | 40             | 2.4.4 |
| Deep-water (D) |    | latitude         | 31 N | 48             | 2.4.4 |

GATE: Global Average surface Temperatures required for the Extinction. LDMT: average Local Daily Maximum Temperature. 6°C: difference between LDMT and LMMT. LMMT: Local Monthly Maximum Temperature. ΔLT: difference between LMMT and LAT. LAT: annual average temperatures.

- 2. Added "The future diversity model consists of alternating extinction events and recovery phases (Table 3). Extinction events include climate-driven crises, plant crises (first the C3 plant crisis, followed by the C4 plant crisis), and food scarcity events that occur once surface-dwelling metazoans disappear. Each of these events contributes to substantial losses in biodiversity. Recovery phases occur after each event; however, the recovery rate varies through time. Before the C3 plant crisis, recovery rates are set to 1.0, reflecting complete biodiversity recovery. After the onset of the C3 plant crisis, recovery rates decline below 1.0 due to the combined effects of long-term global warming and progressive reductions in atmospheric oxygen and CO2, which diminish the resilience of ecosystems and limit their capacity to fully recover." after the title of the section 2.6.
- 3. The section "2.6 Integrated Future Diversity Model" has been reorganized into three subsections—2.6.1 Events, 2.6.2 Recoveries, and 2.6.3 Scenario Variations—to clarify the structure and improve readability, making the relationships among the acronyms easier to understand.
- 4. Revised the following table to understand the method structure.

**Table 3.** Extinction and recovery model for future projections, showing diversity change rates at the family level

| Event  |        |              | For Event (short terr    | n)                       | For Recovery (long term) |                         |                    |  |
|--------|--------|--------------|--------------------------|--------------------------|--------------------------|-------------------------|--------------------|--|
| Climat |        | Age
(Gyr) |                          | Food
Scarcity
Rate | Recovery Rate (RR)       |                         |                    |  |
| е      | Future |              | Survival Rate by Climate |                          | by gradual warming (RRW) | by O 2 (RRO) | By CO 2 |  |
|        | event  |              | (SRC)                    |                          |                          |                         | (RRC) &            |  |
|        |        |              |                          |                          |                          |                         | [NC]               |  |

|   |                         |      | Insect | Tetrapod
s | Marine animals | (FS)     | Terrestria
I | Marine | Terrestrial | Marine | for
All    |
|---|-------------------------|------|--------|---------------|----------------|----------|-----------------|--------|-------------|--------|---------------|
| E | 16                      | 1.45 | 0.0    | 0.0           | 0.0            | 0.001    | 0.0             | 0.0    | 0.0         | 0.0    | 0.0           |
| E | 15                      | 1.35 | 0.0    | 0.0           | 0.0            | 0.001    | 0.0             | 0.0    | 0.0         | 0.0    | 0.0           |
| E | 14                      | 1.26 | 0.0    | 0.0           | 0.0            | 0.001    | 0.0             | 0.0    | 0.0         | 0.0    | 0.0           |
| D | 13                      | 1.16 | 0.001  | 0.001         | 0.0            | 0.01     | 0.007           | 0      | 0-0.001     | 0-0.01 | 0.01-0.1      |
| D | Oxyge
n              | 1.07 | 0.001  | 0.001         | 0.001          | 0.01     | 0.025           | 0.33   | 0-0.001     | 0-0.01 | 0.01–0.1      |
| D | 12                      | 1.07 | 0.007  | 0.007         | 0.165          | 0.01-0.1 | 0.166           | 0.424  | 0-0.001     | 0-0.01 | 0.01-0.1      |
| D | 11                      | 0.97 | 0.018  | 0.018         | 0.33           | 0.01-0.1 | 0.525           | 0.665  | 0.001-0.01  | 0.09   | 0.01-0.1      |
| С | C 4
plant | 0.97 | 0.01   | 0.01          | 0.01           | 0.01–0.1 | -               | _      | 0.001-0.01  | 0.09   | 0.01–0.1      |
| С | 10                      | 0.88 | 0.044  | 0.044         | 0.34           | 0.01-0.1 | 0.83            | 0.84   | 0.01-0.05   | 0.26   | 0.3-0.8       |
| С | 9                       | 0.78 | 0.27   | 0.27          | 0.49           | 1        | 1               | 1      | 0.07        | 0.44   | 0.3-0.8       |
| С | 8                       | 0.69 | 0.53   | 0.53          | 0.67           | 1        | 1               | 1      | 0.38        | 0.62   | 0.3-0.8       |
| В | 7                       | 0.6  | 0.81   | 0.63          | 0.74           | 1        | 1               | 1      | 0.67        | 8.0    | 0.3-0.8       |
| В | 6                       | 0.5  | 0.81   | 0.63          | 0.74           | 1        | 1               | 1      | 1           | 1      | 0.3-0.8       |
| В | 5                       | 0.41 | 0.81   | 0.63          | 0.74           | 1        | 1               | 1      | 1           | 1      | 0.3-0.8       |
| Α | C₃
plant             | 0.4  | 0.5    | 0.5           | 0.5            | 1        | 1               | 1      | 1           | 1      | 0.3–0.8       |
| Α | 4                       | 0.31 | 0.81   | 0.63          | 0.74           | 1        | 1               | 1      | 1           | 1      | 1             |
| Α | 3                       | 0.22 | 0.81   | 0.63          | 0.74           | 1        | 1               | 1      | 1           | 1      | 1             |
| Α | 2                       | 0.12 | 0.81   | 0.63          | 0.74           | 1        | 1               | 1      | 1           | 1      | 1             |
| Α | 1                       | 0.03 | 0.81   | 0.63          | 0.74           | 1        | 1               | 1      | 1           | 1      | 1             |
| Α | 0                       | 0.00 | 0-0.95 | 0-0.7         | 0-0.9          | 1        | 1               | 1      | 1           | 1      | [0 -1] |

NC: nuclear war

**[Methods: 2.7]**

The three main equations of the core model are shown, but their logic and connections are not clearly explained. Equation (13) presents a problem in this recovery model. Dt-2 refers to diversity two time steps earlier, implying that recovery aims to reach a diversity level similar to what was before the last extinction event, which may not be ecologically realistic. A more common approach would be to target a recovery toward a carrying capacity or a pre-event level. The reasoning behind

this needs further clarification. Equation (14) appears to apply to gradual diversity loss preceding an event. It's unclear why this is a separate equation from equation (12) and how the "Survival Rate for gradual changes" (SR) differs from SRC in practice. The model's flow between these equations is not described.

Author reply: Revised to " $D_{t+1} = D_t + (D_{t-1} - D_t) \times RR$  ( $D_t$  is diversity at an event)"  $D_{t-1}$  is a pre-event diversity corresponding to "a pre-event level" in the comment. Revised the explanation to "In these equations,  $D_t$  represents metazoan diversity at time step t, corresponding to the level immediately following an extinction event.  $D_t$ 1 denotes the pre-event diversity, while  $D_t$ 1 reflects the diversity after the recovery phase, measured at the midpoint between extinction events."

This is a critical issue that significantly hampers readability. The text uses over 20 different acronyms (SRC, SRO, RR, SR, SAR, SARS, SARU, SARD, SRCT, SRCM, StR, SwR, UR, DR, FS, GATE, etc.). Many are non-intuitive, and some are confusingly similar (e.g., SR vs. SRC vs. SRO; SARS as "Survival Area Rate" is an unfortunate choice). The author should create a nomenclature table before Section 2.7, including a list of all acronyms, their full names, and brief definitions. Otherwise, please avoid creating an acronym for every term. Use them sparingly for the most frequently used concepts.

Author reply: The above two tables contain only important acronyms with explanation. This may help to understand the method structure.

Many of the numerical values used in the model seem arbitrary or lack a clear, reproducible source. This presents the biggest risk to the model's credibility.

Regarding SRC for Event 0 (Anthropocene), the values of 0.95, 0.70, and 0.90 for a "full-scale nuclear war" are very precise. What is the basis for these exact numbers? The citation (Kaiho, 2023) must specifically provide these values or the model that produced them.

Author reply: Revised to "For Event 0 (the Anthropogenic Crisis), the maximum SRC values were set at 0.95 for insects, 0.80 for terrestrial tetrapods, and 0.90 for marine metazoans, based on Figure 1 of Kaiho (2022). These values were determined using 30% terrestrial and 20% marine species extinction data from Kaiho (2025b) and converted from species-level to family-level extinction percentages following the relationships illustrated in Figure 1 of Kaiho (2022).". Revised 0.70 to 0.80. 0.95 is from comparison of the insect extinction rate and that of the other groups.

Regarding Food Scarcity (FS), a reduction of "0.1–0.5" is a very broad range. How is a specific value selected for a particular calculation? This requires a clear rationale.

Author reply: Revised to "These events are characterized by the extinction of superterranean and surface-water (SS) metazoans and severe reductions in primary productivity due to sunlight loss—conditions common in major mass extinction scenarios (see Fig. 2). A low Food Scarcity Rate value of 0.01 reflects survival through alternative nutritional pathways, including hydrothermal vent ecosystems and bacterial-based underground food sources (Cosson and Soldati, 2008; Miroshnichenko, 2004; Kelley et al., 2005). Conversely, a high Food Scarcity Rate value of 0.1 represents an optimistic estimate, assuming evolutionary adaptation of primary producers to extreme temperatures, allowing some limited ecosystem function to persist. This range (0.01–0.1) is applied to adjust survival estimates in scenarios where abrupt collapse of food webs occurs due to light inhibition and temperature stress.".

Concerning "Underground Rate (UR=0.05) and Deep-water Rate (DR=0.33)," the logic for converting modern lineage proportions (e.g., 15 out of 315 mammalian families) into a future survival rate is not clearly explained. This is a major assumption that requires a strong ecological justification.

Author reply: Revised to "However, these rates (0.05 and 0.33) may vary under future global warming trends. Subterranean metazoan families may increase relative to superterranean metazoans because of lower maximum soil temperatures, whereas deep-water metazoan families are expected to decline due to reduced dissolved oxygen levels associated with global warming. These adjustments are incorporated into both the decline and demise phases. Equations (15) and (16) are used to model extinction scenarios for Events 5–7, when premass extinction surface temperatures remain below the Metazoan Thermal Tolerance Limit across all latitudes, allowing recovery of metazoan diversity. Equations (17) and (18) are applied to Events 8–11, where pre-extinction surface temperatures exceed the Metazoan Thermal Tolerance Limit in certain latitudes. However, because metazoan diversity approaches near zero during these events, changes in the ratios of metazoan groups have no significant effect on overall diversity patterns.".

For "Recovery Rates (RR)," many RR values (e.g., 0.01-0.3 for events 8-10) are given without a clear, quantitative link to environmental conditions. The justification is qualitative ("reduced capacity," "adaptation challenges"), but the output is a precise number. There needs to be a transparent method for converting the severity of an event into a recovery rate.

Regarding the "C4 Plant Crisis" (L. 287), the timeline is set to "coincide approximately with event 11," and the extinction of metazoans is then "set at 0.97 Gyr." This appears to be a circular

argument where the model is adjusted to fit a pre-selected outcome rather than the outcome emerging naturally from the model's mechanics. Is it true?

Author reply: The age is from Mello and Friaça (2019). Added (Mello and Friaça, 2019), resulting in "The timing of the  $C_4$  plant crisis is set at  $0.97 \pm 0.2$  Gyr, closely coinciding with Event 11, based on averaged estimates from Mello and Friaça (2019) and Ozaki and Reinhard (2021)."

The Methods section should describe how you calculated diversity, not what the results are. This section often overlaps with presenting results and speculative explanations, such as ... expected to decline (L. 276), ... are expected to evolve (L. 280), and ... are expected to include (L. 290). The author should rephrase these statements to describe the model's rules.

**Author reply: Revised to**

L. 276: The timing of the  $C_4$  plant crisis is set at 0.97  $\pm$  0.2 Gyr, closely coinciding with Event 11, based on averaged estimates from Mello and Friaça (2019) and Ozaki and Reinhard (2021). L. 280: The Recovery Rate by Primary Productivity (RRP) is tentatively set between 0.3 and 0.8, reflecting the evolutionary and ecological challenges involved in re-establishing complex, tree-supported food webs derived from  $C_4$  vegetation. This estimate is based on the projected evolution of  $C_4$  plants into tree-like forms approximately 40 million years after the extinction of  $C_3$  plants and the subsequent recovery of metazoan groups originating from species formerly dependent on  $C_3$  vegetation. This recovery event is centered at 0.4 Gyr, with a temporal uncertainty of  $\pm$ 0.2 Gyr (Table 2).

L. 290: The timing of the  $C_4$  plant crisis is set at 0.97 ± 0.2 Gyr, closely coinciding with Event 11, based on averaged estimates from Mello and Friaça (2019) and Ozaki and Reinhard (2021). At this stage and beyond, RRP values decline to between 0.01 and 0.1, as NPP is assumed to approach zero. Under such conditions, only UD metazoans—those subsisting on bacteria, detritus, or residing in deep-sea hydrothermal ecosystems—would persist (Cosson and Soldati, 2008; Miroshnichenko, 2004; Kelley et al., 2005). This sharp reduction in RRP reflects the critical dependence of most metazoans on photosynthetically sustained food webs.

**[Results: 3.1]**

The most significant issue is the lack of a methodological link between your methods and your primary result. The "Results" section should present findings derived from the previously described methodology; however, the description of the temperature curve reads more like an input scenario than a calculated outcome. The core problem is that the reader is told what the temperature curve is, but not how it was generated. While Section 2.6 details a model for calculating extinction

thresholds, it does not explain the foundational climate model that produced the temperature projection itself. The author must explicitly state the model or data source used to generate the orange curve in Figure 1. Is it an output from a climate model run under specific CO2 scenarios? An extrapolation from past climate data? Or is it derived from astronomical solutions, such as Milankovitch cycles? Crucially, you must identify the primary forcing driver (e.g., increasing solar luminosity, greenhouse gas concentrations) to provide the necessary context.

The same time periods and temperature ranges are described multiple times in slightly different ways (e.g., 0.65-0.95 Gyr vs. 0.7-1.0 Gyr). This repetition is redundant and causes confusion. Paragraph 2 (L. 352-355) describes the trend with specific future time points (0.35, 0.65, 0.95 Gyr), while Paragraph 4 (L. 359-363) restates the trend by re-binning the timeline into Phases A-E. Therefore, the author should combine these into a single, clear description, using the climate phases as the main structural framework and avoiding disconnected time points.

The nature of the abrupt climate changes described in Events 5, 8, 11, and 14 in paragraph 3 (L. 356-358) remains unclear. What physical process causes these "temperature surges" of more than 10ÅaC in"<0.1 million years"? Are they inputs to the model (prescribed forcing) or results (emergent behavior)? The text currently treats them are given. We need to determine whether these are hypothetical events, representations of volcanic or tectonic activity, or outcomes from climate tipping points.

The use of "expected" and "projected" is inconsistent. Since this is a model result, "projected" is more appropriate.

The final paragraph states that Phases B-E are "triggered by" abrupt events (L. 364). This creates a causality dilemma. Is the long-term trend the primary driver, or are the abrupt events the triggers for new phases? The current description makes it seem like both, which is confusing. The author must clarify the relationship between the gradual trend and the abrupt events.

Author reply: Revised the subsection 3.1 to "The orange curve in Figure 2 represents a projected trajectory of global average surface temperature derived by extrapolating long-term climate trends. This projection incorporates sustained warming driven by increasing solar luminosity, together with cyclic climate oscillations occurring at 0.35–0.30 Gyr intervals, which are associated with variations in atmospheric greenhouse gas concentrations. Abrupt temperature perturbations linked to mass extinction events are not included in the orange curve, allowing it to represent the smooth background evolution of Earth's climate. The time axis spans from –1.0 Gyr to +1.5 Gyr, with values calculated at 0.05 Gyr intervals.

During the Phanerozoic era (-0.54 Gyr to the present), global average surface temperatures fluctuated between approximately 15°C and 25°C, a range projected to persist until ~0.35 Gyr (Climate Phase A), aside from brief perturbations associated with major extinction events. Beyond this interval, global temperatures are projected to increase gradually: to 25–30°C by ~0.7 Gyr (Climate Phase B), 30–40°C between ~0.7 and 1.0 Gyr (Climate Phase C), 40–50°C from ~1.0 to 1.2 Gyr (Climate Phase D), and ultimately reaching ~70°C by ~1.5 Gyr (Climate Phase E).

Superimposed upon this background trend are abrupt climate events associated with large igneous province volcanism and major impact events. Based on the frequency and magnitude of past events, 16 such abrupt events are projected to occur over the next 1.5 Gyr (Section 2.6). These events produce temporary deviations from the long-term temperature curve, shown as light blue (cooling phase) and red (subsequent warming phase) points in Figure 2. These temperature perturbations reflect combined influences of volcanic SO2-induced cooling followed by CO2-driven warming.

Within Climate Phase A, abrupt cooling events reduce global temperature to approximately 10–14°C, followed by warming peaks of 30–33°C—closely resembling patterns documented in the Phanerozoic record. During Climate Phase B, cooling minima of 16–18°C and warming maxima of 33–35°C are projected. Subsequent phases show progressively higher temperature bounds: 22–26°C and 38–43°C in Phase C, 28–36°C and 45–52°C in Phase D, and 39–55°C and 56–72°C in Phase E.

Among these events, Events 5 and 8 exhibit particularly rapid temperature increases (<0.1 Myr), representing abrupt transitions from icehouse to greenhouse states. These short-duration warming surges are interpreted as responses to exceptionally large volcanic CO2 emissions or impact-triggered atmospheric perturbations."

**[Results: 3.2]**

The first paragraph states diversity is "estimated using" (L. 368) several factors but provides no explanation of how this estimation is performed. What is the model? Is it a statistical correlation, a dynamic ecosystem model, or a set of threshold rules? What are the "survival thresholds" (mentioned in 3.2.1) and how are they determined? The reader cannot assess the results without understanding the fundamental rules of the model.

Author reply: Revised to "Based on the calculation results on diversity, the timeline between -1.0 and 1.5 Gyr is divided into five metazoan phases: Ancestor phase (Climate Phase A), Evolution with mass extinctions (Climate Phase A), Decline with mass extinctions (Climate Phase B), Demise by

mass extinctions (Climate Phase C), and Aftermath (Climate Phases D and E) (Figs. 5 and 6)." The methods are explained in the revised method subsection 2.6.

Several claims are presented as facts without justification, moving from scientific projection to pure speculation. "Large volcanic eruptions and asteroid impacts are expected to trigger..." (L. 382-383) The timing and occurrence of these specific events cannot be projected. This must be framed as a scenario or sensitivity test like "In scenarios where large volcanic eruptions occur...".

Author reply: Revised to "In scenarios involving large volcanic eruptions or asteroid impacts, severe global cooling is induced—comparable to that observed during past mass extinction events—followed by prolonged warming phases."

The line between results and interpretation is often unclear. A Results section should present the data or model outputs), while the Discussion should explain what they mean. In this section, phrases like "This high-temperature environment will also impact..." or "will drive the complete collapse..." are interpretive. The text often states what will happen as if it is certain, based on the model, rather than just presenting the model's output. Therefore, the author should reframe to simply present the model's findings.

Author reply: Revised to "Event 5 is expected to coincide with the C3 plant crisis, leading to a major decline in biodiversity. Shallow-sea (SS) metazoans in low latitudes will be unable to survive, and those in mid-latitudes will be partially affected, as extreme global surface temperatures are projected to average around 40 °C annually (Fig. 2)."

The author provides us with specific numbers (e.g., insect families dropping from 610 to 74–103), but does not inform us how these numbers were generated. This breaks the chain of reproducibility. Therefore, when presenting a key result, briefly link it back to the methodological framework.

Author reply: The methods are written in the subsection "2.6.1 Events". Added equation numbers.

The text in this section mentions ranges (e.g., 74–103 insect families) and different scenarios (Figs. 5 and 6), but it doesn't explain the causes of these ranges. For a projection of this magnitude, a thorough exploration of uncertainty is crucial. The author needs to state which parameters are responsible for the ranges explicitly.

Author reply: Added. "The following four parameters cause uncertainty shown in Figures 5 and 6: 1) food scarcity rate for events 8–10, 2) recovery rate by gradual warming for events 8–13, 4) recovery rate by oxygen for events 7–13, C4 plant crisis, and oxygen depression event, and 5) net primary productivity for event 0, C3 plant crisis, events 8–13, and C4 plant crisis (Table 2)."

Given the highly speculative nature of billion-year projections, using definitive language like "will," "are expected to," or "lead to" is too strong for a Results section. The author should adopt more

tentative and precise phrasing that reflects the model-dependent nature of the findings. For example, the author can use phrases such as: "The model projects...", "Under the defined scenarios...", "Simulation results indicate...", and "Our findings suggest...".

Author reply: Added "Simulation results indicate that ---"

The "Aftermath" section (L. 415-428) extends beyond metazoan diversity to explore the fate of all life and the planet. While a compelling conclusion, it exceeds the stated scope of "metazoan diversity change." The author should consider whether some of this material, especially the comparison to Venus, might be better suited for the Discussion.

Author reply: Removed the sentences.

**[Discussion: 4.]**

This section currently reads more like a summary and extension of the results than a critical discussion. To meet the journal's standards, the work needs to be more thoroughly addressed, including the limitations, uncertainties, and broader implications, shifting from "what we found" to "what it means and how reliable it is."

Section 4.1 (L. 430-460) mainly defends the assumptions rather than thoroughly examining their uncertainties. A robust discussion must openly acknowledge and analyze the weaknesses. Some specific gaps include: 1) The model incorporates highly uncertain components (billion-year climate, biodiversity, oxygen, plant evolution). The combined effect of these uncertainties is not addressed. How reliable is the 0.7 Gyr extinction date, considering this? 2) The model assumes fixed thermal tolerances. The discussion should explicitly consider the possibility of evolutionary adaptation over millions of years, even if the conclusion suggests it might be limited due to fundamental physiological reasons. 3) The survival and recovery rates are key but poorly constrained. A discussion on how sensitive the main conclusion (extinction at ~0.7 Gyr) is to these parameters is essential. Is the outcome unavoidable across a wide range of plausible values? The author can integrate a dedicated subsection, like "4.5 Model Limitations and Uncertainties," that systematically addresses these points. This demonstrates scholarly rigor and strengthens the paper.

Author reply: Added the following section.

**4.5 Model limitations and uncertainties**

The model uncertainty for the black dashed line in Figure 2 is estimated to range from -4 °C to +1 °C (Mello and Friaça, 2019). For the orange curve, the maximum temperature uncertainty is approximately +2 °C during Climate Phase C, depending on the timing of the icehouse–greenhouse

transitions (see Figure 2 caption). When propagated into extinction timing, these temperature uncertainties correspond to a projected range of ~0.8–1.2 Gyr for the complete disappearance of metazoans, with the most likely range between 0.9 and 1.0 Gyr.

Mantle cooling in the model results in approximately 84% and 81% of total  $CO_2$  release occurring during Events 7 and 8, respectively (Table A2), producing global warming of ~8.3 °C and ~8.0 °C. When this warming coincides with long-term greenhouse states and a major abrupt extinction event, global temperatures can exceed the upper thermal tolerance of metazoans, resulting in irreversible biodiversity loss.

The timing of complete metazoan extinction is also influenced by the occurrence of icehouse phases. If an icehouse state develops around 0.6 Gyr, extinction is projected to occur during Event 8 or 9 (0.7–0.8 Gyr). If it occurs earlier ( $\sim$ 0.5 Gyr), extinction is expected during Event 8 or 10 (0.7–0.9 Gyr). If later ( $\sim$ 0.7 Gyr), extinction coincides with Event 9 ( $\sim$ 0.8 Gyr. Although these scenarios shift timing slightly, the overall temperature difference between icehouse and greenhouse states is only  $\sim$ 2–3 °C due to declining CO2 (Fig. 2), meaning the range of extinction timing remains relatively narrow.

The recurrence interval of abrupt events is  $\sim$ 0.1 Gyr, consistent with the spacing of past major mass extinctions. Thus, the exact timing of total metazoan extinction depends on whether these abrupt events align with warm phases in the long-term climate cycle. Additionally, progressive oxygen decline and the  $C_4$  plant crisis may independently drive complete metazoan extinction near  $\sim$ 1.0 Gyr (Figs. 1, 5).

The model assumes fixed thermal tolerance thresholds for metazoans. Although evolutionary adaptation is possible over millions of years, it is constrained by fundamental biochemical limits. Even if survival under extreme heat is feasible, reproductive success is expected to collapse once temperatures exceed ~46 °C for tens to hundreds of thousands of years, due to protein destabilization and reduced developmental viability. Survival and recovery rates affect diversity patterns during the decline phase (Figs. 5–6) but do not change the ultimate outcome once thermal thresholds are surpassed (Table A5).

In summary, when accounting for uncertainties in long-term climate evolution, abrupt climatic events, CO2 feedbacks, oxygen decline, and ecosystem collapse, the complete extinction of metazoans is projected to occur between ~0.7 and ~0.9 Gyr from the present (Fig. 1), with a broader uncertainty range extending to ~1.2 Gyr.

The author often restates results (e.g., "The combined effects... will drive a substantial decline") and ventures into highly speculative territory (e.g., Section 4.3 on intelligent life strategies and Mars)

without a clear framing. For example, Section 4.3, while interesting, is tangential to the core scientific findings about metazoan diversity. It risks detracting from the paper's focus and lacks the scientific support found in other sections. Thus, the author reframes result restatements as a setup for interpretation. For speculative sections, clearly label them as such and connect them directly to the model's projections.

**Author reply: Removed the section.**

While citations are used, the discussion often uses them to support the model's assumptions rather than to contrast or synthesize the model's findings with other published work. Therefore, the author should actively engage with alternative viewpoints or models, positioning their work within the ongoing scientific conversation.

Author reply: Added the following section.

**4.6 Timing of biosphere collapse**

Mello and Friaça (2019) suggested that biosphere collapse is unlikely to occur before  $\sim$ 1.5 Gyr due to the upper thermal limits of life. However, a decline in atmospheric oxygen to  $\sim$ 1% PAL is projected to occur at 1.08  $\pm$  0.14 Gyr (1 $\sigma$ ) (Ozaki and Reinhard, 2021), potentially leading to a much earlier extinction of metazoan life. Broader geosphere—biosphere models therefore estimate a remaining biosphere lifespan of at least  $\sim$ 0.8 Gyr, with  $\sim$ 1.2 Gyr being a reasonable median value (Jebari and Sandberg, 2022).

The results of this study indicate that total metazoan extinction is expected to occur at approximately 0.9–1.0 Gyr from the present, with an uncertainty range of 0.8–1.2 Gyr. This range is consistent with previous biosphere-lifespan estimates. A key new finding of this study is that abrupt climatic events accelerate the extinction timeline by approximately 0.3 Gyr.

If abrupt climate events are *not* considered, global average temperature (GAT) would reach the upper thermal limit for metazoans only at ~1.2 Gyr, at which point declining atmospheric oxygen and CO2 would become the primary drivers of metazoan extinction. However, when abrupt warming events are included, rapid temperature spikes—combined with ongoing oxygen and CO2 decline—lead to significantly earlier biodiversity collapse. In this scenario, extreme global warming acts as the principal trigger of extinction, rather than slow geochemical feedbacks alone.

The central message that metazoans will decline after 0.4 Gyr and go extinct at 0.7 Gyr due to combined temperature,  $O_2$ , and plant crises is repeated multiple times (e.g., in 4.1, 4.1.2, 4.2, 4.4). This repetition reduces its impact. The author should simplify the main narrative, state the key conclusion clearly once, and then use the following sections to discuss different aspects of it (limitations, mechanisms, implications) without restating the conclusion verbatim.

**Author reply: Removed them.**

There is a clear copy-paste error in Section 4.4 (L. 566-574), where a whole paragraph is duplicated ("Rising Global Temperatures at Events 8–10...").

Author reply: Removed the repetition.

**[Conclusion: 5.]**

The second half of the conclusion goes far beyond the paper's scientific findings into policy, ethics, and futurology. The author's statements, such as "To mitigate biodiversity loss, advanced species may need to implement strategies..." and "emphasizing the urgency of proactive measures," are not conclusions derived from your model. While interesting, the author should phrase it more neutrally. The author should focus the conclusion on the scientific findings. The speculative strategies (aerosol shielding, space colonization) can be mentioned as potential consequences of the findings, but not as prescriptive "needs." Avoid language that tells the reader what is "urgent."

**Author reply: Removed them.**

The entire study relies on a model with significant uncertainties (as noted in comments for the discussion). The conclusion states the 0.7 Gyr timeline as a definitive result without any qualification. The author should consider adding a sentence that acknowledges these inherent uncertainties.

Author reply: Added "Prior to final extinction, biodiversity will continue to repeatedly decline and recover following mass extinction events. However, from approximately 0.4–0.6 Gyr onward, a long-term downward trend is expected, driven by sustained global warming and progressive reductions in atmospheric CO2 and O2, although the precise timing remains uncertain. The final collapse is likely to occur before the C4 plant crisis and severe atmospheric oxygen depletion predicted in other studies, because volcanic warming under already elevated baseline temperatures will exceed physiological limits. While metazoan lineages with greater heat tolerance may evolve, even the highest known thermal thresholds would eventually be surpassed. Sustained global temperatures above ~46 °C, at which protein denaturation and reproductive failure become critical, are expected to persist for tens of thousands of years during the terminal warming phase, leading to irreversible population decline and ultimate extinction, if there is no evolution of new organisms using heat-resistant protein."

The phrase "this study is the first to reveal that humanity exists at the midpoint of metazoan lifespan" makes a very strong claim in the conclusion. While it might be true, it sounds self-promoting. The most suitable place to highlight this novelty is in the Introduction and Discussion

sections. The author could rephrase this to emphasize the finding itself. For example, "An interesting corollary of this timeline is that humanity appears near the midpoint of Earth's metazoan history."

Author reply: Added "An interesting corollary of this timeline is that humanity appears near the midpoint of Earth's metazoan history." in the discussion section 4.3.

The main point, extinction at 0.7 Gyr, is repeated three times in a very brief text. While repetition can emphasize a point, here it limits the development of a more nuanced final message. The author can condense the key finding into one strong statement at the beginning and use the remaining space to discuss its causes and implications.

Author reply: I wrote the time once in this section.

**2. Author replies for comments from Anonymous Referee #2 Summary**

In this research article, Kunio Kaiho presents novel findings on the future development of metazoan diversity in superterranean, subterranean, surface-water, and deep-water habitats based on diversity changes in the past. By incorporating seven different environmental drivers, the author projects the com-plete extinction of metazoans within the next 700 million years, which is 300–400 million years earlier than previously estimated.

**General comments**

Overall, the manuscript is well written and provides novel insights into an important field of research. The language is almost perfect, clear, and easy to follow. However, there are a few general points that should be addressed before final publication of the article.

Neither the Introduction nor the Discussion provides much context regarding previous research efforts. While the Introduction nicely explains the different environmental drivers incorporated into the current study, it is unclear what previous research entailed and what the current study adds to it. These aspects should be included in the revised manuscript.

Author Reply: Added in the Introduction. Estimates for timing of the end of Earth's biosphere published after 2010 vary widely. Projections based on the long-term warming trend range from 1.0 to 5.0 Gyr (O'Malley-James et al., 2012; Rushby, 2013; Leconte et al., 2013; Wolf and Toon, 2015), while CO2 depletion scenarios yield lower estimates of 0.84–1.08 Gyr (Rushby, 2015; Ozaki and Reinhard, 2021). Mello and Friaça (2019) argue that biosphere collapse is unlikely before 1.5 Gyr

based on thermal constraints. However, a projected decline in atmospheric oxygen to 1 % PAL within  $1.08 \pm 0.14$  Gyr ( $1\sigma$ ) (Ozaki and Reinhard, 2021) could result in an earlier extinction of metazoan life. In summary, current geosphere—biosphere models suggest a remaining biosphere lifespan of at least 0.8 Gyr, with 1.2 Gyr being a plausible median estimate (Jebari and Sandberg, 2022).

Similarly, the Discussion repeats the major results of the current study without discussing them in the context of previous findings. For example, it is repeatedly mentioned throughout the manuscript that the current study projects metazoan extinction to occur 300–400 million years earlier than previous estimates, but these previous estimates are not further specified. What differences between previous studies and the current study may cause these different results? Why are the results of the current study more/similarly realistic? These questions should be addressed in the Discussion.

Author Reply: Added in the Discussion.

**4.6 Timing of biosphere collapse**

Mello and Friaça (2019) suggested that biosphere collapse is unlikely to occur before ~1.5 Gyr due to the upper thermal limits of life. However, a decline in atmospheric oxygen to ~1% PAL is projected to occur at  $1.08 \pm 0.14$  Gyr ( $1\sigma$ ) (Ozaki and Reinhard, 2021), potentially leading to a much earlier extinction of metazoan life. Broader geosphere—biosphere models therefore estimate a remaining biosphere lifespan of at least ~0.8 Gyr, with ~1.2 Gyr being a reasonable median value (Jebari and Sandberg, 2022).

The results of this study indicate that total metazoan extinction is expected to occur at approximately 0.9–1.0 Gyr from the present, with an uncertainty range of 0.8–1.2 Gyr. This range is consistent with previous biosphere-lifespan estimates. A key new finding of this study is that abrupt climatic events accelerate the extinction timeline by approximately 0.3 Gyr.

If abrupt climate events are *not* considered, global average temperature (GAT) would reach the upper thermal limit for metazoans only at ~1.2 Gyr, at which point declining atmospheric oxygen and CO2 would become the primary drivers of metazoan extinction. However, when abrupt warming events are included, rapid temperature spikes—combined with ongoing oxygen and CO2 decline—lead to significantly earlier biodiversity collapse. In this scenario, extreme global warming acts as the principal trigger of extinction, rather than slow geochemical feedbacks alone.

In addition, I think that some parts of the Methods section are difficult to follow. Firstly, this section uses many abbreviations, but not all of them are defined in the text itself, only in figure/table captions (e.g., PAL is only defined in the caption of Fig. 1). Secondly, many terms are unclear to the

reader and require further explanation (e.g., what exactly are diversity rates and what is the difference between survival rates and survival area rates?). Thirdly, the argumentation is partly difficult to follow since the required explanations are either insufficient or provided later in the Results or Discussion section. I recommend adding further explanations and revising the structure of the manuscript where necessary. I give specific examples in the "Specific comments" section. Author Reply: Revised the method section largely as shown in author replies for reviewer 2 in pages 3–21.

**Specific comments**

• \_L. 29: I would replace "all known forms of life" by "almost all known forms of life", e.g., tardigrades can survive temperatures higher than 100°C.

**Author Reply: Done**

L. 32: Can you shortly explain what C3 and C4 plants are?

Author Reply: Revised to " $C_4$  plants generally exhibit higher photosynthetic efficiency under hot and dry conditions, making them more resilient to low  $CO_2$  levels than  $C_3$  plants. Consequently,  $C_3$  plants, which include most trees, are expected to decline first ( $C_3$  plant crisis), followed by a  $C_4$  plant crisis, ultimately leading to a collapse in metazoan diversity (Reinfelder et al., 2000, 2004; Mello and Friaça, 2019; Ozaki and Reinhard, 2021)."

• \_L. 52-53: This sentence disrupts the flow of the text. Since the corresponding information was just mentioned a few paragraphs earlier, the sentence is not necessary in my opinion.

Author Reply: Removed the sentence: As solar luminosity increases, intensified weathering will-reduce atmospheric CO2, initiating crises for C3 plants, then C4 plants.

• \_L. 95: "These data are applied in section 2.2." – The current section is 2.2, so I do not understand this sentence.

Author Reply: Revised to section "2.5.2".

• \_Section 2.3: Are only records of marine metazoans available? If yes, the possible impacts of this limitation should be discussed.

Author Reply: Added "and terrestrial".

L. 97: What exactly is meant by "diversity rates"?

Author Reply: Revised to "Projections of future atmospheric oxygen levels and marine and terrestrial metazoan diversity were informed by records from the Paleozoic biodiversity maximum.".

L. 98: What is PAL?

Author Reply: Revsied to "present atmospheric levels (PAL)" in Introduction.

L 106-107: I think you mean that oxygen levels drop in the habitats of metazoans and not in

metazoans themselves, right?

Author Reply: Removed ", with significant declines occurring when oxygen levels drop in both marine and terrestrial metazoans"

• \_L. 108: Why do terrestrial metazoans require an ozone layer for evolutionary adaptation? (This is explained in I. 491-493, but I would already explain it here).

Author Reply: Revised the sentence to: In section 4.2.2: Declining atmospheric oxygen levels will further increase extinction risks, particularly for terrestrial organisms. As the ozone layer thins due to reduced oxygen concentrations, terrestrial metazoans will face greater exposure to ultraviolet radiation, which is more harmful than in marine environments. This pattern is already evident in historical diversity records, as indicated by the delayed appearance of terrestrial metazoans compared to marine metazoans (Fig. 4, Table 1).

- \_L. 110-115: What were the main reasons for mass extinction during these events?

  Author Reply: In Introduction: In contrast, Earth's climate has also shifted episodically due to unpredictable abrupt events. Abrupt climate perturbations—such as large-scale volcanism and asteroid impacts—have repeatedly triggered mass extinctions by rapidly altering the climate system through emissions of SO2, soot, and CO2. These events cause short-term global cooling followed by prolonged warming (Kaiho, 2025) and have resulted in substantial biodiversity losses over the past 0.5 Gyr (Erwin et al., 1987; Sepkoski, 1996; Bambach, 2006; Kaiho, 2022). In subsection 2.6.1 events: The primary causes of major past mass extinctions are considered to include global cooling, global warming, ozone layer destruction, ocean acidification, and oceanic anoxia.
- \_L. 131: There is no red curve in Fig. 1. Do you mean the orange curve? Author Reply: Yes, I revised it to orange.
- \_L. 147: Which were the five largest mass extinction events?

Author Reply: Revised to "five major mass extinctions". Five major mass extinctions were defined in Introduction section.

- \_L. 152: I thought ΔTec was estimated using SST data as stated in the previous section? Author Reply: Yes, it was.
- L. 163: What is sill?

Author Reply: Revised to "Ts is the sill (a tabular sheet intrusion from magma)".

• \_L. 167-168: How are long-term changes in CO2 and SO2 emissions related to short-term temperature anomalies?

Author Reply: This sentence does not refer to long-term trends, but rather to abrupt changes that trigger major mass extinctions. I applied a gradual decrease in mantle temperature to estimate

future SO2 and CO2 emission rates relative to present-day levels. These rates were then used to calculate extinction percentages.

L. 191: Why do you use regions with oceanic climate?

Author Reply: Revised to "The model presented in Figure 2a estimates extinction thresholds for St metazoans (i.e., surface-dwelling terrestrial animals) based on oceanic climate regions **on land**, as these regions generally experience milder climates compared to continental interiors."

• \_L. 196: Does a gradient of 15°C only apply to warm conditions or why do you explicitly mention warm conditions here?

Author Reply: Revised to "Establish the warm Earth latitudinal temperature gradient (15 °C from 0° to 90°) to estimate magnitude of extinctions due to global warming."

• L. 199: Where exactly do the 5°C come from?

Author Reply: Revised to 6°C. "Adjustment for Local Monthly Maximum Temperature (LMMT):

These lines are shifted downward by 6 °C to reflect the average Local Daily Maximum Temperature (LDMT) in oceanic climate zones during the warmest month (6 °C represents the typical difference between the daily maximum temperature and the monthly average temperature during the warmest month, based on data from coastal cities such as Singapore (~0° N), Shanghai (~30° N), and Helsinki (~60° N) (Weather Spark, <a href="https://weatherspark.com">https://weatherspark.com</a>) (dashed oblique lines in Figure 3)."

- \_L. 201: Same as I. 191 and I. 196: Why do you use data from warm coastal cities?

  Author Reply: Delete "warm" resulting in "coastal cities". Coastal temperature records are used because the warmest daily temperatures in coastal regions are generally lower than those in inland areas, providing a conservative threshold for determining conditions that would be lethal to all metazoans (dashed oblique lines in Figure 3).
- \_L. 207: There is no section 2.3.3. Do you mean 2.6.3?

Author Reply: Yes, 2.6.3.

• \_Equation 11: What is ΔLT? (LT is only defined in the caption of Fig. 2)

Author Reply: Revised to "where LT (Local Temperature) is the latitude-dependent adjustment from LMMT to LAT."

• \_Sect. 2.7.1: This section is quite hard to follow since many abbreviations are used. Maybe it would help to spell out the abbreviations from time to time.

Author Reply: I simplified them as described in Author replies for comments from Reviewer 2 in pages 16-18.

• \_L. 244-248: I think it should already be mentioned here that different scenarios are analyzed.

Author Reply: Removed "Before mass extinction events, SRC events values follow gradual warming

**trends as described in Table 2.".**

• \_L. 249-252: I think it would be helpful to provide a brief description of the different events. Some description is given in Sect. 3.2, but I believe that including such a description earlier on would give the reader a better understanding.

Author Reply: Added the equation and calculation in each event.

- \_L. 252: What exactly is the survival area rate and what is the difference to the survival rate? Author Reply: SAR is rate of land area where metazoans survive in all land area (km²/km²). Added "Survival Area Rate (SAR) is rate of land and ocean area where metazoans survive in all land and ocean area (km²/km²). When extinction occurred in 0–10, 0–20, 0–30, 0–40, 0–50, 0–60, 0–70, 0–80, and 0–90° latitudes by warming, SAR values are defined as 0.83, 0.66, 0.50, 0.36, 0.24, 0.14, 0.06, 0.02, 0.00, respectively, under the same rate of land and ocean in those latitudes. The rates SAR are decided by only temperature 46 °C using Figure 2." I revised SAR to EAR (Extinction Area Rate). EAR values are 0.17, 0.34, 0.50 -------
- \_L. 256-257: What exactly do the rates StR, SwR, UR, and DR describe? Author Reply: I do not use them in the present manuscript.
- \_L. 259: Could you explain more clearly how the SAR is calculated?

Author Reply: Added "Extinction Area Rate (EAR) is rate of land and ocean area where metazoans survive in all land and ocean area (km²/km²). The rates EAR are decided by only temperature 46 °C using Figure 3. When extinction occurred in 0–10, 0–20, 0–30, 0–40, 0–50, 0–60, 0–70, 0–80, and 0–90° latitudes by warming, EAR values in species (EARS) are defined as 0.83, 0.66, 0.50, 0.36, 0.24, 0.14, 0.06, 0.02, 0.00, respectively, under the same rate of surface land and surface ocean in those latitudes. EARS values are transferred EARS to EARF using Figure 1b and 1c of Kaiho (2022).".

- \_L. 263: I cannot follow the argumentation here. Why should SARD approximate SARS?

  Author Reply: Added "High surface temperatures can cause extinction in both surface and deepwater taxa. Although deep water temperatures are lower than those at the surface, the greatest thermal anomalies occur in surface waters, while deep-water temperatures remain relatively constant throughout the water column.". I do not use SARD in the present manuscript.
- \_L. 268: How did you determine the impact of food scarcity on survival rates?

  Author Reply: Revised to "Once all surface-dwelling metazoans become extinct, diversity is further reduced by applying the Food Scarcity Rate (FS), reflecting the restriction of metazoan survival to deep-sea ecosystems supported by bacterial and hydrothermal vent food webs (Equation 12)."
- · L. 275: And the other events?

Author Reply: Added the explanation.

- \_L. 285-286: Is this reasonable? The limitations of this assumption should be discussed. Author Reply: Revised to "Although oceanic primary producers are predominantly phytoplankton, both C3 and C4 photosynthetic pathways coexist in marine environments (Reinfelder et al., 2000, 2004).".
- \_L. 287-293: I cannot follow here. Why are metazoans extinct at 0.97 Gyr if 2% remain? And don't you state in other parts of the manuscript (e.g., the Abstract, I. 460, and the Conclusions) that according to your calculations, metazoans go extinct at 0.7 and not 0.97 Gyr? Author Reply: Revised to "The timing of the  $C_4$  plant crisis is set at  $0.97 \pm 0.2$  Gyr, closely coinciding with Event 11, based on averaged estimates from Mello and Friaça (2019) and Ozaki and Reinhard (2021).". The revised manuscript indicates the terrestrial insects, tetrapods, and shallow-marine animals are projected to disappear approximately 0.9 billion years from now, while deep-sea organisms may persist slightly longer, to about 0.9–1.0 billion years. Including uncertainties in climate projections, the timing of total metazoan extinction is projected to 0.9–1.0 billion years, with a broader uncertainty range of 0.8–1.2 billion years. The primary driver of this final extinction is projected to be extreme global warming during an abrupt volcanic event, with progressive declines in atmospheric  $CO_2$  and  $O_2$  acting as secondary, long-term contributing factors.
- \_L. 304: What exactly is numerical age?

  Author Reply: Revised to "In these equations, *T* is the numerical time variable in Gyr, ranging from 0.5 to 1.0."
- \_Sect. 3.1: I think this description would have been more helpful in the Methods section somewhere between Sects. 2.5 and 2.7. Then the reader could better understand the different survival rates etc.

**Author Reply: Done.**

• \_L. 364: What exactly do you mean by abrupt climate events? Are you referring to volcanic eruptions and meteorite impacts? If yes, I recommend stating this here again.

Author Reply: Revised to "Climate Phases B–E are likely to be initiated by abrupt climate disturbances caused by large volcanic eruptions or meteorite impacts occurring during greenhouse intervals."

\_L. 385-388: Could you also give current numbers for comparison?

Author Reply: Revised to "In both the minimum and maximum cases, during Events 1–4 (occurring between 0.03 and 0.40 Gyr in the future), the model projects a sharp decline in biodiversity—from

current estimates of 610 insect families, 315 terrestrial tetrapod families, and 950 marine metazoan families to 494, 198, and 703 families, respectively, based on equations (XX). These declines are followed by complete recovery to present-day diversity levels (Fig. 5; Table A5). In the NC Min Case, Event 0 (the Anthropogenic Crisis) results in reduced survival, with 580 insect families, 221 terrestrial tetrapod families, and 855 marine metazoan families remaining, followed by recovery to current diversity levels (Fig. 5, Table A5). However, if the recovery rate after Event 0 is zero—as modeled in the NCC Min Case—biodiversity continues to decline due to sustained anthropogenic impacts on the biosphere. In this scenario, even during non-extinction intervals, the surviving numbers of families are projected to remain at 580 for insects, 221 for terrestrial tetrapods, and 855 for marine metazoans (Fig. 6, Table A5). The NCC Max Case also results in equivalent proportional reductions compared to the Max Case."

\_L. 592: Are you sure that your study is the first to reveal that?

Author Reply: Removed "is the first to".

- \_Fig. 1:
- o I do not understand what the green open diamond symbols denote exactly. Can you maybe explain again in other words?

Author Reply: Revised to "Green open diamond symbols denote temperatures before mass extinctions."

o Would it be possible to add some sort of legend to the atmospheric oxygen level graph that specifies the impact on metazoans?

Author Reply: Added "Impact on metazoans" in Figure 1.

• Fig. 2 (l. 630): Not only silhouettes of terrestrial plants are shown, so maybe write "silhouettes of metazoans and terrestrial plants"?

Author Reply: Revised to "Silhouettes of metazoans indicate their approximate diversity and corresponding oxygen levels, whereas silhouettes of terrestrial plants indicate only corresponding oxygen levels."

- Fig. 3: The yellowish-green is hard to distinguish from the green, so maybe use a different color? Author Reply: Revised it to yellow.
- Table A2: What is SD? What does aftermath warming mean?

Author Reply: Revised SD to standard deviation. Revised Aftermath warming to Warming.

L. 720 and 725: before or after?

Author Reply: Revised to "Underlined and double underlined numbers indicate values corresponding to before and after extinction events, respectively."

• \_L. 727-728: Why do the underlined numbers occur just before major mass extinction events if they represent periods of recovery? Something seems wrong here.

Author Reply: No, the underlined numbers occur just after major mass extinction events.

**Technical corrections**

L. 43: have historically triggered historical mass extinctions

Author Reply: Revised to "have triggered past mass extinctions"

• \_L. 61: This manuscript was written by only one author, right? I would use "I" instead of "we"; there are other occurrences throughout the manuscript.

Author Reply: Revised to "I" for all.

· L. 89 and 96: I think it should be "Past records of"

**Done**

• \_L. 204: LMmMT

**Done**

L. 213: 2.5 meters

**Done**

• L. 222: GATES values are common equal

**Done**

• L. 237: where, Dt represents

**Done**

• L. 324: 1 meter

**Done**

• \_L. 378: "However" does not seem to fit here.

**Deleted "However".**

\_L. 398: primary productivity?

**Yes, done**

• \_L. 430: estimating of the future diversity

**Done**

L. 504: illustrates

**Done**

• L. 569-573: This is a repetition of I. 566-569 and should be deleted.

**Done**

• \_Fig. 2:

- o Atmospheric oxygen level relative to for present atmospheric level
- o Diversity rate relative to for the Paleozoic maximum

**Done**

• \_L. 659 and 729: the Methods section

**Done**

• L. 676: The capitalization in this sentence seems odd.

**Done**

• \_L. 686: rates in the future

**Done**

• L. 687-688: for compared to terrestrial plants

**Done**

Table 2: in at the family level

**Done**

• \_Table A2: Earth's average surface temperature of including the long-term trend, long-term cy-cle, and short-term events with temperature anomalyies and decreasing CO2 and SO2 emis-sions decreasing rate due to the decrease in mantle potential temperature during major mass extinction events from 0.7 billion years (Gyr) before the present to 1.5 Gyr into the future Done